EMBO
Molecular Medicine

# Epigenetic dysregulation of IRF9 drives excessive interferon signaling in COPD

Maria Llamazares-Prada [1,2,3,19], Uwe Schwartz [3,4,19], Darius F Pease [5], Stephanie T Pohl [5],
Deborah Ackesson [5], Renjiao Li [5], Annika Behrendt [3], Raluca Tamas [3], Vedrana Stammler [3],
Mandy Richter [3], Thomas Muley [2,6], Michael Scherer [1,2], Joschka Hey [1], Elisa Espinet [7,17],
Claus P Heußel [2,8,9], Arne Warth [2,6,10,18], Marc A Schneider [2,6], Hauke Winter [2,11], Felix JF Herth [2,6,12],
Charles D Imbusch [13], Benedikt Brors [13], Vladimir Benes [14], David Wyatt [15], Tomasz P Jurkowski [5],
Heiko F Stahl [16], Christoph Plass [1,2] & Renata Z Jurkowska [3,5] ✉

## Abstract

**Altered respiratory barrier integrity and impaired lung regeneration are hallmarks of chronic obstructive pulmonary disease (COPD). To investigate the molecular mechanisms driving the impaired regeneration of alveolar epithelial progenitors in COPD, we generated whole-genome DNA methylation and transcriptome maps of sorted human primary alveolar type 2 cells (AT2) at different disease stages. Our analysis revealed aberrant DNA methylation at specific gene promoters in AT2 during COPD, which was anticorrelated with gene expression changes. Interferon signaling was the top-upregulated pathway in COPD, associated with a concomitant loss of promoter-proximal DNA methylation. Integrated pathway analysis revealed transcription factor IRF9 as the master regulator of interferon signaling in COPD. Epigenetic regulation of the interferon pathway was validated by targeted DNA demethylation of the IRF9 gene, mimicking the effects observed in COPD-derived AT2. Our findings suggest that COPD-associated DNA methylation alterations in AT2 cells may impair internal regeneration programs in lung parenchyma.**

**Keywords** COPD; DNA Methylation; Alveolar Type 2 Cells; Interferon Signaling; Epigenetic Editing
**Subject Categories** Molecular Biology of Disease; Respiratory System

## Introduction

Chronic obstructive pulmonary disease (COPD) is the fourth leading cause of death, affecting more than 200 million people worldwide (Safiri et al, 2022). It is a heterogeneous lung disease characterized by irreversible and progressive airflow limitation. COPD is a risk factor for the development of multiple comorbidities, including lung cancer, depression, hypertension, heart failure, rheumatic disease, muscle wasting, and osteoporosis (Stallberg et al, 2016; Yin et al, 2017a). The major risk factor for COPD is cigarette-smoke exposure, but the disease is also influenced by ageing, respiratory infections, air pollutants, biomass fuels, and both genetic and epigenetic factors. At the molecular level, COPD pathology is driven by excessive inflammatory responses, increased oxidative stress, reduced DNA repair, degradome alterations, and apoptosis of lung progenitor cells (Hogg et al, 2004; Calabrese et al, 2005; Sauler et al, 2018; Sakornsakolpat et al, 2019; Sauler et al, 2022; Booth et al, 2023). Overall, these altered processes compromise endogenous lung repair and regeneration pathways, leading to structural changes in the small airways, progressive alveolar destruction (emphysema), and ultimately, airflow obstruction (Hogg et al, 2004; Barnes et al, 2015; Rabe and Watz, 2017; Huang et al, 2019; Booth et al, 2023). Despite the high economic and health burden caused by COPD, there are no curative therapies, and the clinical approaches are directed solely toward symptom relief and exacerbation control (GOLD, 2023). Therefore, understanding the mechanisms that regulate human lung regeneration is essential for defining novel therapeutic strategies aimed at

[1]Division of Cancer Epigenomics, German Cancer Research Center (DKFZ), Heidelberg, Germany. [2]Translational Lung Research Center Heidelberg (TLRC), Member of the German Center for Lung Research (DZL), Heidelberg, Germany. [3]BioMed X Institute, Heidelberg, Germany. [4]NGS Analysis Center Biology and Pre-Clinical Medicine, University of Regensburg, Regensburg, Germany. [5]School of Biosciences, Cardiff University, Cardiff, UK. [6]Translational Research Unit, Thoraxklinik, University Hospital Heidelberg, Heidelberg, Germany. [7]German Cancer Research Center (DKFZ), Heidelberg, Germany. [8]Diagnostic and Interventional Radiology with Nuclear Medicine, Thoraxklinik, University of Heidelberg, Heidelberg, Germany. [9]Diagnostic and Interventional Radiology, University Hospital Heidelberg, Heidelberg, Germany. [10]Pathological Institute, University Hospital Heidelberg, Heidelberg, Germany. [11]Department of Surgery, Thoraxklinik, University Hospital Heidelberg, Heidelberg, Germany. [12]Department of Pneumology and Critical Care Medicine and Translational Research Unit, Thoraxklinik, University Hospital Heidelberg, Heidelberg, Germany. [13]Division of Applied Bioinformatics, German Cancer Research Center (DKFZ), Heidelberg, Germany. [14]Genome Biology Unit, European Molecular Biology Laboratory (EMBL), Heidelberg, Germany. [15]Biotherapeutics Discovery, Boehringer Ingelheim Pharma GmbH & Co. KG, Biberach, Germany. [16]Immunology and Respiratory Disease Research, Boehringer Ingelheim Pharma GmbH & Co. KG, Biberach, Germany. [17]Present address: Department of Pathology and Experimental Therapy, School of Medicine, University of Barcelona (UB), and Molecular Mechanisms and Experimental Therapy in Oncology Program (Oncobell), Bellvitge Biomedical Research Institute (IDIBELL), L'Hospitalet de Llobregat, Barcelona, Spain. [18]Present address: Institute of Pathology, Cytopathology and Molecular Pathology MVZ UEGP Gießen/Wetzlar/Limburg/Bad Hersfeld, Wetzlar, Germany. [19]These authors contributed equally: Maria Llamazares-Prada, Uwe Schwartz. ✉E-mail: jurkowskar@cardiff.ac.uk

restoring lung function and the integrity of the respiratory barrier compromised in COPD.

Alveolar type 2 (AT2) cells are epithelial progenitors of the lung parenchyma essential for distal lung regeneration. Their dysfunction contributes to the development of emphysema in COPD (Barkauskas et al, 2013; Nabhan et al, 2018; Zacharias et al, 2018; Lin et al, 2022). AT2 cells produce and secrete surfactant proteins to maintain lung surface tension and play essential roles in maintaining lung homeostasis and repair by proliferating and differentiating into gas-exchanging alveolar type-1 (AT1) cells. AT2 cells in COPD show signs of increased apoptosis (Kosmider et al, 2011), mitochondrial dysfunction (Kosmider et al, 2019a), DNA damage (Kosmider et al, 2019b), senescence (Tsuji et al, 2006; Tsuji et al, 2010), as well as deregulation of pathways involved in inflammation and lung development, such as interferon (IFN) or Wnt/β-catenin signaling (Fujino et al, 2012; Baarsma and Konigsh-off, 2017; Skronska-Wasek et al, 2017). IFN signaling, which is central to viral immunity, is dysregulated in COPD in immune and structural cells (Lethbridge et al, 2010; Southworth et al, 2012; Briend et al, 2017; Xu et al, 2019; Booth et al, 2023; Rustam et al, 2023), including AT2 cells (Fujino et al, 2012), and contributes to disease exacerbations. Sustained activation of type I and type III interferon pathways is associated with exacerbated lung pathology in respiratory viral infections due to defective repair caused by disruption of epithelial cell proliferation and differentiation (Major et al, 2020). Similarly, tight control of Wnt/β-catenin signaling is essential for AT2 self-renewal and differentiation (Nabhan et al, 2018). Reduced Wnt/β-catenin signaling in AT2 cells in COPD is linked to their decreased self-renewal and repair (Baarsma and Konigshoff, 2017; Skronska-Wasek et al, 2017; Conlon et al, 2020). Therefore, both signaling pathways are central to alveolar regeneration, and their dysregulation in AT2 cells in a chronically injured environment may drive emphysema progression (Conlon et al, 2020). However, the regulatory circuits that drive aberrant gene expression programs in human AT2 cells in COPD are poorly understood.

Epigenetic mechanisms, in particular DNA methylation at CpG sites, are heritable and play a critical role in the regulation of gene expression (Jurkowska and Jurkowski, 2019; Popov and Jurkowska, 2025). During differentiation, ageing, and in response to environmental cues, the epigenome is modified, allowing for major changes in transcriptional programs (Jurkowska, 2024). Alterations in DNA methylation patterns have been implicated in ageing, chronic inflammatory diseases, cancer, and chronic lung diseases (Bergman and Cedar, 2013; Zhao et al, 2021; Liu et al, 2023; Jurkowska, 2024). In addition, cigarette smoke alters DNA methylation in several clinically relevant samples (Belinsky et al, 2002; Chen et al, 2013; Zeilinger et al, 2013; Wan et al, 2015), and is associated with altered expression of genes important in COPD pathology (Liu et al, 2010; Vucic et al, 2014; Wan et al, 2015; Yoo et al, 2015; Morrow et al, 2016; Song et al, 2017; Sundar et al, 2017; Clifford et al, 2018; Casas-Recasens et al, 2021; Schwartz et al, 2023). However, previous studies mostly assessed DNA methylation using heterogeneous material with complex cellular composition (e.g., epithelium, blood, or lung tissue) and focused on selected parts of the genome only (mostly gene promoters). To date, no high-resolution, unbiased DNA methylation profiles of purified, not cultured AT2 cells from COPD lungs are available. Therefore, we set out to profile DNA methylation of human AT2 cells at single-CpG

resolution across COPD stages to identify epigenetic changes associated with COPD. We identified genome-wide remodeling of the AT2 DNA methylation landscape in COPD associated with global transcriptomic changes. Integrative analysis of the epigenetic and transcriptomic data revealed a strong anticorrelation between gene expression and promoter-proximal DNA methylation in COPD AT2 cells, suggesting that aberrant epigenetic changes may drive COPD phenotypes in human AT2.

## Results

### Cohort selection criteria and AT2 isolation for an unbiased profiling study in COPD

To identify epigenetic changes associated with COPD, we collected lung tissue from patients with different stages of COPD, which we stratified into three groups based on their lung function data: (1) no COPD (controls), (2) COPD I [stage I, according to the Global Initiative for Chronic Obstructive Lung Disease (GOLD) (GOLD, 2023)] and (3) COPD II–IV (GOLD stages II–IV) (Fig. 1A,B). There were no significant differences between the control group and the COPD patients for gender, age, BMI, and smoking exposure (pack-year), but as expected, the COPD groups could be clearly separated from the control group based on lung function (Fig. 1B; Dataset EV1). Of note, all donors included in this study (no COPD and COPD) were ex-smokers to avoid acute smoking-induced inflammation as a confounding factor (van der Vaart et al, 2004). Tissue samples fulfilling the inclusion criteria were cryopreserved and subjected to a thorough pathological characterization before AT2 isolation and epigenetic profiling (Fig. 1A–C). This step was critical to avoid potential confounding effects due to the inclusion of samples with additional lung pathologies in the control group, as previously documented (Llamazares-Prada et al, 2021). AT2 cells were isolated by fluorescence-activated cell sorting (FACS) from cryopreserved distal lung parenchyma depleted of visible airways and vessels of three no COPD controls, three COPD I and five COPD II–IV patients, as previously described (Fujino et al, 2011; Fujino et al, 2012; Chu et al, 2020) (Fig. 1D–G). We achieved AT2 purity values ranging between 95 and 97%, as indicated by FACS reanalysis of the sorted cells (Fig. 1G). The isolated cells were positive for HT2-280, a known AT2 marker (Gonzalez et al, 2010), as confirmed by immunofluorescence (Fig. 1H), validating the identity and high enrichment of the isolated AT2 populations.

### The epigenome of AT2 cells is severely altered in COPD

To identify genome-wide DNA methylation changes associated with COPD in purified human AT2 cells, we performed tagmentation-based whole-genome bisulfite sequencing (T-WGBS) (Wang et al, 2013) (Fig. 1A; Dataset EV2). High-quality global DNA methylation profiles at single CpG resolution were generated from 10 to 20 thousand FACS-sorted AT2 cells (Dataset EV2). No global changes in DNA methylation levels were observed between COPD and control samples when looking at genome-wide CpG methylation frequency, but highly methylated CpG sites (> 90% methylation) were significantly underrepresented in COPD II–IV (Fig. 2A). Unsupervised principal component analysis (PCA) of the

most variable CpG sites revealed a separation of COPD II–IV from no COPD on the first principal component (Fig. 2B), suggesting that variation in DNA methylation across the samples is associated with COPD. COPD samples from donors with a cancer background clustered together with COPD samples from lung resections, confirming that we detected COPD-relevant signatures (Fig. 2B).

As DNA methylation is spatially correlated and methylation changes in larger regions are more likely to have a biological function, we did not perform differential analysis of individual CpG sites but instead focused on differentially methylated regions (DMRs) between no COPD and COPD II–IV. Considering only CpG sites with at least 4× coverage in all samples, we identified 25,028 DMRs with at least 10% methylation difference between no COPD and COPD II–IV AT2 cells (see "Methods" for details of DMR calling, Dataset EV3). DMRs contained on average 11 CpG sites and were ~640 bp in length, indicating that large regions are altered (Fig. EV1A,B). Among the identified DMRs, 6,063 regions showed DNA methylation gain in COPD (hypermethylation, 24% of all DMRs), while 18,965 regions displayed DNA methylation loss (hypomethylation, 76% of DMRs), indicating a more open and permissive chromatin landscape in COPD (Fig. 2C). Overall, we identified widespread, region-specific differences in DNA methylation patterns between primary AT2 cells isolated from COPD patients and no COPD controls.

## Epigenetic changes in AT2 cells occur in regulatory regions

To better understand the functional role of aberrant methylation in COPD, we investigated the distribution of DMRs across the genome. Both hypo- and hypermethylated DMRs were predominantly located in intronic and intergenic regions (Fig. 2C). Notably, compared to the genomic background, both types of DMRs were overrepresented at regulatory and gene coding sequences, with hypomethylated DMRs showing the highest enrichment at gene promoters (Fig. 2D). We further intersected the identified DMRs with known regulatory genomic features in lung tissue annotated by the ENCODE Chromatin States from the Roadmap Epigenomics Consortium (Kundaje et al, 2015). When compared to the genomic background, both hyper- and hypomethylated DMRs were overrepresented at enhancers (Enh), and regions flanking transcription start sites (TSS) of active promoters (TssAFlnk) (Fig. 2E). In addition, hypermethylated DMRs were overrepresented at Polycomb-repressed regions (ReprPC) (Fig. 2E). The enrichment of DMRs in gene regulatory regions, including promoters and enhancers, suggests that aberrant DNA methylation in AT2 cells may regulate gene expression in COPD.

To gain insight into cellular processes and pathways affected by aberrant DNA methylation changes in COPD AT2 cells, we linked DMRs to the nearest gene and performed gene ontology (GO) enrichment analysis using the Genomic Regions Enrichment of Annotations Tool (GREAT) (McLean et al, 2010). We identified tube development, epithelial morphogenesis, and Wnt signaling among the top categories for hypermethylated DMRs, while genes with hypomethylated DMRs were associated with regulation of reactive oxygen species (ROS) metabolism, protease activity, and negative regulation of MAPK and ERK1/ERK2 cascades (Fig. EV1C; Dataset EV4). Disease-relevant examples of DMRs include two hypermethylated regions upstream and downstream of the TSS of

the endothelin receptor B gene (EDNRB, 31.3% methylation gain in COPD AT2, Fig. 2F), which could impair the expression of EDNRB in COPD. Furthermore, we found a large hypomethylated DMR region in the first intron of interleukin-33 (IL33, 24,5% methylation loss in COPD AT2, Fig. 2F), an alarmin associated with inflammatory responses and linked to autoantibody production against AT2 cells, exacerbations, and disease severity in COPD (Zou et al, 2018; Allinne et al, 2019; Gabryelska et al, 2019). Further DMR examples are provided in Fig. EV1B.

To investigate whether epigenetic dysregulation may occur early in COPD development and to identify methylation changes associated with mild disease, we included T-WGBS data from AT2 cells isolated from COPD I patients ($n = 3$) and performed k-means clustering on all identified DMRs across all samples (Fig. 2G). Consistent with the unsupervised PCA (Fig. 2B), COPD I samples showed variable methylation changes (Fig. 2G). Donor 34 displayed a methylation profile similar to the control samples, donor 33 showed an intermediate pattern, and donor 4 exhibited a profile resembling that of COPD II–IV patients (Fig. 2G).

## Transcription factor binding sites are enriched at DMRs

Since DMRs were overrepresented at gene regulatory sites, such as promoters and enhancers (Fig. 2E), we performed motif enrichment analysis to footprint transcription factors (TF) that may mediate the effects of aberrant methylation changes in AT2 cells in COPD (Stadler et al, 2011). Overall, 252 transcription factor binding motifs were significantly enriched in the differentially methylated regions (Fig. EV1D; Dataset EV5). The specific enrichment in hypomethylated DMRs was obtained for p53/p63, as well as TFs involved in inflammation control, IFN signaling, cell cycle regulation, and cell fate commitment, including members of the bZIP (Fra1), homeobox (SIX1), and ETS (ELF3) families (Fig. EV1D). Analysis of the hypermethylated DMRs revealed the highest motif enrichment for TFs central to lung development and specification, such as NKX-2 and C/EBP (Boggaram 2009; Miglino et al, 2012), suggesting that their binding and function may be affected by DNA hypermethylation in COPD (Fig. EV1D).

To identify TFs reported to change their binding affinity upon methylation of their recognition sites, we used the motifs generated by a systematic methylation sensitivity analysis (Yin et al, 2017b). Interestingly, the motifs of JUN, ELF, and ETV proteins, which preferentially bind unmethylated regions, were enriched in hypomethylated DMRs, suggesting that these TFs could bind with higher affinity in COPD and thereby regulate transcription of downstream genes (Fig. 2H; Dataset EV5). In contrast, C/EBP TF, which favors binding to methylated CpGs, was enriched in both hypermethylated and hypomethylated DMRs (Fig. 2H), indicating site-specific methylation changes.

Collectively, these results suggest that the aberrant epigenetic makeup of COPD AT2 cells may alter the binding of key TFs associated with inflammation, lung development, senescence, apoptosis, and differentiation, processes strongly implicated in COPD development and progression.

## The global AT2 transcriptome is altered in COPD

The identification of DMRs at gene regulatory sites and the enrichment of transcription factor motifs in the identified DMRs

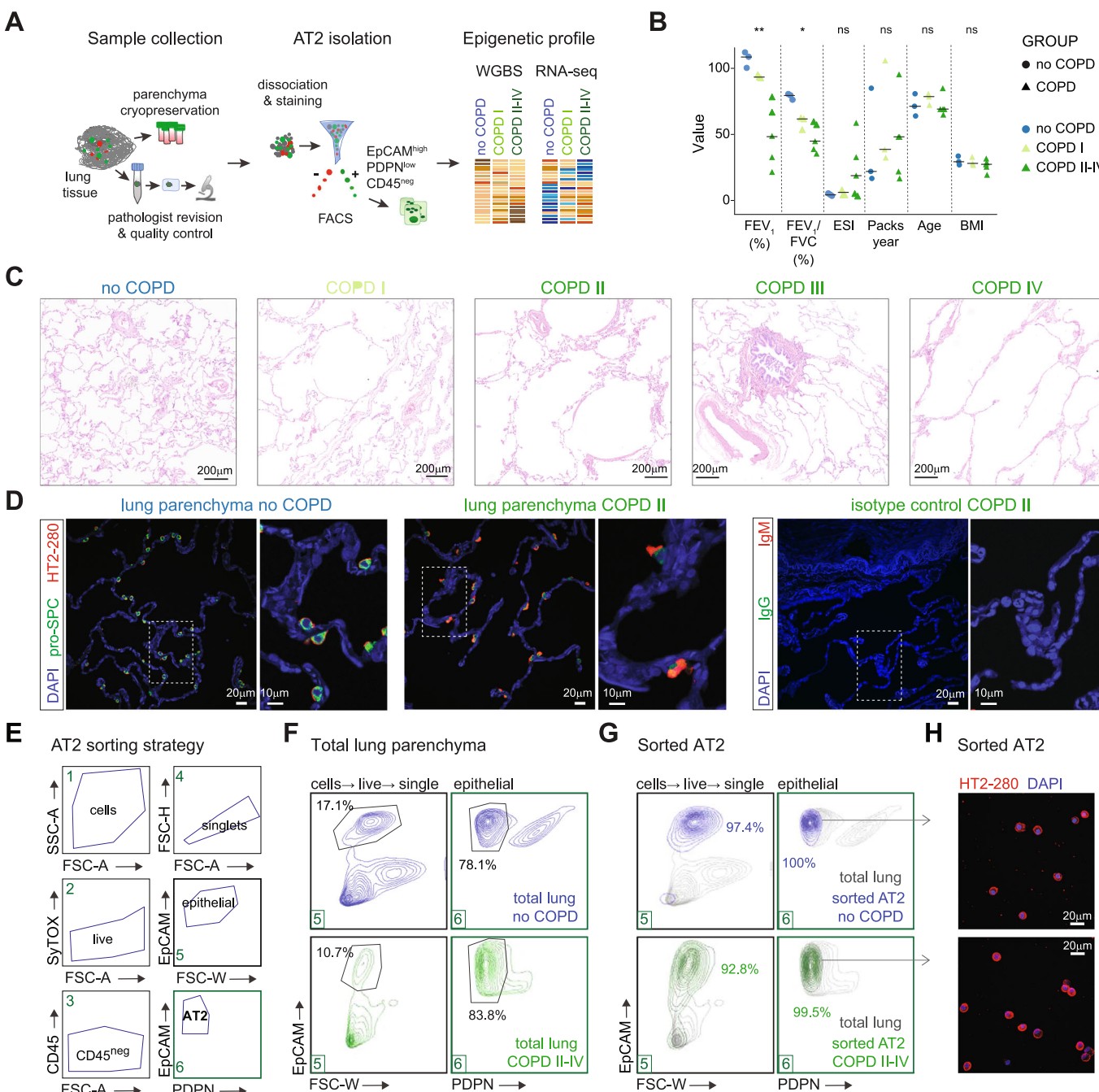

**Figure 1.  Patient selection and AT2 isolation.**

(A) Graphical representation of the experimental approach used in this study, including human lung sample collection, histological characterization, and experimental workflow for profiling. (B) Patient characteristics. Data points represent values for each donor, and horizontal bars denote the group median. Significant values are indicated: ***≤0.001; **≤0.01; *≤0.05. (C) Representative examples of hematoxylin and eosin (H&E) images of human lung tissue samples from the study cohort: no COPD, COPD I, COPD II, COPD III, and COPD IV. Scale bars = 200 μm. (D) Representative immunofluorescence panels of lung parenchyma from no COPD control (left) and COPD (middle) donors showing the presence of AT2 cells (double-positive HT2-280, red and pro-SPC, green) or isotype controls (right). Nuclei are counterstained with DAPI, left scale bar = 20 μm. Right, ×3 magnification from the marked region on the left panels, scale bar = 10 μm. (E) Scheme depicting the sorting strategy employed for isolating AT2 cells from the human lung. (F) Representative FACS plots of AT2 cells sorted from control (blue) and COPD (green) donors. Gates and cell percentages are indicated in the graphs. (G) Example of FACS plots showing the reanalysis of sorted AT2 cells from control (blue) and COPD (green) donors displayed over total cell lung suspensions from (F) (gray). (H) Representative immunofluorescence staining images of HT2-280 expression in sorted AT2 cells from no COPD (top) and COPD (bottom) donors. Nuclei (blue) were stained with DAPI, left scale bars = 20 μm; right = 10 μm. Data information: In (B), data points represent each donor's value, and horizontal bars, the group median. A non-parametric Kruskal–Wallis test was employed to compare the lung function ($FEV_1$ and $FEV_1$/FVC values) between control (no COPD, n = 3), COPD I (n = 3), and COPD II-IV donors (n = 5). Exact P values are: $FEV_1$, P value = 0.0006; $FEV_1$/FVC, P value = 0.0076. $FEV_1$ forced expiratory volume in 1 s, FVC forced vital capacity.

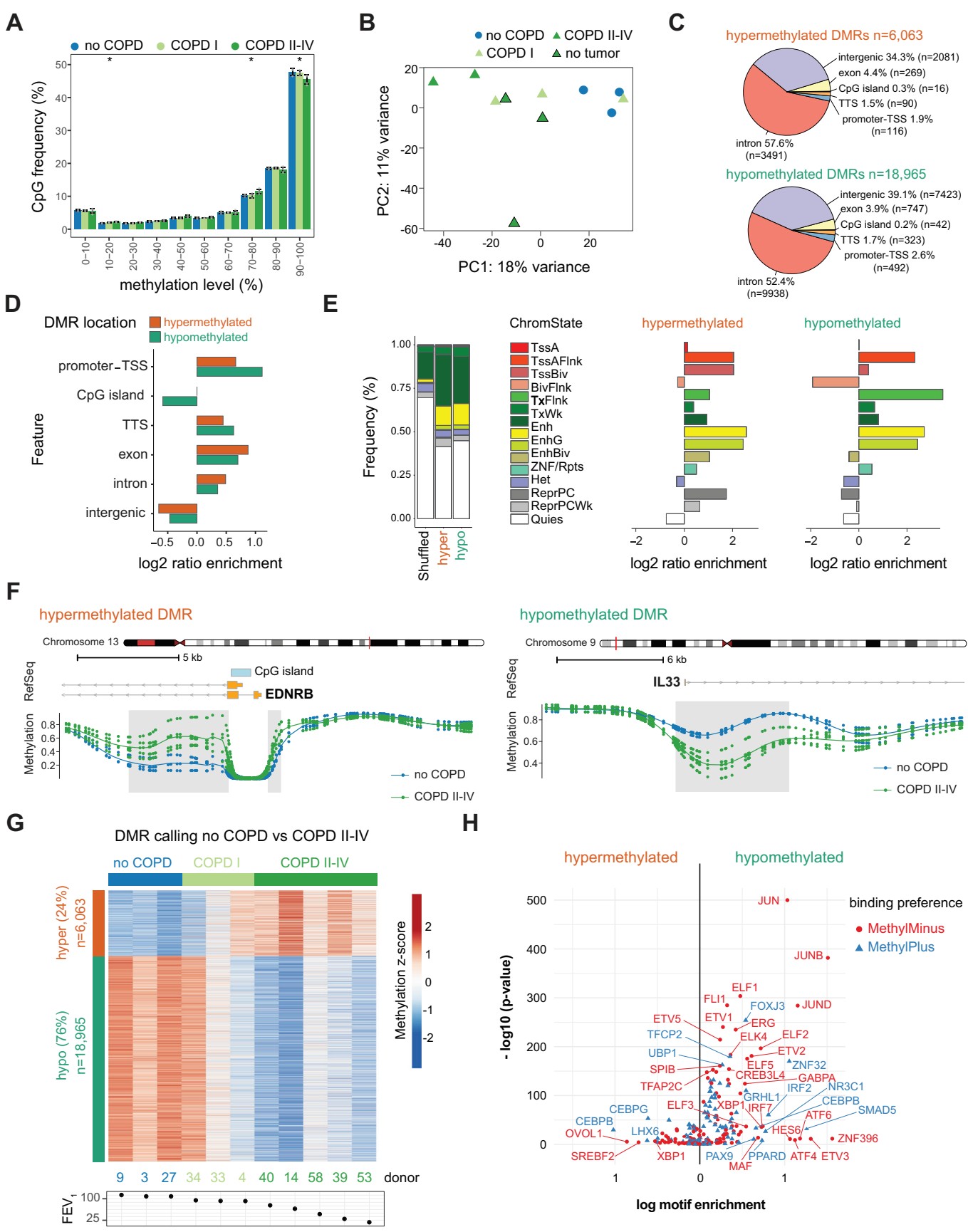

◄

**Figure 2. Genome-wide DNA methylation changes occur at regulatory regions in AT2 cells in COPD.**

Tagmentation-based WGBS DNA methylation data of sorted human AT2 cells from no COPD ($n = 3$), COPD I ($n = 3$), and COPD II–IV ($n = 5$). (A) Genome-wide frequency of CpG methylation levels in no COPD, COPD I, and COPD II–IV. Methylation levels were binned into deciles. Barplots show the group average and the standard deviations across the samples are indicated. *$P$ value < 0.05. (B) Principal component analysis (PCA) of methylation levels at CpG sites with >fourfold coverage in all samples. COPD I and COPD II–IV samples are represented in light and dark green triangles, respectively, and no COPD samples are represented as blue circles. COPD samples without a cancer background are displayed with a black contour. The percentage indicates the proportion of variance explained by each component. (C–E) Genomic feature annotation of differentially methylated regions (DMRs). Distribution of hyper- (top) and hypomethylated (bottom) DMRs at genomic features (C). TSS transcription start site, TTS transcription termination site. Enrichment of genomic features at hyper- (orange) and hypomethylated (green) DMRs compared to the genome-wide distribution (D). Left panel, distribution of human whole lung tissue-specific chromatin states at hypo- and hypermethylated DMRs compared to genome background (shuffled; sampled from regions with matching GC content exhibiting no significant change in methylation) (E). Right panel, chromatin state enrichment relative to the sampled genome background. Abbreviations of chromatin states: Tss Transcription Start Site, A Active, Flnk Flanking, Biv Bivalent, Enh Enhancer, Repr Repressed, PC Polycomb. (F) Detailed view of representative hyper- (left, gain) and hypomethylated (right, loss) DMRs (gray box) showing group average (lines) methylation profile and specific CpG methylation levels (dots) of three no COPD (blue) and five severe COPD donors (GOLD II–IV, dark green). RefSeq annotated genes and CpG islands are indicated, if present. (G) Heatmap of 25,028 DMRs identified in sorted human AT2 cells from COPD II–IV (versus no COPD). Mild COPD (COPD I), which were not used for the DMR calling, are shown within no COPD and COPD II–IV groups. (H) Enrichment of methylation-sensitive binding motifs at hypo- (right) and hypermethylated (left) DMRs. Methylation-sensitive motifs were derived from Yin et al (Yin et al, 2017b). Transcription factors, whose binding affinity is impaired upon methylation of their DNA binding motif (MethylMinus), are shown in red, and transcription factors, whose binding affinity upon CpG methylation is increased (MethylPlus), are shown in blue. Data information: In (A), Kruskal–Wallis test was used to test statistical significance. The specific $P$ values at the defined CpG methylation frequency levels are the following: At 10–20%, the $P$ value = 0.04; at 70–80%, the $P$ value = 0.04; at 90–100%, the $P$ value = 0.04. In (B), motif enrichment was calculated using HOMER, which uses ZOOPS scoring (zero or one occurrence per sequence) coupled with the hypergeometric enrichment calculations. Exact $P$ values are included in Dataset EV5.

suggest that changes in DNA methylation may directly impact gene expression during COPD development. To assess whether epigenetic changes are associated with gene expression changes in AT2 cells in COPD, we performed low-input RNA-seq analysis on the RNA isolated from the same FACS-purified AT2 cell pellets as those used for T-WGBS (Fig. 1A; Dataset EV2). Known AT2-specific genes, including ABCA3, LAMP3, and the surfactant genes (SFTPA2, SFTPB and SFTPC) were among the top highly expressed genes, and they were not significantly changed in COPD AT2s (Fig. EV2A; Dataset EV6), confirming the AT2-characteristic transcriptional signature of our isolated cells.

Unsupervised principal component analysis (PCA) on the top 500 variable genes revealed a clear influence of the COPD phenotype in separating no COPD and COPD II–IV samples, as previously observed with DNA methylation analysis, irrespective of the cancer background of COPD samples (Figs. 3A and EV2B). COPD I samples showed a mixed pattern on PC1 projection and were distributed between no COPD and COPD II–IV, with one of the COPD I patients (donor 34) clustering together with no COPD (Fig. 3A), mirroring DNA methylation data. However, on PC4 projection (Fig. EV2B), COPD I samples separated from the other groups, suggesting a mild-specific expression sub-pattern.

Differential gene expression analysis identified 2261 upregulated and 916 downregulated genes in AT2 cells from the COPD II–IV group compared to no COPD (Fig. 3B, |log2 fold change| >0.5, at FDR of 10%, Dataset EV6), providing the transcriptional signature of COPD. Interestingly, AT2 cells from COPD I already showed transcriptional changes, with 332 upregulated and 44 down-regulated genes, providing unique insight into early disease (Fig. 3C). In total, 297 differentially expressed genes (DEGs) were shared between COPD I and COPD II–IV samples (Fig. 3D). The most upregulated genes in AT2 cells in COPD were the metalloprotease ADAMTS10 and the Na + -dependent OH− transporter SLC4A11 (Fig. EV2D). ADAMTS10 is involved in microfibril biogenesis (Kutz et al, 2011) and may regulate extracellular growth factor signaling mediated by transforming growth factor beta (TGFβ) and bone morphogenetic proteins (BMP), cytokines essential for proper organ development and tissue architecture (Kutz et al, 2011; Hubmacher and Apte, 2015).

SLC4A11 upregulation, required for NRF2-mediated antioxidant gene expression (Guha et al, 2017), may be a mechanism to dampen excessive oxidative stress in AT2 cells, a common feature of COPD. PDLIM3 and the potassium channel KCNJ5 were the top two downregulated genes (Fig. EV2D). PDZ-LIM proteins can act as signal modulators, influence dynamics and cell migration, regulate cell architecture, and control gene transcription (Krcmery et al, 2010). In the mouse lung, PDLIM5 deficiency has been associated with Smad3 downregulation and emphysema development (Krcmery et al, 2010; Warburton et al, 2013). Our data suggest that PDLIM3 reduction in human AT2 cells from COPD I and COPD II–IV patients may play a role in the development of emphysema.

## COPD-relevant pathways are transcriptionally altered in AT2 cells

To further resolve gene expression signatures of COPD states, we performed self-organizing map (SOM) clustering (Wehrens and Kruisselbrink, 2018) using the combined DEGs from COPD I and COPD II–IV, identifying three clusters (Fig. 3D; Dataset EV6). The largest cluster contained 2219 genes upregulated in both COPD II–IV and in the COPD I donor with the lowest FEV$_1$ value (donor 4, FEV$_1$ = 92.1%, Fig. 3D). Pathway and gene ontology enrichment analysis of this cluster identified processes associated with structural changes in the lung, such as extracellular matrix organization, activation of matrix metalloproteinases, and collagen formation (Fig. 3E; Dataset EV7). Antimicrobial peptide signature and interferon signaling were also enriched, suggesting a deregulation of immune and inflammatory signaling in AT2 cells in COPD. Closer examination of the antimicrobial peptide genes revealed that most antimicrobial peptide genes were significantly upregulated in COPD in AT2 cells (Fig. EV2C; Dataset EV6). Within the IFN signaling pathway, we observed upregulation of several genes encoding HLA class-I and -II molecules (HLA-F, HLA-DQB2), GTPases (MX1 and 2), antiviral enzymes (OAS1, 2, 3), immunoproteasome subunits (PSMB8, 9, 10) and transcription factors (IRF5, 7, 9) (Fig. EV2C; Dataset EV6). Many of these genes were already upregulated in COPD I samples, suggesting an early

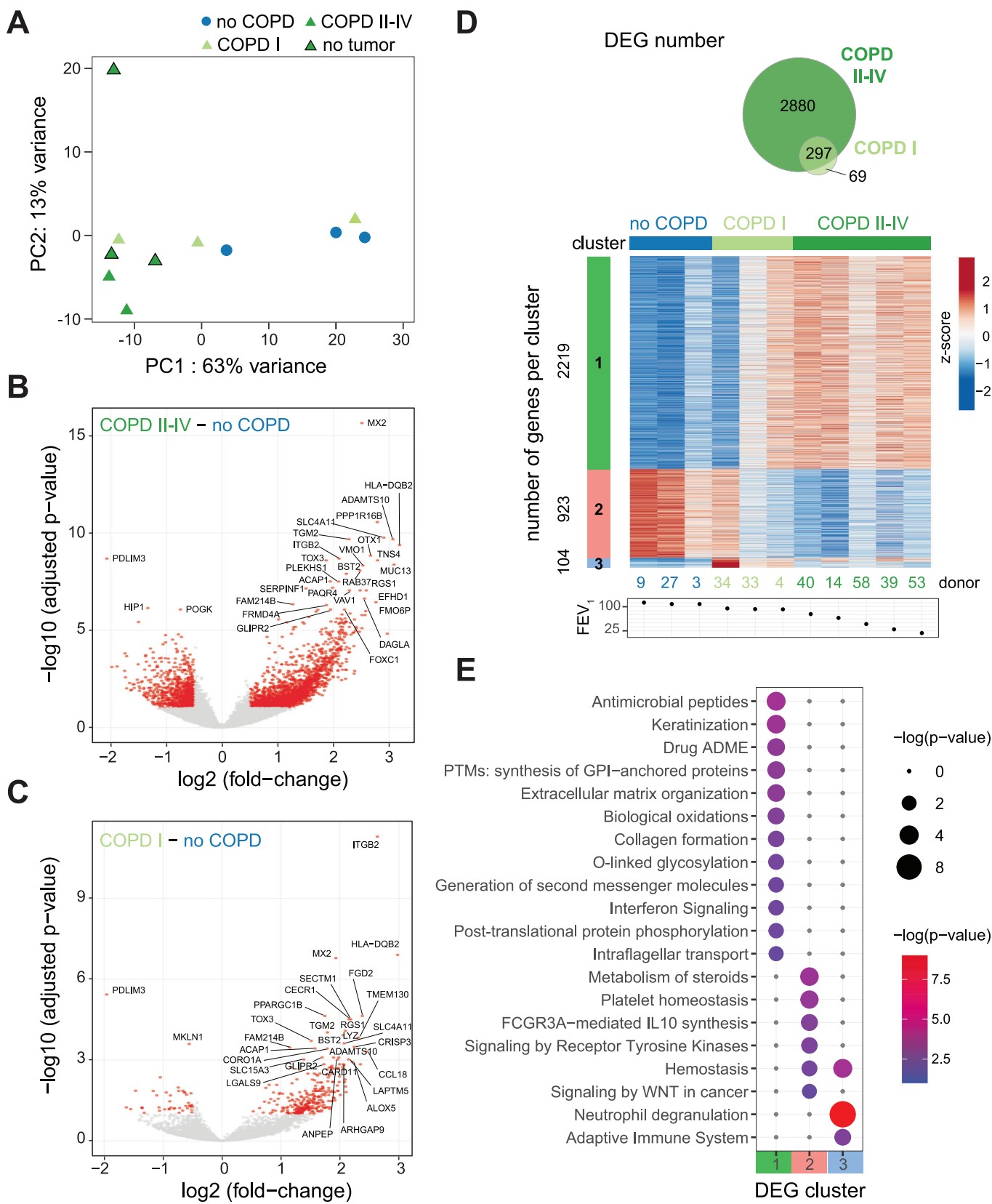

◀ **Figure 3. Global transcriptional changes occur in AT2 cells in COPD.**

(**A**) Unsupervised principal component analysis (PCA) of the 500 most variable genes in RNA-seq from sorted human AT2 cells. COPD I and COPD II–IV samples are represented in light and dark green triangles, respectively, and no COPD samples are represented as blue circles. COPD samples without a cancer background are displayed with a black contour. The percentage indicates the proportion of variance explained by each component. (**B, C**) Volcano plots of differentially expressed genes (DEG) (red dots; FDR of 10% and | log2(fold change) | > 0.5) in COPD II–IV ($n = 5$) (**B**) or COPD I ($n = 3$) (**C**) compared to no COPD ($n = 3$). (**D**) Top panel, Venn diagram displaying the overlap of DEGs in COPD I and COPD II–IV. Bottom, SOM clustering of the DEG displayed in (**B, C**) using three clusters. Donors were arranged according to decreasing FEV$_1$ value (indicated below the heatmap). (**E**) Metascape enrichment analysis of DEGs from each cluster displayed in (**D**). Enrichment $P$ values are indicated by node size and color. Data information: In (**B, C**), $P$ values were calculated using DESeq2, which uses a negative binomial GLM (generalized linear model) and Wald statistics. The Benjamini–Hochberg method was applied to correct for multiple testing, revealing adjusted $P$ values. The exact $P$ values for DEGs are included in Dataset EV6 and for metascape analysis in Dataset EV7.

activation of the IFN signaling pathway in AT2 cells during COPD pathogenesis. In the second SOM cluster, we identified 923 genes that were downregulated in COPD II–IV and in 2 COPD I donors (Fig. 3D). DEGs from this cluster are involved in the regulation of cholesterol biosynthesis, tyrosine kinase signaling, WNT signaling, as well as platelet biogenesis and hemostasis (Figs. 3E and EV2C). The smallest cluster contained 104 genes upregulated in COPD with a dysregulation already in COPD I donors, and included genes involved in interleukin signaling, neutrophil degranulation and adaptive immune system, emphasizing the dysregulation of inflammatory pathways in AT2 cells as an early event in COPD (Fig. 3E). To further characterize the DEGs, we performed gene set enrichment analysis (GSEA) and identified overrepresentation of genes related to chronic obstructive pulmonary disease (Fig. EV2E) and epithelial cell differentiation (Fig. EV2F).

Next, we examined pathways involved in AT2 stemness and lung regeneration. We observed a gradual downregulation of axin2, FZD5, LRP2, LRP6, as well as TCF7L1, known components of Wnt signaling (Dataset EV6; Fig. EV2C), consistent with the dysregulation of Wnt signaling in COPD (Baarsma and Konigshoff, 2017). In parallel, MCC, a negative regulator of the canonical Wnt/β-catenin signaling, WNT5B, and the cyclin-dependent kinase inhibitors CDKN2A, C, and D were upregulated (Fig. EV2C), suggesting a potential loss of repair capacity in AT2 cells in COPD. This was further confirmed when we compared the transcriptional signature of our COPD AT2 with that of alveolar epithelial progenitors (AEPs) identified by Zacharias et al (Zacharias et al, 2018). We observed a significant negative correlation of gene expression between the DEGs identified in these two datasets ($P < 2.2e^{-16}$, linear regression analysis), indicating a decrease in progenitor markers in COPD AT2 cells (Fig. EV2G; Dataset EV8). In addition, we observed an upregulation of several keratins (KRT5, KRT14, KRT16, KRT17) and mucins (MUC12, MUC13, MUC16, MUC20), suggesting a potential dysregulation of alveolar epithelial cell differentiation programs in COPD (Dataset EV6; Fig. EV2F). Immunofluorescence staining confirmed the presence of KRT5-positive cells in the distal lung in COPD and identified positive cells for both KRT5 and HT2-280 already in COPD I samples (Fig. EV2H). Collectively, these results indicate a dysregulation of stemness and identity in the alveolar epithelial cells in COPD.

## Integrated analysis reveals epigenetically regulated pathways in COPD

DNA methylation is a key mechanism of gene regulation (Jurkowska 2024). The similarity of the methylation and gene expression profiles in the PCAs suggested that epigenetic and transcriptomic changes in human AT2 cells during COPD might be interrelated (Figs. 2B and 3A). To gain a deeper understanding of the molecular pathways affected by DNA methylation changes in promoter proximity, we assigned identified DMRs to genes in proximity (maximum distance to respective TSS ± 6 kb) (Fig. EV3A–C). Overall, 755 DEGs, 23.8% of the total, had at least one associated DMR, indicating that they might be regulated by promoter-proximal methylation (Fig. EV3A; Dataset EV9). We observed a significant overrepresentation of DMRs associated with DEGs compared to non-DEGs (Fig. EV3B, Fisher's exact test, $P$ value = $2e^{-12}$). Notably, we observed preferentially a negative relationship between DNA methylation and gene expression, with hypermethylated DMRs mainly associated with downregulated genes and hypomethylated DMRs correlated with upregulated genes (Fig. EV3C). The negative correlation was independent of the location of the DMR relative to the TSS, with hypomethylated DMRs preferentially located downstream of the TSS (Fig. EV3C). Next, we performed Spearman correlation analysis of the promoter-proximal DMRs and the corresponding gene expression changes in COPD across all samples. While non-DEGs showed the expected normal distribution, indicating no dependency between promoter-proximal methylation and gene expression (Fig. 4A, blue line), DEGs displayed a bimodal curve enriched at high absolute correlation coefficients (Fig. 4A, orange line). Among the identified DEGs, 76.5% ($n = 492$) displayed a negative correlation (16.8% of the total DEGs), consistent with a repressive role of promoter DNA methylation. Interestingly, 23.5% of the identified DEGs ($n = 151$) showed a positive correlation between gene expression and DNA methylation. We extracted all genes with a Spearman correlation value of >0.5 or < −0.5 and plotted the methylation differences and expression changes between all COPD and no COPD samples (Fig. 4B). We observed a clear association between methylation differences and expression changes, with prominent examples including IL33, TMPRSS4, IRF9, and OAS2 (Dataset EV9). Enrichment analysis of the negatively correlated genes identified IFN signaling as the top deregulated pathway controlled by promoter-proximal methylation (Fig. 4C; Dataset EV10), indicating that aberrant IFN signaling in COPD may be epigenetically regulated. Additionally, motif analysis of DMRs that were highly correlated (|Spearman correlation coefficient|>0.5) with DEGs revealed a prominent enrichment of the cognate motif for ETS family transcription factors, such as ELF5, SPIB, ELF1, and ELF2 at hypomethylated DMRs (Fig. EV3D). Interestingly, SPIB has been shown to facilitate the recruitment of IRF7 and activate interferon signaling (Miyazaki et al, 2020). Consistently, our WGBS data uncovered SPIB motifs at hypomethylated DMRs, which aligns with its binding preference to unmethylated DNA (methyl minus, Fig. EV3D).

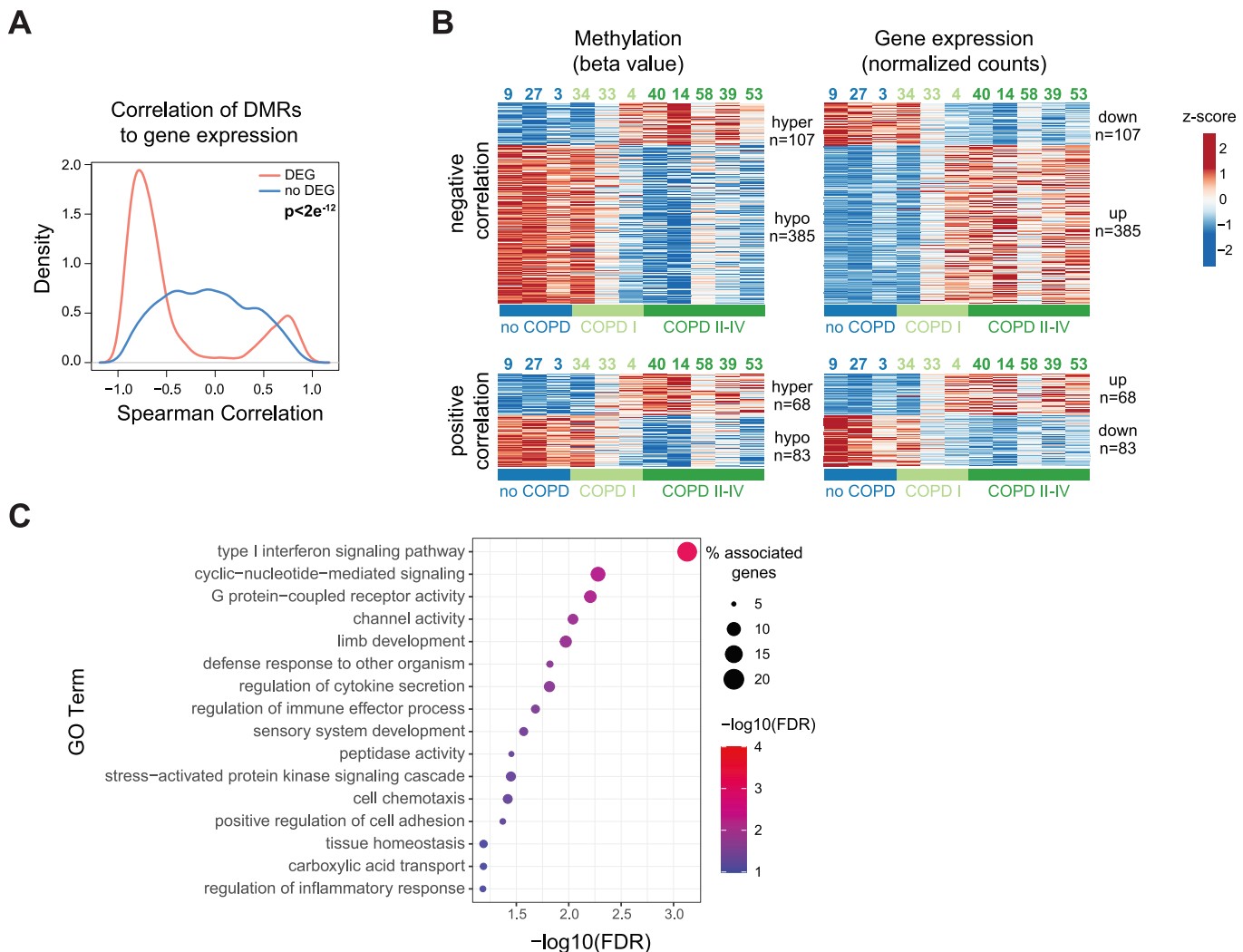

**Figure 4. Integrated analysis reveals epigenetic regulation of key AT2 pathways in COPD.**

(A) Spearman correlation between gene expression and DMR methylation. DMRs within +/− 6 kb from the transcription start site (TSS) were considered. Gene-DMR pairs were split into DEGs (red) and not significantly changed genes (no DEG, blue). (B) DMR methylation (left, beta-value) and gene expression (right, normalized expression counts) of 492 candidate genes with negative correlation (top, Spearman correlation < −0.5) and 151 genes with positive correlation (bottom, Spearman correlation >0.5) between DNA methylation and gene expression. Values are represented as z-scores. Donor AT2 identifiers are indicated above. (C) GO-Term overrepresentation analysis of negatively correlated DEGs. The adjusted *P* value is indicated by the color code, and the number of associated DEGs is indicated by the node size. Data information: In (C), *P* values were calculated using ClueGO, which uses a hypergeometric test. Exact *P* values are included in Dataset EV10.

To harness the full resolution of our whole-genome DNA methylation data, we extended the analysis beyond promoter-proximal regions and assessed how epigenetic changes in distal regulatory regions (enhancers) may relate to transcriptional differences in COPD. Because assigning enhancer elements to their target genes is challenging, we applied two complementary approaches. First, we used the GeneHancer database (Fishilevich et al, 2017) to link DMRs to regulatory genomic elements (GeneHancer element). Of the 25,028 DMRs, 18,289 DMRs (73%) coincided with at least one GeneHancer element. Of those 2144 DMR-GeneHancer associations were linked either to protein-coding or lncRNA genes. Next, we filtered high-scoring gene GeneHancer associations ("Elite"), leaving 1485 DMR-GeneHancer Elite associations. Of those, we selected the GeneHancer elements, which are linked to genes differentially expressed in our RNA-seq

analysis, resulting in 376 DMR-GeneHancer associations (Dataset EV9). Similar to the promoter-proximal analysis, we assessed the correlation of expression and methylation changes of the DMR-GeneHancer associations, demonstrating a high proportion of negatively and positively correlated events (Fig. EV3E). Finally, we performed gene enrichment analysis for the positively and negatively correlated genes. We detected significant GO term enrichments for negatively correlating genes only (Fig. EV3F; Dataset EV10), with the most pronounced term "regulation of tumor necrosis factor". In an alternative approach, we linked proximal and distal (within 100 kb from TSS) DMRs to the next gene using GREAT (McLean et al, 2010) (Fig EV1C; Dataset EV4), and calculated Spearman correlation between DMRs and associated DEGs. 147 DMRs were associated with high correlation rates with 93 genes from the Wnt/β-catenin pathway (Fig. EV3G), suggesting

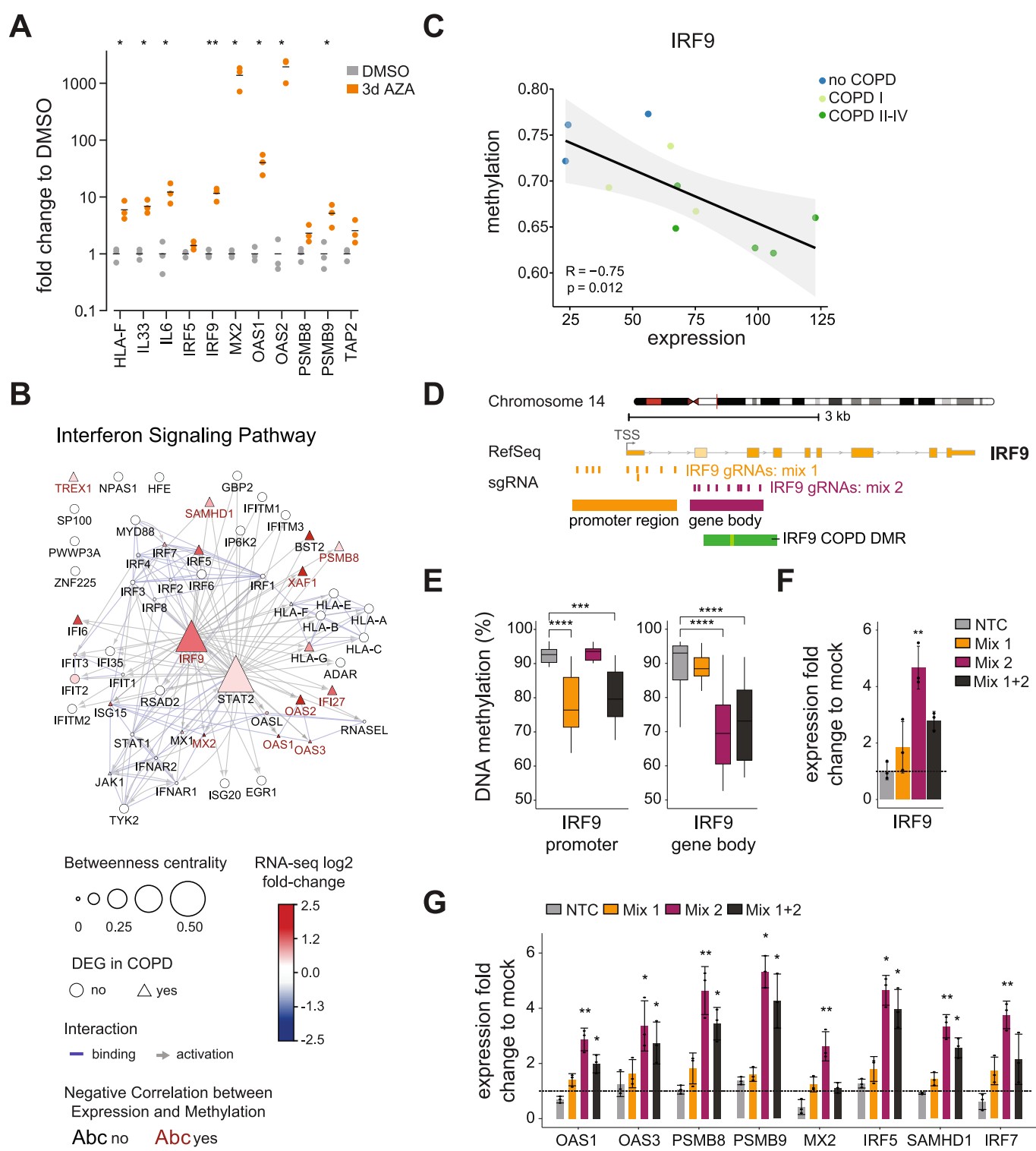

that DNA methylation may also drive the expression of genes of the Wnt/β-catenin family.

## Epigenetic control of IFN and Wnt/β-catenin pathways

IFN and Wnt signaling are two central pathways associated with AT2 stemness, differentiation, and repair (Fujino et al, 2012;

Baarsma and Konigshoff, 2017; Skronska-Wasek et al, 2017; Nabhan et al, 2018; Conlon et al, 2020; Major et al, 2020). To evaluate whether changes in DNA methylation may regulate the expression of selected key genes of the IFN- and Wnt/β-catenin signaling pathways identified from our AT2 study in COPD, we performed a DNA demethylation assay in the alveolar lung cell line A549. A549 cells were treated with increasing doses of the

◀ **Figure 5. Epigenetic regulation of the IFN pathway in AT2 cells.**

(A) Relative changes in expression ($2^{-\Delta\Delta C_T}$) of the indicated IFN pathway genes in A549 cells upon 5-Aza-2'-deoxycytidine (AZA, orange) treatment. Gene expression was measured by RT-qPCR using DMSO treatment as a control (gray) and RPLP0 as a housekeeping control. Each point represents the average of two technical replicates, and bars represent the median of three independent experiments ($n = 3$ biological replicates). Paired $t$ test, FDR corrected using the two-stage set-up method of Benjamini, Krieger, and Yekutieli. *$P$ value < 0.05; **$P$ value < 0.01. (B) Cytoscape analysis of the components of the IFN pathway, identifying IRF9 as a master regulator in sorted primary AT2 cells. Nodes are connected based on their protein-protein interaction annotation in the categories binding (blue line) or activation (gray arrow) in the STRING database. The node size represents the betweenness centrality within the network. DEGs are shown as triangles, and log2(fold change) in COPD II–IV is indicated by the node color. Red labeled nodes exhibit a negative Spearman correlation (< −0.5) with promoter proximal associated DMRs. (C) Scatter plot showing correlation between gene expression and DNA methylation of the promoter-proximal DMR for IRF9 in sorted human AT2 cells. Each dot represents an individual donor. Dots are color-coded according to the disease state. Gene expression is illustrated as normalized counts. Methylation is illustrated as the average beta value of the corresponding DMR. Correlation coefficient and $P$ value were calculated by the Spearman correlation method. (D) Detailed view of the IRF9 locus, featuring a core and extended DMR (light and dark green rectangle) identified between no COPD and COPD II–IV. Orange boxes represent the location of individual gRNAs used for targeting the promoter (mix 1), and dark purple boxes indicate the location of individual gRNAs targeting the gene body region (mix 2). (E-G) Targeted DNA demethylation of IRF9 using CRISPR-based epigenetic editing in A549 cells. Boxplots displaying percentage DNA methylation at CpG sites across the IRF9 promoter (left) or gene body regions (right) after transfection with the dCas9-VPR-mTet3 demethylating construct and IRF9 targeting gRNA mixes (or non-targeted control; NTC, which contained dCas9-VPR-mTet3 but no gRNA) (E). For each sample, the average methylation was calculated per CpG from three independent biological replicates ($n = 3$), and it was aggregated into bins containing CpGs from either the promoter or the gene body target regions. Boxes represent the median and interquartile range (IQR) of the data, and whiskers represent the full range of non-outlier values. Statistical significance of data was analyzed using the Kruskal–Wallis multiple comparison test, followed by Dunn's post hoc analysis comparing each sample, and adjusting $P$ values using the Benjamini–Hochberg correction method (*$P$ value < 0.05; **$P$ value < 0.01; ***$P$ value < 0.001; ****$P$ value < 0.0001). (F, G) Relative changes in expression ($2^{-\Delta\Delta C_T}$) of IRF9 (F) and a panel of its downstream targets (G). Displayed are the mean fold-changes in gene expression induced by transfection with the dCas9-VPR-mTet3 demethylating construct and IRF9 targeting gRNA mixes (or non-targeted control; NTC), normalized to mock transfection control. Biological replicate data ($n = 3$) is represented by individual data points, and standard deviation by the error bars. Statistical significance of data was analyzed by a Kruskal–Wallis test, followed by Dunn's post hoc analysis (*$P$ value < 0.05; **$P$ value < 0.01). Data information: Exact $P$ values for Fig. 5 can be found in Table EV2.

demethylating agent 5-Aza-2'-deoxycytidine (5-AZA). Using the Mass Array, we observed efficient demethylation of the long interspersed nuclear elements (LINEs), indicating successful DNA demethylation upon treatment (Fig. EV4A). Notably, even cells exposed to low doses of AZA (0.5 µM) showed a clear upregulation of genes from the IFN and Wnt/β-catenin signaling pathways (Figs. 5A and EV4B), confirming that their expression may be regulated by DNA methylation.

However, 5-AZA is a global demethylating agent, and the observed effects may not be direct. To further validate the epigenetic regulation of the important AT2 pathways, we took advantage of the locus-specific epigenetic editing technology (Jurkowski et al, 2015). We focused on the IFN pathway because it was the most significantly enriched Gene Ontology (GO) term in our integrative analysis of TWGBS and RNA-seq data. Several IFN pathway members exhibited hypomethylated DMRs within promoter-proximal regions, accompanied by increased gene expression (Figs. 4C and EV2C; Dataset EV9). In addition, we confirmed the elevated expression of the IFN-related genes with associated DMRs identified in our study in AT2 cells and AT2 cell subclusters from a recently published COPD scRNA-seq cohort (Hu et al, 2024) (Fig. EV4E,F). Network analysis identified the transcription factor IRF9 as a key master regulator of the upregulated IFN pathway in COPD (Fig. 5B). Notably, IRF9 upregulation in COPD AT2 cells is associated with a concomitant loss of DNA methylation, suggesting that IRF9 itself may be directly regulated by DNA methylation (Fig. 5C). This observation was supported by the increased expression of IRF9 (and its downstream target genes) upon 5-AZA treatment of A549 (Fig. 5A). To investigate the epigenetic regulation of IRF9 expression, we used CRISPR-Cas9-based epigenetic editing to specifically demethylate the IRF9 locus in A549 in a targeted manner. For this, we employed an epigenetic activating domain consisting of the catalytically inactive Cas9 (dCas9) fused to the transcriptional activator VPR and an engineered Tet3 DNA demethylase domain (see Methods

for details of the constructs). We targeted the fusion construct to the IRF9 gene using guide RNAs (gRNAs) specific to the IRF9 promoter (gRNAs mix 1) or IRF9 gene body (gRNAs mix 2), which contained the demethylated region identified in our profiling data in COPD AT2 (IRF9 COPD DMR) (Figs. 5D and EV4G). Specific demethylation of the target regions in the IRF9 locus was validated by amplicon bisulfite sequencing (Figs. 5E and EV4G,H). We observed a significant upregulation of IRF9 expression upon specific targeting and DNA demethylation of the IRF9 gene (Fig. 5F). This effect was not observed in the mock transfected cells (mock), nor in cells transfected with the untargeted dCas9-VPR-TET3 (no gRNA, NCT), where there was also no methylation change (Figs. 5E,F and EV4H), confirming the specificity of the assay. Interestingly, demethylation of the IRF9 region located within the gene body and identified as a DMR in our methylation profiling in COPD AT2 (gRNAs mix 2), showed a stronger demethylation and induction of IRF9 expression than targeting the IRF9 gene promoter itself (gRNA mix 1, Fig. 5E,F), suggesting the presence of a cis-regulatory element in this region.

Notably, targeted DNA demethylation of IRF9 also resulted in a robust activation of its downstream target genes, including OAS1, OAS3, PSMB8, PSMB9, MX2, and IRF7, demonstrating that IRF9 demethylation is sufficient to activate the IFN-signaling pathway (Fig. 5G). These experiments confirm the epigenetic regulation of the IFN pathway and validate IRF9 as a master regulator of IFN signaling in alveolar epithelial cells. Taken together, our data suggest that early activation of the IFN signaling pathway in AT2 cells in COPD occurs as a consequence of epigenetic remodeling of this pathway.

## Discussion

Understanding the mechanisms that drive human lung regeneration is essential for defining novel therapeutic strategies aimed at

restoring respiratory barrier integrity and lung function impaired in COPD and other chronic lung diseases. AT2 cells, as alveolar stem cells, are critical for the maintenance and regeneration of the alveolar epithelium after injury (Nabhan et al, 2018; Zacharias et al, 2018). COPD and emphysema are associated with increased apoptosis, senescence, and decreased regenerative capacity of the alveolar niche (Tsuji et al, 2006; Kosmider et al, 2011; Rustam et al, 2023). However, the epigenetic regulation of AT2 cell dysfunction in COPD remains poorly understood. In this work, we demonstrate that genome-wide DNA methylation changes occur in human AT2 cells and may drive COPD pathology by dysregulating key pathways that control inflammation, viral immunity, and AT2 regeneration.

Previous studies using various clinical samples provided evidence of dysregulated DNA methylation patterns in smokers and COPD patients (Sood et al, 2010; Qiu et al, 2012; Vucic et al, 2014; Wan et al, 2015; Yoo et al, 2015; Busch et al, 2016; Morrow et al, 2016; Sundar and Rahman 2016; Song et al, 2017; Carmona et al, 2018). However, they relied on low-resolution, microarray-based approaches that covered only a fraction of the genome and analyzed complex samples with heterogeneous cellular composition, making it difficult to determine the contribution of individual cell types. Two studies investigated DNA methylation changes in isolated lung fibroblasts (Clifford et al, 2018; Schwartz et al, 2023), but the epigenetic dysregulation of AT2 cells across COPD stages remained uncharted.

Using high-resolution epigenetic profiling, we uncovered widespread alterations in the DNA methylation landscape of human AT2 cells in COPD, associated with pronounced changes in gene expression. Consistent with our earlier methylation data from COPD lung fibroblasts (Schwartz et al, 2023), epigenetic changes in AT2 were enriched at regulatory regions, including enhancers and promoters, indicating that they may drive aberrant transcriptional programs in COPD. This was further supported by the strong anticorrelation between DNA methylation and gene expression, with more than 500 dysregulated genes showing corresponding DNA methylation changes in promoter-proximal regions. Wnt and IFN signaling, two central pathways orchestrating the regenerative capacity of AT2 cells (Nabhan et al, 2018; Zacharias et al, 2018; Katsura et al, 2020) were enriched for genes with anticorrelated DNA methylation signatures in promoters and distal regions, respectively. Hence, our data suggest that epigenetic dysregulation of key pathways involved in AT2 proliferation, renewal, and differentiation may contribute to impaired alveolar-cell renewal in COPD.

Chronic inflammation has long been recognized as a pathogenic driver exacerbating COPD phenotypes, including airway remodeling and progressive alveolar destruction (Barnes 2019; Mehta et al, 2020; Booth et al, 2023). Notably, the inflammatory processes in COPD persist long after smoking cessation (Shapiro 2001), suggesting an epigenetic regulation, yet the molecular mechanisms driving inflammation and lung tissue destruction in COPD are not well understood. We observed an upregulation of multiple IFN genes in AT2 in COPD, consistent with a previous expression array study (Fujino et al, 2012), and we confirmed that our IFN signature genes were upregulated in AT2c and AT2rb subsets in COPD from a recent scRNA-seq study (Hu et al, 2024). IFNα/β signaling was also enriched in COPD patients in the inflammatory AT2 cluster (iAT2) in a recent scRNA-seq study (Watanabe et al, 2022). Notably, the upregulation of the IFN genes in our data was

correlated with a concomitant loss of DNA methylation at their promoters, indicating their epigenetic control in COPD. Epigenetic remodeling of IFN-signaling genes was already evident in early-stage COPD (COPD I), suggesting that it may contribute not only to disease exacerbation but also to its development.

Interferons are essential for antiviral host defense. They are induced upon viral recognition through the binding of interferon-responsive transcription factors (IRFs) and coordinate the expression of IFN-stimulated genes (ISGs) in the infected and neighboring cells (Barrat et al, 2019). Dysregulated IFN signaling in smokers and COPD patients is associated with impaired antiviral immunity, increased susceptibility to infections, and disease exacerbations (Hilzendeger et al, 2016; Hsu et al, 2016; Wu et al, 2016; Garcia-Valero et al, 2019; Mehta et al, 2020). In addition to its pro-inflammatory and antiviral roles, IFN also has anti-proliferative and apoptotic effects (Parker et al, 2016). In particular, sustained activation of IFN induces lung tissue remodeling and destruction. Targeted pulmonary overexpression of IFN-γ in mice causes inflammation and leads to emphysema development via induction of proteolytic enzymes (e.g., MMP-12) (Wang et al, 2000). IFN-γ is also a potent inducer of DNA damage and apoptosis in airway and AT2 cells (Zheng et al, 2005). More recently, a direct effect of IFN on alveolar epithelial repair was demonstrated, as prolonged IFN production reduced AT2 proliferation and differentiation during recovery from influenza infection (Major et al, 2020). In addition, generation of progenitor terminal airway-enriched secretory cells (TASCs) was suppressed by IFN-γ signaling, which is increased in the distal airways of COPD patients (Rustam et al, 2023), implicating IFN in impaired distal airway regeneration in COPD. Our data show that increased IFN-signaling in AT2 cells is regulated by the decrease of DNA methylation at IFN genes, providing a mechanistic explanation for the reduced repair capacity of alveolar epithelial cells in COPD. We also observed a strong downregulation of alveolar epithelial progenitor (AEP) markers (Zacharias et al, 2018) in AT2 cells from COPD patients, further supporting their impaired regenerative potential, consistent with a recent study (Hu et al, 2024).

Currently, we do not know whether the identified DNA methylation changes are the cause or the consequence of the disease process, and not much is known about the correlation of DNA methylation with disease severity. 13 genes with altered methylation patterns have been identified in the lung tissue of COPD GOLD I and II patients compared to non-smoking controls (Casas-Recasens et al, 2021). Our previous study revealed that genome-wide DNA methylation changes are present in lung fibroblasts from COPD I patients compared to controls with matched smoking history, demonstrating that epigenetic changes occur in mild COPD and may provide a sensitive biomarker for early disease detection (Schwartz et al, 2023). Interestingly, we observed heterogeneous methylation profiles within the mild COPD group in AT2 cells, despite very similar lung function data of the three COPD I donors, suggesting that epigenetic profiling may provide additional information for differentiating mild COPD patients and disease progression. However, our study is cross-sectional, our cohort included only 3 COPD I donors, and we did not have any follow-up data on the patients, so future large-scale profiling of mild disease (or even pre-COPD cohorts) in an extended patient cohort will be crucial for a better understanding of early disease and its progression trajectories.

Currently, it is unclear how cigarette smoking leads to changes in DNA methylation patterns in human AT2 and how epigenetic changes translate into biological phenotypes in COPD. DNA methylation in regulatory regions can modulate the binding of transcription factors to DNA (Stadler et al, 2011), thus methylation profiling allows the identification of transcriptional regulators potentially mediating the epigenetic changes. In line with this, in the identified DMRs (both in promoter proximal regions and in enhancers), we detected a significant enrichment of binding sites of transcription factors associated with lung development, apoptosis, senescence, inflammation, and differentiation, processes critical for tissue repair and regeneration that are altered in COPD. Binding sites for the lineage transcription factors NKX2 and TEAD were enriched in hypermethylated DMRs, suggesting that their changed binding could impact cell fate determination of human AT2 cells during lung regeneration (Little et al, 2021). Moreover, TP53 binding sites were overrepresented in hypomethylated DMRs in COPD, which could explain the increased apoptosis of AT2 cells reported in COPD lungs (Kosmider et al, 2011). Consistent with this hypothesis, recent studies demonstrated that p53 mediates apoptosis of cycling cells mobilized during tissue repair in lungs exposed to prolonged inflammation during COVID-19 and influenza infection (Katsura et al, 2020; Major et al, 2020). Taken together, our data suggest that NKX2 and TP53 may mediate some of the downstream biological effects in COPD AT2 and contribute to COPD phenotypes. Future work is needed to delineate and experimentally validate the target genes directly bound and regulated by these transcription factors in AT2.

Notably, we have identified the transcription factor IRF9 as a key epigenetically regulated master regulator of IFN activation. IFNs can induce extensive remodeling of the epigenome, including histone marks, through binding of the IRFs to gene regulatory elements and regulation of chromatin accessibility at promoters and enhancers (Park et al, 2017; Kamada et al, 2018). They recruit chromatin-remodeling enzymes, leading to the transcriptional activation of ISGs (Barrat et al, 2019). IFN-induced epigenetic changes can persist beyond the period of IFN stimulation, conferring transcriptional memory and sustained expression of ISGs (Kamada et al, 2018). This "epigenetic priming" may mediate the sensitivity of AT2 cells to subsequent environmental exposures, contributing to impaired antiviral responses and reduced alveolar regeneration in COPD. Through treatment with a demethylating drug and targeted epigenetic editing, we demonstrated the ability to modulate the expression of IFN-stimulated genes through the demethylation of IRF9. Therefore, effective modulation of IFN signaling to restore the robust induction of ISGs upon injury/exposure, followed by timely downregulation of IFN responses to allow efficient alveolar epithelial repair, may protect against disease exacerbations and enhance the regenerative capacity of AT2 cells in COPD patients.

Overall, our findings suggest that rewiring the epigenomic landscape in COPD AT2 cells to revert aberrant transcriptional programs may have the potential to restore the internal regenerative programs lost in the disease.

The strengths of our study include the use of purified human alveolar type 2 epithelial progenitor cells from a well-matched and carefully validated cohort of human samples, including mild and severe COPD patients, providing high relevance to human COPD. Importantly, we matched the smoking status and smoking history

of all donors, which is key in epigenetic studies, as cigarette smoking profoundly impacts the DNA methylation landscape of tissues (Hoang et al, 2024). With the first genome-wide high-resolution methylation profiles of isolated cells across COPD stages, we offer novel insights into the epigenetic regulation of gene expression in epithelial progenitor cells in COPD, expanding our understanding of how alterations in regulatory regions and specific genes could contribute to disease development. We identified IRF9 as a key IFN transcription factor regulated by DNA methylation. Notably, by targeting IRF9 through epigenetic modifications, we modulated the activity of the IFN pathway, which plays a crucial role in the immune response and lung tissue regeneration. Epigenetic editing techniques could offer a novel therapeutic strategy for COPD by downregulating IFN pathway activation and promoting the regeneration of epithelial progenitor cells in the lungs. Further preclinical and clinical studies are needed to validate the efficacy and safety of epigenetic editing approaches in COPD treatment (Jurkowska, 2024).

However, we acknowledge several limitations of our study that warrant further investigation. First, the sample size was small. The use of strict quality criteria for donor selection limited the available samples, particularly for the ex-smoker control group. This resulted in an unequal distribution of COPD and control samples. This impacts the power of statistical analysis, particularly in the WGBS analysis, where millions of regions genome-wide are tested. Nevertheless, the clear negative correlation between promoter-proximal methylation and corresponding gene expression highlights the robustness of the DMR selection. Additionally, we were able to experimentally validate interferon-associated DMRs using epigenetic editing, highlighting the power of integrated epigenetic profiling in identifying disease-relevant regulators.

Overall, we detected a higher number of correlated DMR-DEG associations using our simple promoter-proximal linkage compared to the GeneHancer approach. Assigning enhancers to their target genes with high confidence is a complex and challenging task. Enhancers are often located far from the genes they regulate and can interact with their target genes through three-dimensional chromatin loops. Furthermore, enhancers can operate in a highly context-dependent manner, with the same enhancer regulating different genes depending on the cell type, developmental stage, or environmental signals. Determining which enhancer is active under specific conditions remains a hurdle in the field, especially since the AT2-specific chromatin profiles of enhancer marks are not yet available.

In addition, while WGBS provides unprecedented resolution and high coverage of the DNA methylation sites across the genome, it does not allow distinguishing 5-methylcytosine from 5-hydroxymethylcytosine. Therefore, we cannot exclude that some methylated sites we detected are 5-hydroxymethylated. However, 5-hydroxymethylcytosine is present at very low levels in the lung tissue (Li and Liu, 2011).

Finally, despite careful removal of airways from distal lung tissue using a dissecting microscope, we cannot exclude the presence of some terminal/respiratory bronchiole cells in our FACS-isolated EpCAM$^{pos}$/PDPN$^{low}$ population. Recent scRNA-seq studies provided an unprecedented resolution and identified several epithelial subpopulations and transitional cells residing in the terminal/respiratory bronchioles and alveoli, including respiratory airway secretory cells (Basil et al, 2022), terminal airway-enriched

secretory cells (Rustam et al, 2023), terminal bronchiole-specific alveolar type-0 (AT0) (Kadur Lakshminarasimha Murthy et al, 2022), and emphysema-specific AT2 cells (Hu et al, 2024). These cells may contribute to alveolar repair in healthy and COPD lungs; however, with our bulk DNA methylation and RNA-seq study, we are unable to resolve all these subpopulations. Future development of single-cell methylation and non-reference-based algorithms for DNA methylation deconvolution will enable deeper epigenetic phenotyping of specific AT2 and bronchiolar cell subsets.

# Methods

### Reagents and tools table

| Reagent/resource | Reference or source | Identifier or catalog number |
| --- | --- | --- |
| **Experimental models** | | |
| Sorted primary human AT2 cells | This study | |
| A549 cells | ATCC | CCL-185 |
| **Recombinant DNA** | | |
| empty U6_gRNA vector | Addgene | Plasmid #41824 |
| dSPn-VPR-mTET3del1ΔC | Stepper, 2020 | |
| pEGFP puro | Addgene | Plasmid #45561 |
| pUC19 | Addgene | Plasmid # 50005 |
| phage lambda DNA unmethylated | Promega | D1521 |
| **Antibodies** | | |
| Human TruStain FcX | Biolegend | 422301 |
| Anti-human CD326 (EpCAM-PE) | Affymetrix eBioscience | 12-9326-42 |
| Anti-human podoplanin-AlexaFluor 647 | BioLegend | 337007 |
| Anti-human CD45-PerCP-Cy5.5 | Affymetrix eBioscience | 45-9459-42 |
| Anti-human CD45-APC Cy7 | BioLegend | 304014 |
| Anti-human CD45-Bv605 | BD | 564047 |
| Mouse monoclonal anti-human HT2-280 IgM | Terrace Biotech | HT2-280 |
| Rabbit anti-human pro-SFTPC IgG | Abcam | ab196677 |
| Rabbit anti-human KRT5 IgG | Abcam | ab52635 |
| Goat anti-mouse IgM-AF568 isotype control | Thermo Fisher Scientific | A-21043 |
| Rabbit IgG isotype control | Thermo Fisher Scientific | 02-6102 |
| **Oligonucleotides and other sequence-based reagents** | | |
| gRNAs mix 1 and 2 | This study | Table EV1_Primers |
| qPCR primers epi editing | This study | Table EV1_Primers |
| Bisulfite PCR primers IRF9 | This study | Table EV1_Primers |
| RPLP0_TaqMan | Thermo Fisher Scientific | Hs00420895_gH |
| HLA-F_TaqMan | Thermo Fisher Scientific | Hs01587837_g1 |
| IL33_TaqMan | Thermo Fisher Scientific | Hs04931857_m1 |

| Reagent/resource | Reference or source | Identifier or catalog number |
| --- | --- | --- |
| IL6_TaqMan | Thermo Fisher Scientific | Hs00174131_m1 |
| IRF5_TaqMan | Thermo Fisher Scientific | Hs00158114_m1 |
| IRF9_TaqMan | Thermo Fisher Scientific | Hs00196051_m1 |
| MX2_TaqMan | Thermo Fisher Scientific | Hs01550814_m1 |
| PSMB8_TaqMan | Thermo Fisher Scientific | Hs00544758_m1 |
| PSMB9_TaqMan | Thermo Fisher Scientific | Hs00160610_m1 |
| OAS1_TaqMan | Thermo Fisher Scientific | Hs00973635_m1 |
| OAS2_TaqMan | Thermo Fisher Scientific | Hs00942643_m1 |
| TAP2_TaqMan | Thermo Fisher Scientific | Hs00241060_m1 |
| KiCqStart® SYBR® Green primers | Merck | Table EV1_Primers |
| LINE-1_Bisulfite_Fw | This study | Table EV1_Primers |
| LINE-1_Bisulfite_ Rv | This study | Table EV1_Primers |
| **Chemicals, enzymes, and other reagents** | | |
| CO$_2$-independent medium | Thermo Fisher Scientific | 18045054 |
| BSA | Carl Roth | T844.3 |
| Penicillin/streptomycin | Fisher Scientific | 15140122 |
| Amphotericin B | Sigma | 15290026 |
| UltraPure™ 0.5 M EDTA, pH 8.0 | Fisher Scientific | 15575020 |
| HBSS without Ca2 +, Mg2 +, and phenol red | Fisher Scientific | 14175053 |
| High-glucose DMEM GlutaMAX™ | Thermo Fisher Scientific | 31966047 |
| Dimethylsulfoxide (DMSO) | Carl Roth | A994.1 |
| Fetal Bovine Serum | Gibco | 10270106 |
| Human tumor dissociation kit | Miltenyi Biotec | 130-095-929 |
| ROCK inhibitor, Y-27632 2HCl | Adooq Bioscience | 129830-38-2 |
| DNase I | ProSpec-Tany TechnoGene | enz-417 |
| ACK lysis buffer | Sigma-Aldrich | A10492-01 |
| SyTOX blue | Thermo Fisher Scientific | S34857 |
| BD FACSDiva CS&T research | BD Bioscience | 655050 |
| BD FACSDiva accudrop beads | BD Bioscience | 345249 |
| Xylene | Sigma-Aldrich | 534056-4 L |
| Ethanol | Sigma-Aldrich | 32205-2.5L-M |
| Citric acid | Abcam | ab93678 |
| Tween 20 | Thermo Fisher Scientific | 3005 |
| Triton-X100 | Carl Roth | P270 |
| PBS 1X | Fisher Scientific | 18912014 |
| AlexaFluor™ 488 Tyramide SuperBoost™ Kit, goat anti-rabbit IgG | Thermo Fisher Scientific | B40943 |
| ProLong Antifade reagent with DAPI | Thermo Fisher Scientific | P36931 |

| Reagent/resource | Reference or source | Identifier or catalog number |
|---|---|---|
| QIAamp Micro Kit | Qiagen | 56304 |
| RNase A | Qiagen | 19101 |
| RNase-Free DNase Set | QIAGEN GmbH | 79254 |
| AMPure beads | Beckman Coulter | A63881 |
| EZ DNA Methylation kit | Zymo | D5001, D5002 |
| Tn5 transposase | Epicentre | 0020015725 |
| Arcturus Picopure RNA Isolation kit | Thermo Fisher Scientific | KIT0204 |
| RNase-Free DNase Set | Qiagen | 79254 |
| Ambion Nuclease-free water | Thermo Fisher Scientific | AM9939 |
| Agilent RNA 6000 Pico Kit | Agilent | 5067-1513 |
| RNA ladder | Agilent | 5067-1535 |
| Qubit RNA HS assay kit | Thermo Fisher Scientific | Q32852 |
| Qubit dsDNA HS assay kit | Thermo Fisher Scientific | Q32851 |
| RNA Pico 6000 Assay Kit | Agilent Technologies | 50671513 |
| Ham's F12 medium | PAN Biotech | P04-14550 |
| Glutamax | Gibco | 35050061 |
| 5-Aza-2'-deoxycytidine | SIGMA | A3656 |
| AllPrep DNA/RNA Micro Kit | Qiagen | 80284 |
| Revertaid 1st cDNA synthesis kit | Thermo Fisher Scientific | K1622 |
| Fast SYBR® Green | Thermo Fisher Scientific | A46109 |
| Gibco Puromycin Dihydrochloride | Thermo Fisher Scientific | A1113803 |
| Sodium L-ascorbate | Sigma-Aldrich | A7631-25G |
| Cryostor | Sigma-Aldrich | C2874 |
| **Software** | | |
| YACTA-software | Lim et al, 2016 | |
| GraphPad Prism 10.4.1 | http://www.graphpad.com | |
| Zen software | http://www.zeiss.com | |
| FlowJo software (Tree Star) | https://www.flowjo.com/ | |
| MethylCtools | Hovestadt et al, 2014 | |
| Trimmomatic v0.36 | Bolger et al, 2014 | |
| BWA MEM | Li and Durbin, 2009 | |
| Picard MarkDuplicates | http://picard.sourceforge.net/ | |
| Bsseq / Bsmooth | R/Bioconductor (Hansen et al, 2012) | |
| methylkit | R/Bioconductor (Akalin et al, 2012) | |
| HOMER | Heinz et al, 2010 | |
| Gviz | R/Bioconductor (Hahne and Ivanek, 2016) | |
| GREAT | McLean et al, 2010 | |
| Spliced Transcripts Alignment to a Reference (STAR) aligner | Dobin et al, 2013 | |

| Reagent/resource | Reference or source | Identifier or catalog number |
|---|---|---|
| bamUtil package | https://github.com/statgen/bamUtil | |
| Subread package | Liao et al, 2014 | |
| Qualimap's RNA-seq QC module | Okonechnikov et al, 2016 | |
| DESeq2 package | Love et al, 2014 | |
| Metascape | Zhou et al, 2019; https://metascape.org/ | |
| clusterProfiler | Bioconductor (Wu et al, 2021) | |
| Cytoscape | https://cytoscape.org/ | |
| ClueGO v2.5.6 | Bindea et al, 2009 | |
| CluePedia | Bindea et al, 2013 | |
| Seurat | Satija et al, 2015 | |
| E-CRISPR | Heigwer et al, 2014 | |
| CRISPOR | Concordet and Haeussler 2018 | |
| Trim Galore v.0.6.10 | https://www.bioinformatics.babraham.ac.uk/projects/trim_galore/ | |
| Bismark v.0.24.9 | https://www.bioinformatics.babraham.ac.uk/projects/bismark/ | |
| SeqMonk v.1.48.1 | https://www.bioinformatics.babraham.ac.uk/projects/seqmonk/ | |
| GeneHancer database v5.14 | Fishilevich et al, 2017 | |
| EpyTyper | Agena Bioscience | |
| **Other** | | |
| Superfrost Plus slides | Thermo Fisher Scientific | P10144 |
| Hydrophobic pen | Abcam | ab2601 |
| Zeiss LSM780 confocal fluorescent microscope | Zeiss, ZMBH imaging facility | NA |
| HiSeq2500 | Illumina, NGX Bio service | NA |
| Bioanalyzer 2100 system, model G2939A | Agilent Technologies | NA |
| NextSeq 500 High output | Illumina, Genecore EMBL | NA |
| EpiTyper MassARRAY | Agena Bioscience | NA |
| QuantStudio™ 5 Real-Time PCR System | Applied Biosystems | NA |
| Gentle MACS dissociator | Miltenyi Biotec | 130-093-235 |
| MACSmix™ tube rotator | Miltenyi Biotec | 130-090-753 |
| FACS Aria IIu sorter 605 85 mm – 3-laser, 9-color (5-3-2) | BD Bioscience | NA |
| Mr. Frosty | Thermo Fisher Scientific | 5100-0001 |
| Cryo-tubes 2.0 mL | Sarstedt | 72.38 |
| Gentle MACS™ C tubes | Miltenyi Biotec | 130-093-237 |
| 100 µm Falcon cell strainer | Neolab Migge | 352360 |
| 70 µm Falcon cell strainer | Neolab Migge | 352350 |

| Reagent/resource | Reference or source | Identifier or catalog number |
|---|---|---|
| 40 μm Falcon cell strainer | Neolab Migge | 352340 |
| Falcon® 5 mL round-bottom polystyrene tube, with cell strainer snap cap | Neolab Migge | RN003621 |
| 384-well hardshell plate clear, 20 pcs pk | Thermo Fisher Scientific | 4483285 |
| MicroAmp® Optical Adhesive Film, 100 covers | Fisher Scientific | 10299204 |

## Study approval

The protocol for tissue collection was approved by the ethics committees of the University of Heidelberg (S-270/2001, biobank vote), as well as the Cardiff School of Biosciences (ref. 23 08-02) and the NHS REC (ref. 25/PR/1279) (study votes). The experiments followed the principles set out in the WMA Declaration of Helsinki and the Department of Health and Human Services Belmont Report. All patients provided written informed consent and remained anonymous in the context of this study.

## Patient samples

Strict patient inclusion criteria were established at the beginning of the tissue collection to ensure the best possible matching of control and disease groups in terms of age, BMI, gender, smoking status, and smoking history, as well as tumor type (for cancer resection samples). To avoid the acute inflammation effects due to smoking, all collected donors have ceased smoking for at least 12 months. None of the donors received chemotherapy or radiation within 4 years before surgery. In addition, lung function results, based on the forced expiratory volume in 1 s ($FEV_1$) and $FEV_1$/forced vital capacity ratio ($FEV_1$/FVC ratio), quantitative emphysema score index (ESI) based on chest CT, as well as medical history, were collected for each patient for the best possible characterization of the included samples. All tissue samples underwent strict evaluation by an experienced lung pathologist, who confirmed the absence of tumors in all samples and the lack of extensive fibrosis and emphysema in the control group (Fig. 1C). Besides, the COPD-relevant phenotypes, such as emphysema, airway thickening, and immune infiltration, were evaluated. In total, 3 ex-smoker controls (no COPD), 3 mild COPD ex-smoker donors (GOLD I, COPD I), and 5 moderate-to-severe COPD ex-smoker donors (GOLD II–IV, COPD II–IV) were profiled (Fig. 1A–C; Dataset EV1). Importantly, three of the profiled samples from donors with severe COPD (HLD39, 53, and HLD58) came from lung resections, thus representing tissue without a cancer background, an important control that ensured that the observed changes relate to COPD and are present in cancer-free material.

## Emphysema index determination

Lung and emphysema segmentation were performed by calculating the emphysema score index from clinically indicated preoperative CT scans taken with mixed technical parameters. After automated lung segmentation using the YACTA software, a threshold of −950 HU was used with a noise-correction range between −910 and −950 HU to calculate the relative amount of emphysema in % of the respective lung portion (Lim et al, 2016). While usually global ESI was measured, only the contralateral non-affected lung side was used if one lung was severely affected by the tumor.

## FFPE and H&E

Representative slices from different areas of the tissue samples were taken and fixed O/N with 10% neutral buffered formalin. Next, fixed slices were washed and kept in 70% ethanol. Sample dehydration, paraffin embedding, and hematoxylin and eosin (H&E) staining were performed at Morphisto (Morphisto GmbH, Frankfurt, Germany). Two 4-μm-thick sections were cut per sample on a Leica RM2255 microtome with integrated cooling station and water basin and transferred to adhesive glass slides (Superfrost Plus, Thermo Fisher Scientific). Subsequently, sections were dried to remove excess water and enhance adhesion. Anonymized H&E-stained slides were evaluated by an experienced lung pathologist at the Thorax Clinic in Heidelberg.

## Cryopreservation of lung parenchyma

Distal lung tissue pieces were transported in CO2-independent medium (Thermo Fisher Scientific) supplemented with 1% BSA (Carl Roth), 1% penicillin/streptomycin (Fisher Scientific), and 1% amphotericin B (Sigma) and processed as previously published (Llamazares-Prada et al, 2021; Pohl et al, 2023). Briefly, tissue pieces were inflated with cold HBSS (Fisher Scientific) supplemented with 1% BSA (Carl Roth), 2 mM EDTA (Fisher Scientific), 1% Amphoterin B, and 1% penicillin/streptomycin (Fisher Scientific) (later referred as HBSS$^{++++}$), and exemplary samples of the lung piece were collected for histological analysis (see above). Pleura, airways, and vessels were carefully removed, and the airway and vessel-free parenchyma was further minced into 4 ×4 mm pieces and cryopreserved in high-glucose DMEM GlutaMAX™ (Thermo Fisher Scientific) containing 10% DMSO (Carl Roth) and 20% FBS (GE Healthcare). The tubes were flipped to distribute the medium within the tissue pieces, and kept on ice for fifteen minutes. The tubes were placed into Mr. Frosty containers (Thermo Fisher Scientific) and transferred to −80 °C to ensure a cooling rate of 1 °C/min. For long-term storage, tubes were kept in liquid nitrogen.

## Tissue dissociation, viability check, and FACS sorting

To minimize potential technical bias, samples from no COPD and COPD donors were processed in parallel in groups of three (one no COPD and 2 COPD samples). Cryotubes containing airway and vessel-depleted distal lung parenchyma were thawed, washed, and minced into smaller pieces before mechanical and enzymatic dissociation as previously reported (Llamazares-Prada et al, 2021). Tissue pieces were dissociated into single-cell suspensions with the human tumor dissociation kit following the manufacturer's instructions (Miltenyi Biotec). Briefly, 0.5–1 g of minced tissue was added to a MACS C tube containing 4.5 mL of in $CO_2$-independent medium (Thermo Fisher Scientific) supplemented with 1% BSA (Carl Roth), 1% penicillin/streptomycin (Fisher Scientific) and 1% amphotericin B (Sigma), and the enzyme mix from the human tissue dissociation kit (Miltenyi Biotec), consisting of 200 μL enzyme H, 100 μL enzyme R, 50 μL enzyme A,

10 µM ROCK inhibitor (Y-27632, Adoq Bioscience), and 100 µL DNase I (ProSpec-Tany TechnoGene, enz-417, final concentration 1 µg/mL). Tubes were tightly closed and introduced into the MACS dissociator (Miltenyi Biotec) for mechanic disruption, and the following program was selected: program h_tumor_01, followed by a 15-min incubation at 37 °C on a rotator; h_tumor_01, plus 15 min at 37 °C on a rotator; h_tumor_02, and 15 min at 37 °C on a rotator for a final enzymatic dissociation and a last mechanical shearing using the program h_tumor_02. The samples were pipetted up and down to help with disaggregating. Finally, the enzymatic reaction was stopped by adding 20% FBS (Gibco, Thermo Fisher Scientific). Single cells were washed and collected by sequential filtering through 100-, 70-, and 40-µm cell strainers (BD Falcon). Cells were spun down, incubated for 4 min in ACK lysis buffer (Sigma-Aldrich) at room temperature for red blood cell lysis. After two washes with HBSS$^{++++}$ to neutralize cell lysis buffer, Fc receptors were blocked with human TruStain FcX (Biolegend, 5 µL/10$^6$ cells in 0.1 mL HBSS + +++) for 30 min on ice. Cells were counted before staining. Immune and epithelial cells were labeled using EpCAM, PDPN, and CD45 antibodies as indicated below (Reagents Tools Table). Anti-human CD326 (EpCAM-PE, 12-9326-42, Affymetrix eBioscience, 5 µL/5 × 10$^5$ cells in 0.1 mL HBSS + +++), Anti-human podoplanin (PDPN)-AlexaFluor 647 (337007, BioLegend, 8 µL/5 × 10$^5$ cells in 0.1 mL HBSS + +++), Anti-human CD45-PerCP-Cy5.5 (45-9459-42, Affymetrix eBioscience, 5 µL/5 × 10$^5$ cells in 0.1 mL HBSS + +++), CD45-APC Cy7 (BioLegend, 304014, 5 µL/5 × 10$^5$ cells in 0.1 mL HBSS + +++), or CD45-Bv605 (564047, BD, 5 µL/5 × 10$^5$ cells in 0.1 mL HBSS + +++) were incubated for 30 min in the dark at 4 °C following manufacturer instructions. Stained samples were washed, resuspended in HBSS$^{++++}$, and added to FACS tubes with 40 µm cell strainer caps. To discriminate between live and dead cells, we used 1 µL SyTOX blue per 1 mL of cell suspension as recommended by the manufacturer (Thermo Fisher Scientific). We sorted live, single cells, gated EpCAM$^{high}$ PDPN$^{low}$ cells as previously published (Fujino et al, 2011; Fujino et al, 2012; Chu et al, 2020) using a FACS Aria IIu cell sorter. Sorted epithelial cells were manually counted, aliquoted, spun down, and flash-frozen in liquid nitrogen. Cell pellets were kept at –80 °C until the full cohort was collected for subsequent RNA-seq and T-WGBS to avoid batch effects. RNA and genomic DNA for RNA-seq and T-WGBS were isolated from identical aliquots of sorted cell pellets. The remaining cells were fixed for immunofluorescence studies. FlowJo software (Tree Star) was used to analyze the FACS results.

## Immunofluorescence

FFPE lung tissue samples were cut into 4-µm-thick sections and added to Superfrost Plus slides (Thermo Fisher Scientific), deparaffinized and rehydrated by immersing in Xylene and gradually decreasing solutions of ethanol as published (Llamazares-Prada et al, 2021). Antigen retrieval was conducted at 99 °C for 20 min in a pressure cooker containing citrate buffer (10 mM citric acid, 0.05% Tween 20, pH 6.0) and allowed to cool down for 90 min at room temperature. Tissue samples were surrounded with a hydrophobic pen (Abcam) and permeabilized with 1% Triton-X100 (Carl Roth) in PBS 1× for 20 min at RT without shaking. Slides were washed with PBS 1× and endogenous peroxidase was quenched with 3% H$_2$O$_2$ for 1 h at RT following the manufacturer's

instructions (Tyramide SuperBoost kit (TSB), Thermo Fisher Scientific). Slides were washed with PBS 1X, blocked with 10% goat serum (TSB kit, Thermo Fisher Scientific) for 1 h at RT before overnight incubation with HT2-280 (IgM mouse monoclonal, 1:30 dilution, Terrace Biotech), pro-SFTPC (IgG rabbit polyclonal, 1:200 dilution, Abcam ab196677) and/or KRT5 (IgG rabbit monoclonal, 1:200 dilution, A ab52635) primary antibodies at 4 °C in incubation buffer (TSB kit, Thermo Fisher Scientific). Mouse IgM and rabbit IgG were used as controls. Tissues were washed thoroughly with PBS 1× before incubating 1 h at RT with secondary antibody solution from TSB kit containing secondary goat-anti-rabbit IgG coupled to HRP in 10% normal goat serum. Secondary goat-anti-mouse-IgM-AF568 (1:500 dilution, Thermo Fisher Scientific) was added to the mix. Slides were washed with PBS 1× before signal amplification was performed following the manufacturer's instructions. Briefly, slides were incubated for 3 min with tyramide working solution (inactive tyramide connected to AlexaFluor 488), and the reaction was blocked with working stop solution for 1 min at RT. Finally, tissue slides were mounted with ProLong Antifade reagent containing DAPI (Thermo Fisher Scientific). Microscope slides were left to dry overnight before imaging. Imaging of cells was conducted at the ZMBH imaging facility (Heidelberg) using the Zeiss LSM780 confocal fluorescent microscope.

## DNA extraction and T-WGBS

Genomic DNA was extracted from 1 to 2 × 10$^4$ sorted alveolar epithelial cells isolated from cryopreserved lung parenchyma from 11 different donors in parallel using QIAamp Micro Kit (Qiagen, Hilden, Germany) following the manufacturer's protocol, with an additional RNase A treatment step (Qiagen, Hilden, Germany). T-WGBS was performed as described previously (Wang et al, 2013; Schwartz et al, 2023) using 30 ng of genomic DNA as input. In total, 15 pg of unmethylated DNA phage lambda was spiked in as a control for bisulfite conversion. Tagmentation was performed in TAPS buffer using an in-house purified Tn5 assembled with load adapter oligos (Wang et al, 2013) at 55 °C for 8 min. Tagmentation was followed by purification using AMPure beads, oligo replacement, and gap repair as described (Wang et al, 2013). Bisulfite treatment was performed using EZ DNA Methylation kit (Zymo) following the manufacturer's protocol. Four sequencing libraries were generated per sample using 11 amplification cycles. Equimolar amounts of all four libraries were pooled and sequenced on two lanes of a HiSeq2500 (Illumina, San Diego, California, US) machine using NGX Bio service (San Francisco), with 100 bp, paired-end reads. The T-WGBS library preparations were performed for all donors in parallel and sequenced in a single batch to minimize batch effects and technical variability.

## T-WGBS read alignment and methylation quantification

Read alignment and methylation quantification were performed as described (Schwartz et al, 2023). Briefly, the MethylCtools pipeline was modified for T-WGBS data and used for whole-genome bisulfite sequencing mapping (Hovestadt et al, 2014). Adapter sequences were trimmed using Trimmomatic (Bolger et al, 2014). For alignment, cytosines in the reference and read sequences are converted to thymines before alignment. The reads were aligned to the transformed strands of the hg19 reference genome using BWA MEM, then reverted to their original states. Duplicate reads were

marked using Picard MarkDuplicates (http://picard.sourceforge.net/). For methylation calling, the cytosine frequency was used to determine methylated CpGs, whereas cytosine to thymine conversion indicated unmethylated CpGs. Only bases with a Phred-scaled quality score of ≥ 20 were considered, excluding the 10 bp at the ends of the reads and CpGs on sex chromosomes.

## DMR calling

DMR calling between severe COPD and no COPD was performed as described (Schwartz et al, 2023). Shortly, R/Bioconductor package bsseq was used to detect differentially methylated regions (Hansen et al, 2012). The data were first smoothed using the Bsmooth function, and only CpG sites with at least 4x coverage were used for further analysis. A t-statistic was calculated between the two groups using the Bsmooth.tstat function. Differentially methylated regions (DMRs) were identified by selecting the regions with the 5% most extreme t-statistics, filtering for regions with at least 10% methylation difference and containing at least 3 CpGs with a maximum distance of 300 bp between them. An additional non-parametric Wilcoxon test was applied to remove potentially false-positive regions, since the t-statistic is not well-suited for not normally distributed values, as expected at very low/high (close to 0%/100%) methylation levels. A significance level of 0.1 was used without further FDR correction. In addition, DMRs which were <5 kb apart from each other were stitched together if they had the same direction of methylation change (hyper/hypo) and the resulting average methylation level within the DMR did not drop below 10%.

## DMR downstream analysis

The R/Bioconductor package methylkit was used to generate the CpG methylation frequency distribution and PCA plot, excluding CpGs with less than four fragment coverage and the CpGs with 0.1% highest coverage (Akalin et al, 2012). Genome feature annotation and known transcription factor motif enrichment analyses were performed using the HOMER functions annotatePeaks.pl and findMotifsGenome.pl (Heinz et al, 2010). To obtain information about methylation-dependent binding for transcription factor motifs enriched at DMRs, the results of a recent SELEX study (Yin et al, 2017b) were integrated into the analysis. They categorized transcription factors based on the binding affinity of their corresponding DNA motif to methylated or unmethylated motifs. Those whose affinity was impaired by methylation were categorized as MethylMinus, while those whose affinity increased were categorized as MethylPlus. A motif database of 1787 binding motifs with associated methylation dependency was constructed. The log odds detection threshold was calculated for the HOMER motif search as follows. Bases with a probability > 0.7 got a score of log(base probability/0.25); otherwise, the score was set to 0. The final threshold was calculated as the sum of the scores of all bases in the motif. Motif enrichment analysis was carried out against a sampled background of 50,000 random regions with matching GC content using the *findMotifsGenome.pl* script of the HOMER software suite, omitting CG correction and setting the generated SELEX motifs as the motif database.

Genome tracks were plotted using the R/Bioconductor package Gviz (Hahne and Ivanek, 2016). Roadmap chromatin states were obtained for lung tissue (E096) (Roadmap Epigenomics et al, 2015). DMRs within 100 kb distance were assigned to the next gene and

subjected to gene ontology enrichment analysis using GREAT (McLean et al, 2010). To define significant associations with pathways, we used the default settings of the GREAT tool, which are as follows: FDR < 0.05 in both binomial and hypergeometric tests and a minimum region-based fold enrichment of 2.

## RNA isolation and RNA-seq

RNA was isolated from flash-frozen pellets of $1–2 \times 10^4$ sorted AT2 cells from 11 different donors in parallel using the Arcturus Picopure RNA Isolation kit (Thermo Fisher Scientific, KIT0204) following the manufacturer's instructions. DNA was removed by on-column DNase treatment (Qiagen) before elution with nuclease-free water (Thermo Fisher Scientific). RNA concentration and integrity were measured using the RNA Pico 6000 Assay Kit of the Bioanalyzer 2100 system (Agilent Technologies, Santa Clara, CA), and only samples with RIN > 8 were included in RNA-seq. Low-input, stranded mRNA (poly-A enriched) libraries were manually prepared at the Genomics core facility (GeneCore) at EMBL, Heidelberg (Germany). The obtained libraries were pooled in equimolar amounts. 1.8 pM solution of each library was pooled and loaded on the Illumina sequencer NextSeq 500 High output and sequenced unidirectionally, generating ~450 reads per run, each 75 bases long. The RNA-seq library preparation for all donors was performed in parallel, and all samples were sequenced in a single batch to minimize batch effects and technical variability.

## Alignment and transcript abundance quantification

The raw sequence data were processed using Trimmomatic v0.36, a flexible read trimming tool for Illumina sequence data (Bolger et al, 2014). The parameters for Trimmomatic were set to remove adapters (ILLUMINACLIP), trim low-quality bases from the start (LEADING:3), the end (TRAILING:3), and perform a sliding window trimming (SLIDINGWINDOW:4:15), with a minimum length of the reads set to 36 bases (MINLEN:36). The trimmed reads were then aligned to the reference genome hg19 and transcriptome (Ensemble release 87) using the Spliced Transcripts Alignment to a Reference (STAR) aligner (Dobin et al, 2013). The parameters were set to filter out alignments with high mismatch rates and multi-mapping reads. Quality reports were generated using Qualimap's RNA-seq QC module (Okonechnikov et al, 2016). Duplicate reads were marked using the bam dedup command from bamUtil package (https://github.com/statgen/bamUtil). The counting was performed using the featureCounts function from the Subread package (Liao et al, 2014). The parameters were set to count reads in reverse-stranded libraries (-s 2), ignore duplicate reads (--ignoreDup).

## Differential gene expression analysis

The DESeq2 package (Love et al, 2014) was used to read the count table and to create a DESeqDataSet object from the count data and metadata. The count data was subset to include only autosomes and only lincRNA and protein-coding genes. An exploratory analysis was performed to visualize the distribution of raw or regularised log-transformed counts, and to run PCA. The DESeq2 package was also used to perform differential gene expression analysis between no COPD and COPD II–IV, as well as no COPD and COPD I

groups. The results were filtered to include only significant hits (adjusted $P$ value < 0.1 and absolute log2 fold change >0.5). Gene set overrepresentation analysis was carried out using the Metascape online tool (Zhou et al, 2019). For this analysis, the background was defined as all expressed genes. Significantly differentially expressed genes from both no COPD/COPD II–IV and no COPD/COPD I groups were combined and categorized based on self-organizing map clustering. The following settings were applied: $P$ value cutoff of 0.01, and minimum enrichment of 1.5. Enrichment was performed using the Reactome Gene Set. This analysis helped identify the key functional categories that were overrepresented in the respective cluster. Selected overrepresented terms from Metascape analysis were visualized using the Bioconductor package clusterProfiler (Wu et al, 2021).

## Integrated analysis

Cytoscape was used to analyze negatively or positively correlated DMR-DEG pairs. ClueGO (v2.5.6) analysis was conducted using all DEG associated with a promoter proximal DMR ( + /− 6 kb from TSS) and the Spearman correlation coefficient < −0.5 or >0.5 (Bindea et al, 2009). The following settings were used: statistical test used = Enrichment (Right-sided hypergeometric test), correction method used = Benjamini–Hochberg, Min GO Level = 4, Max GO Level = 10, Kappa Score Threshold = 0.4. Next, genes associated with the top-enriched term "type I interferon signaling pathway" were extracted, and a gene interaction network was built using the Cytoscape plugin CluePedia (Bindea et al, 2013).

For enhancer analysis, the GeneHancer database version 5.14, which annotates 392,372 regulatory genomic elements (GeneHancer element) on the hg19 reference genome, was used (Fishilevich et al, 2017). Of the 25,028 DMRs 18,289 DMRs coincided with at least one GeneHancer element, resulting in 19,661 DMR-GeneHancer associations. Next, the GeneHancer elements were filtered for association with protein-coding or long-non-coding RNA genes and high-scoring gene GeneHancer associations ("Elite"), leaving 1485 DMR-GeneHancer associations. Of those, the GeneHancer elements were selected, which are linked to differentially expressed genes in COPD, resulting in a final table of 376 DMR-GeneHancer associations. Similar to the promoter-proximal analysis, the Spearman correlation of expression and methylation changes of the DMR-GeneHancer associations was assessed. GO gene enrichment analysis for positively and negatively correlating genes was done using Metascape (Zhou et al, 2019).

## A549 cell culture and 5-Aza-2′-deoxycytidine demethylation assays

The human AT2-like cell line A549 (CCL-185, ATCC) was grown in Ham's F12 medium (PAN Biotech, P04-14550) supplemented with 10% fetal bovine serum (FBS, Gibco, 10270106), 1% Glutamax (Gibco, 35050061) and 1% penicillin-streptomycin (Fisher Scientific, 15140122) at 37 °C in 5% $CO_2$ atmosphere, as recommended to preserve the AT2-like phenotype (Cooper et al, 2016). Cells were purchased from the ATCC and routinely tested negative for Mycoplasma throughout the experiments. For demethylation assays, cells were seeded at $10^3$ cells per $cm^2$ in 21 $cm^2$ cell-culture-treated dishes (Falcon). Forty-eight hours later, cells received 5-Aza-2′-deoxycytidine (0.5 μM AZA, SIGMA, A3656) or DMSO, and the medium was replaced 48 h after. 72 h after treatment

initiation. Cells were left to recover for 48 h in complete Ham's F12 medium without AZA. Each experiment was performed in three independent biological replicates. Total RNA and DNA were isolated using AllPrep DNA/RNA Micro Kit (QIAGEN, 80284).

## Validation of LINE demethylation with Mass Array

DNA and RNA were isolated using the AllPrep DNA/RNA micro kit (QIAGEN) according to the manufacturer's instructions. 1 μg of genomic DNA was bisulfite-converted using the EZ DNA Methylation Kit (Zymo Research). Demethylation of LINE elements was validated using matrix-assisted time-of-flight mass spectrometry (MassARRAY; Agena Bioscience), a sequencing-independent method. The MassAR-RAY assay was performed as described previously (Ehrich et al, 2005). Specific primers targeting LINE-1 were used on bisulfite-treated genomic DNA from A549 cells treated with AZA (either 0.5, 1, or 3 μM) or DMSO control (Fw: TTTATATTTTGGTATGATTTTG-TAG; Rv: TTTATCACCACCAAACCTACCCT). The EpyTyper Software (Agena Bioscience) and a DNA methylation standard with defined ratios of in vitro methylated whole genome amplified DNA were included (0, 20, 40, 60, 80, and 100%) to quantify the level of methylation of the 3 CpG sites.

## Gene expression analysis using quantitative PCR

A549 cells were harvested, and RNA and DNA were isolated using the AllPrep DNA/RNA micro kit (QIAGEN) according to the manufacturer's instructions. In total, 1 microgram of total RNA was reverse-transcribed using Revertaid 1st cDNA synthesis kit (Thermo Fisher Scientific) according to the supplier's protocol. To quantify the expression of IFN pathway genes, real-time PCR was performed with 10 ng of cDNA and gene-specific TaqMan assays as suggested in the manual (RPLP0: Hs00420895_gH; HLA-F: Hs01587837_g1; IL33: Hs04931857_m1; IL6: Hs00174131_m1; IRF5: Hs00158114_m1; IRF9: Hs00196051_m1; MX2: Hs01550814_m1; PSMB8: Hs00544758_m1; PSMB9: Hs00160610_m1; OAS1: Hs00973635_m1; OAS2: Hs00942643_m1; TAP2: Hs00241060_m1). To quantify the expression of other genes, real-time PCR was performed using 10 ng of cDNA, Fast SYBR® Green (ThermoFisher), and specific KiCqStart® SYBR® Green primers (Merck). MicroAmp™ Optical 384-Well Reaction with 10 μL reactions were loaded into a QuantStudio™ 5 Real-Time PCR System (Applied Biosystems) and run according to the following recommended program 10 min 55 °C, 1 min 95 °C, followed by 40 cycles of 10 s 95 °C, 1 min 60 °C. For each biological replicate, all reactions were run in duplicates, and the average $C_T$ values between duplicates were used for analysis. The fold change in gene expression upon AZA treatment was calculated using the ($2^{-\Delta\Delta C_T}$) compared to control DMSO-treated cells after normalization to RPLP0 expression. The list of primers used in this study is provided in the Reagents and Tools Table and Table EV1.

## Validation of IFN gene upregulation in a published scRNA-seq dataset

scRNA-seq data from (Hu et al, 2024), generously provided by M Köningshoff, were processed using the default Seurat workflow (Satija et al, 2015). Expression of IFN-related genes was extracted and plotted as log-normalized gene expression levels in AT2 cells from control and COPD donors. Seurat's AddModuleScore() function was used to compute a gene set score for a custom IFN

program using the genes listed in Fig. EV4E and to analyze the IFN gene set scores in AT2 cell subclusters identified in (Hu et al, 2024). Briefly, average gene expression scores were computed for the gene set of interest, and the expression of control features (randomly selected) was subtracted as described in (Tirosh et al, 2016).

## Epigenetic editing of IRF9

A549 cells were maintained as described above. 19 different sgRNAs targeting promoter (P), or gene body (GB) regions of the human *IRF9* gene were designed using E-CRISP (Heigwer et al, 2014) and CRISPOR (Concordet and Haeussler, 2018), and purchased from Integrated DNA Technologies (IDT) as separate oligo strands with overhangs complementary to the BbsI cleavage site. The two strands were annealed at equimolar ratio by heating to 95 °C and allowed to cool and ligated into the BbsI-HF-digested empty U6_gRNA vector (a gift from George Church, Addgene plasmid #41824 (Mali et al, 2013)), modified to contain two BbsI sites at the gRNA sequence position (Stepper, 2020). For targeted DNA demethylation, the plasmid dSPn-VPR-mTET3del1ΔC, consisting of an engineered catalytic domain of mouse TET3 fused to tripartite VP64-p65-Rta (VPR) transcriptional activator, was used (Stepper, 2020).

For epigenetic editing, A549 cells were seeded into a 6-well plate, 600,000 cells per well, to achieve 70% confluency the following day. The cells were transfected using polyethyleneimine (PEI, MW 40,000; 1 mg/ml, Polysciences) using 2 μg of plasmid DNA and 6 μl of PEI. Several different setups were used. Transfection control contained 5% pEGFP puro (a gift from Michael McVoy, Addgene plasmid #45561 (Abbate et al, 2001)), and either (i) 20% pUC19 and 75% empty modified U6_gRNA vector for the mock transfection control; (ii) 20% dSPn-VPR-mTET3del1ΔC and 75% empty modified U6_gRNA vector for the non-targeted control; or (iii) 20% dSPn-VPR-mTET3del1ΔC and 75% pooled IRF9 gRNAs (either mix 1, gRNAs 1–10, targeting the promoter (P) region of *IRF9*, mix 2, gRNAs 11–19, targeting the gene body (GB) region of *IRF9* or a mix containing all 19 gRNAs targeting both P and GB regions of *IRF9*). 24 h later, the medium was replaced with 2 mL of prewarmed full growth medium containing 2 μg/mL of puromycin and 25 μM of sodium L-ascorbate (Sigma-Aldrich). Puromycin selection was performed for 2 days by replacing the medium every 24 h. Upon reaching confluency, cells were transferred to T25 and maintained until Day 10 post-transfection in the presence of 25 μM sodium L-ascorbate. On Day 10, cells were pelleted, flash-frozen in liquid nitrogen, and stored at −80 °C until RNA and DNA isolation.

## Bisulfite amplicon sequencing

Genomic DNA was extracted from 3 independent replicates following the Bio-On-Magnetic-Beads (BOMB) DNA extraction protocol using silica-coated BOMB beads (Oberacker et al, 2019). Extracted DNA was converted using the EZ DNA Methylation Kit (Zymo Research). Bisulfite PCR primers targeting the IRF9 P and GB regions were designed against the human genome assembly 38 (hg38) using the Zymo Research Bisulfite Primer Seeker online design tool (https://zymoresearch.eu/pages/bisulfite-primer-seeker). The list of PCR primers is provided in Table EV1. HotStarTaq plus PCR reagents (Qiagen) were used for PCR amplification. Bisulfite PCR amplicons were purified using the

carboxyl-coated BOMB bead Clean Up protocol (Oberacker et al, 2019). Amplicons were phosphorylated with T4 PNK (NEB) and ligated with TruSeq adapters using T4 DNA ligase (NEB) (Glenn et al, 2019). Dual-indexing was carried out with the TruSeq panel of indexing primers in a short-cycle PCR using Luna Universal Probe One-Step RT-qPCR reagents (NEB). The pooled adapter libraries were purified and size-selected using AMPure XP beads (Beckman Coulter) or carboxyl-coated BOMB beads (Oberacker et al, 2019), and quantified with Qubit Fluorometer using dsDNA-HS reagents (Invitrogen). Samples were sequenced by the Cardiff University School of Biosciences Genomics Research Hub on an Illumina MiSeq using Nano flow cells with V2 reagents in paired-end mode with 500 cycles to yield 450 bps forward, and 50 bps reverse reads. The sequencing library was loaded at a concentration of 8 pM with 25% PhiX DNA added to maintain read complexity.

## Sequence analysis

Analysis of sequencing data was performed as previously described (Stepper et al, 2017). Poor quality reads, and adapters were filtered and trimmed with Trim Galore (v.0.6.10; https://www.bioinformatics.babraham.ac.uk/projects/trim_galore/) using default settings. Bismark (v.0.24.0; https://www.bioinformatics.babraham.ac.uk/projects/bismark/) was used to align reads to the human genome (hg38) with default settings and to extract methylation data that were then processed using SeqMonk (v. 1.48.1; https://www.bioinformatics.babraham.ac.uk/projects/seqmonk/) to quantify percentage methylation at each CpG site. This data was further visualized using the UCSC Genome Browser's custom tracks function. At least 60-fold coverage was achieved for all but one sample.

## Quantification of gene expression after epigenetic editing

The RNA was isolated from three independent replicates following the Bio-On-Magnetic-Beads (BOMB) protocol (Oberacker et al, 2019). RevertAid First Strand cDNA Synthesis Kit (Thermo Fisher Scientific) was used for cDNA synthesis following the manufacturer's instructions. The quantitative PCR reaction mix contained 5 μl of the Luna Universal qPCR Master Mix (2×, New England BioLabs), 1 μl of the forward primer (10 μM), 1 μl of the reverse primer (10 μM), 0.5 μl of 10× SYBR dye, and 0.5 μl of nuclease-free water. 10 ng of cDNA was used per reaction, and technical triplicates were run for each primer pair. The primers used for qPCR can be seen below. The samples were run in QuantStudio™ 5 Real-Time PCR System using the following program: 50 °C for 2 mins, 95 °C for 10 min, 40 cycles at 95 °C for 15 s and 60 °C for 1 min, followed by 95 °C for 15 s, 60 °C for 1 min and 95 °C for 0.1 s. For each run, the melting curves were collected at the end of the experiment to assess the amplification specificity. The qPCR data were normalized to the housekeeping gene REEP5. To calculate the expression change of a target gene, $2^{-\Delta\Delta C_T}$ values were normalized to the mock transfection control (pUC19). The list of qPCR primers is provided in Table EV1.

## Statistical analysis

Statistical analysis was performed with GraphPad Prism software, version 8.0.1 or with R. The significance level was set at 0.05 unless

## The paper explained

### Problem

Chronic Obstructive Pulmonary Disease (COPD) is the fourth leading cause of death worldwide, affecting over 200 million people. This progressive lung disease causes breathing difficulties and has no cure; current treatments only manage symptoms. A major unresolved question is why the lungs of COPD patients cannot efficiently repair themselves. The lung alveoli contain specialized epithelial stem cells, called alveolar type 2 (AT2) cells, that normally regenerate damaged alveolar tissue, but in COPD, these cells lose their repair capacity. While we know that smoking and inflammation damage the lungs and the AT2 cells, we do not understand what goes wrong at the molecular level. Understanding the mechanisms of lung repair is essential for developing curative therapies that restore lung function rather than merely treating symptoms.

### Results

This study generated, for the first time, high-resolution molecular maps of purified AT2 stem cells from COPD patients at different disease stages and controls. The researchers discovered widespread changes in DNA methylation, a chemical modification that controls which genes are turned on or off without changing the DNA sequence itself. These epigenetic changes were concentrated in regulatory regions that control gene activity and closely matched the abnormal gene expression patterns observed in COPD cells. Most strikingly, the team found that the interferon signaling pathway, normally used by cells to fight viral infections, was excessively activated in COPD patients due to the loss of DNA methylation at a master regulator gene called IRF9. To validate this connection, the researchers used precise molecular tools to artificially remove DNA methylation from the IRF9 gene in lung cells, which successfully triggered a similar excessive interferon response as seen in COPD patients. This demonstrates that epigenetic changes drive disease-relevant cellular behaviors.

### Impact

These findings reveal that COPD is partly an epigenetic disease, one where chemical modifications of the DNA, rather than DNA mutations, impair the lung's natural regeneration programs. The discovery that IRF9 acts as a master switch for excessive interferon signaling provides a specific molecular target for new therapies. Because epigenetic changes are potentially reversible, unlike DNA mutations, this opens promising new therapeutic avenues. In the future, targeted epigenetic editing technologies could be used to restore normal DNA methylation patterns in COPD cells, potentially reactivating the lung's natural repair mechanisms. For COPD patients, who currently face a progressive disease with no curative options, understanding and correcting these epigenetic defects could offer a transformative strategy aimed at restoring lung regeneration rather than merely slowing functional decline.

otherwise indicated. The number of replicates and the statistical test used are described in the figure legends for each of the panels and in Table EV2 (for Fig. 5). As this was the first genome-wide methylation discovery study in purified AT2 in COPD, no sample size estimation was performed.

## Data availability

The datasets produced in this study are available in the following databases: [T-WGBS and RNA-seq data]: European Genome-phenome Archive (EGA, https://ega-archive.org), hosted by the EBI and the CRG. Access to patient data is controlled by a data access committee (DAC). RNA-seq: EGAS00001007387. T-WGBS:

EGAS00001007386. Source data files have been deposited in the Cardiff University Research Data Repository (https://doi.org/10.17035/cardiff.31136152).

The source data of this paper are collected in the following database record: biostudies:S-SCDT-10_1038-S44321-026-00386-9.

## Peer review information

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

## Acknowledgements

This study was supported by Boehringer Ingelheim. The work was partly funded by the School of Biosciences (Cardiff University) (SP), the Academy of Medical Sciences Springboard Award (SBF007\100176, DP/RZJ), UKRI Future Leaders Fellowship (MR/X032914/1, RZJ/SP), and the German Center for Lung Research (DZL) (82DZL004C4 and 82DZLT84C4 [CP, MLP], 82DZL004C2 and 82DZLT84C2 [TM, MAS, FJFH, CPH, HW]). We would like to thank Lung Biobank (Heidelberg, Germany)—a member of the Biomaterial Bank Heidelberg (BMBH), the tissue bank of the National Center for Tumor Diseases (NCT), and the Biobank platform of the German Center for Lung Research (DZL) for providing Biomaterials and Data. We also thank Christa Stolp for her help with collecting primary material. We acknowledge excellent sequencing service and helpful discussions from the Genomics core facility (GeneCore, EMBL, Germany) for RNA-seq, from NGX Bio (San Francisco, USA) for T-WGBS sequencing, and from Cardiff University School of Biosciences Genomics Research Hub for bisulfite amplicon sequencing, as well as support from the ZMBH imaging facility (Heidelberg, Germany) for immunofluorescence. We thank Morphisto GmbH (Frankfurt, Germany) for excellent histological service, Peter Stepper (Stuttgart University, Germany) for providing the dSPn-VPR-mTET3del1ΔC vector, and Olivier Mücke (DKFZ, Germany) for running the Mass array analysis. We also thank Pavlo Lutsik (KU Leuven), Christoph Mayr (Boehringer Ingelheim), Christian Tidona (BioMed X Innovation Center), and Markus Koester (Boehringer Ingelheim) for helpful project discussions. Finally, we thank Melanie Königshoff (University of Pittsburgh, USA) for sharing the Seurat objects of the scRNA-seq data from (Hu et al, 2024).

## Author contributions

**Maria Llamazares-Prada**: Conceptualization; Formal analysis; Validation; Investigation; Visualization; Methodology; Writing—original draft; Writing—review and editing. **Uwe Schwartz**: Conceptualization; Data curation; Formal analysis; Validation; Investigation; Writing—original draft; Writing—review and editing. **Darius F Pease**: Formal analysis; Investigation; Visualization; Writing—review and editing. **Stephanie T Pohl**: Investigation; Visualization. **Deborah Ackesson**: Investigation. **Renjiao Li**: Methodology. **Annika Behrendt**: Investigation. **Raluca Tamas**: Methodology. **Vedrana Stammler**: Methodology. **Mandy Richter**: Methodology. **Thomas Muley**: Resources. **Michael Scherer**: Formal analysis; Investigation; Visualization. **Joschka Hey**: Investigation. **Elisa Espinet**: Methodology. **Claus P Heussel**: Formal analysis. **Arne Warth**: Investigation. **Marc A Schneider**: Resources. **Hauke Winter**: Resources. **Felix JF Herth**: Resources. **Charles D Imbusch**: Resources. **Benedikt Brors**: Resources. **Vladimir Benes**: Resources; Methodology. **David Wyatt**: Supervision. **Tomasz P Jurkowski**: Resources. **Heiko F Stahl**: Supervision. **Christoph Plass**: Supervision. **Renata Z Jurkowska**: Conceptualization; Formal analysis; Supervision; Funding acquisition; Investigation; Visualization; Writing—original draft; Project administration; Writing—review and editing.

Source data underlying figure panels in this paper may have individual authorship assigned. Where available, figure panel/source data authorship is listed in the following database record: biostudies:S-SCDT-10_1038-S44321-026-00386-9.

## Disclosure and competing interests statement

RZJ, MLP, VS, US, RT, SP, and AB received research funding from Boehringer Ingelheim Pharma GmbH & Co KG while employed at BioMed X Institute. HS and DWy are employees of Boehringer Ingelheim Pharma GmbH & Co KG and receive compensation as such. TM received a research grant, non-financial support, and has patent applications with Roche Diagnostics GmbH outside of the described work. CPH holds stock in GSK; received research funding from Siemens, Pfizer, MeVis and Boehringer Ingelheim; consultation fees from Schering-Plough, Pfizer, Basilea, Boehringer Ingelheim, Novartis, Roche, Astellas, Gilead, MSD, Lilly Intermune and Fresenius, and speaker fees from Gilead, Essex, Schering-Plough, AstraZeneca, Lilly, Roche, MSD, Pfizer, Bracco, MEDA Pharma, Intermune, Chiesi, Siemens, Covidien, Boehringer Ingelheim, Grifols, Novartis, Basilea, and Bayer, outside the submitted work. HW received consultation fees from Intuitive and Roche.

# Expanded View Figures

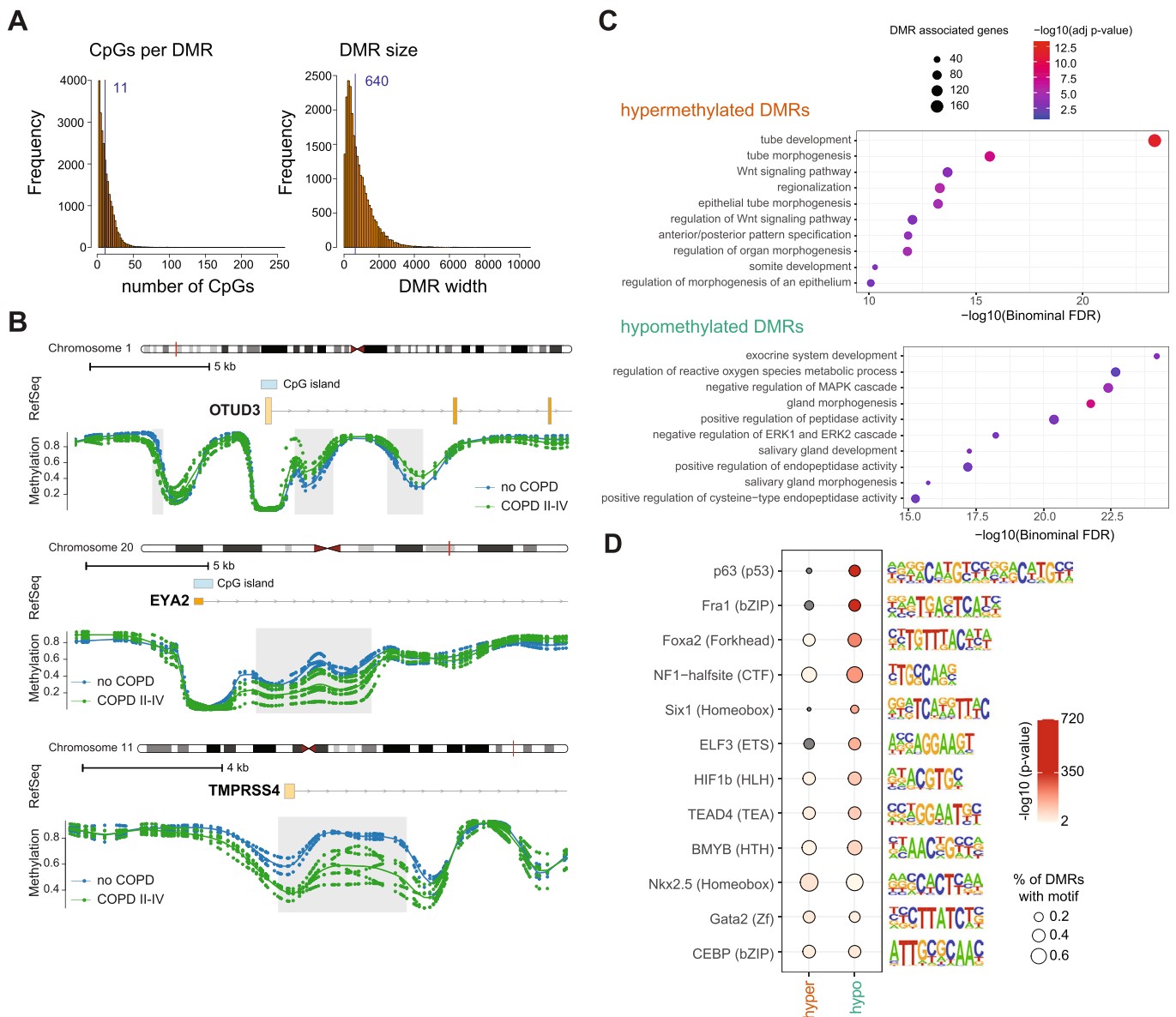

**Figure EV1.  Genome-wide DNA methylation changes occur at regulatory regions in AT2 cells during COPD (supporting information for Fig. 2).**

(A) Number of CpG sites (left panel) and the width of DMRs (right panel) identified between no COPD and COPD II–IV. Median values are indicated in the histograms in dark blue. (B) Detailed view of DMRs showing the methylation profiles of no COPD ($n = 3$) and COPD II–IV ($n = 5$) samples at the indicated genomic regions. DMR locations are highlighted as gray boxes. (C) Functional annotation of genes located next to hypermethylated (top) and hypomethylated (bottom) DMRs using GREAT. Hits were sorted according to the binomial adjusted $P$ value, and the top 10 hits are shown. The adjusted $P$ value is indicated by the color code, and the number of DMR-associated genes is indicated by the node size. (D) Transcription factor motif enrichment in hypermethylated (left) and hypomethylated (right) DMRs. The top motif of each transcription factor family (in brackets) is shown. The node size indicates the percentage of DMRs containing the respective motif, and the color represents the $P$ value of the enrichment analysis. Data information: In (C), $P$ values were calculated using GREAT, which uses a binomial test. The Benjamini–Hochberg method was applied to correct for multiple testing, revealing adjusted $P$ values. Exact $P$ values are included in Dataset EV4. In (D), motif enrichment $P$ values were calculated using HOMER, which uses ZOOPS scoring (zero or one occurrence per sequence) coupled with the hypergeometric enrichment calculations. Exact $P$ values are included in Dataset EV5.

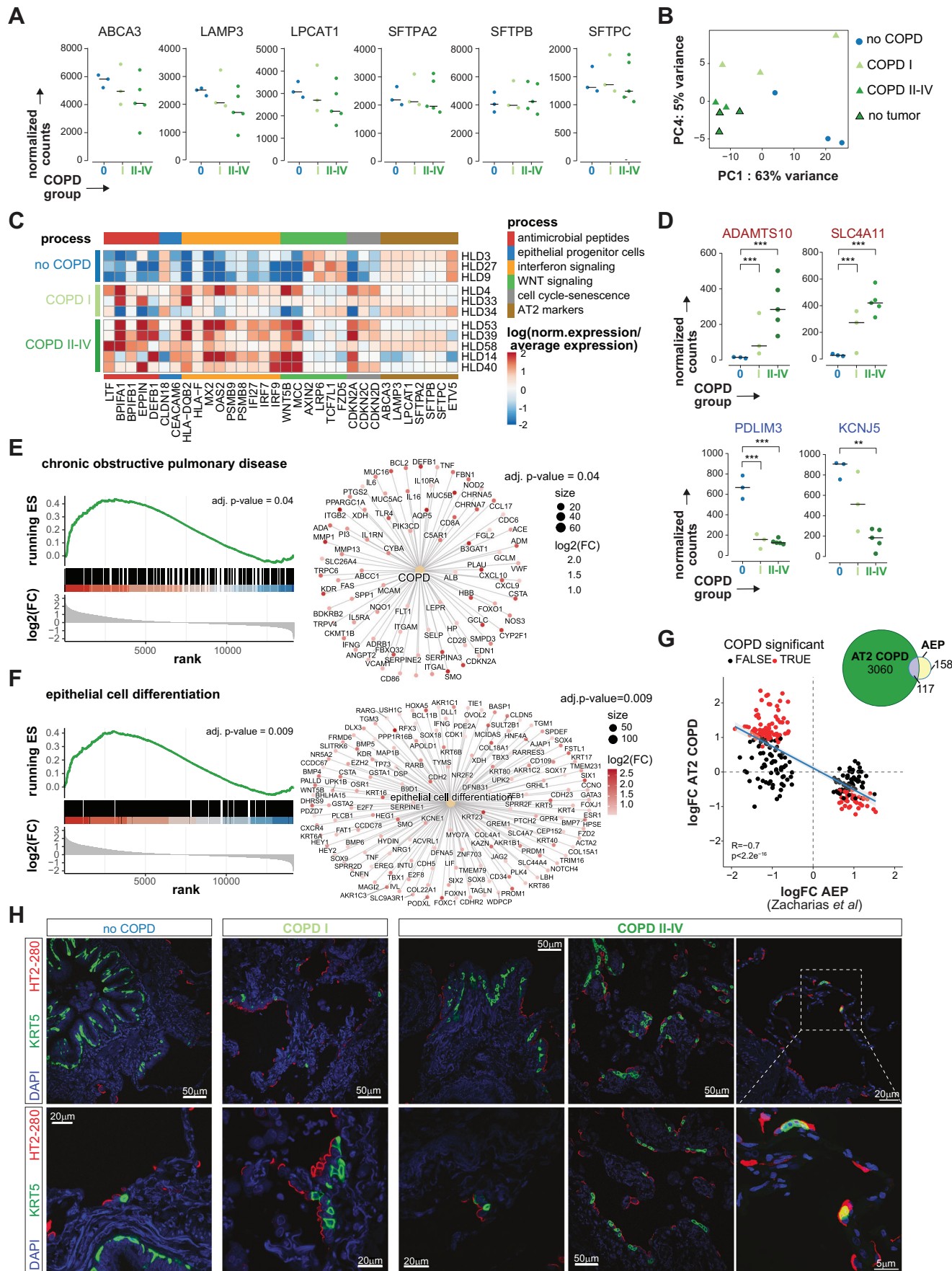

**Figure EV2.  AT2 transcriptome is altered in COPD as disease progresses (Supporting information for Fig. 3).**

(A) Normalized read counts from RNA-seq data for AT2-specific genes in sorted AT2 cells from each donor (dots). Data points represent normalized counts from no COPD (blue, $n = 3$), COPD I (light green, $n = 3$), and COPD II–IV (dark green, $n = 5$). The group median is shown as a black bar. (B) Principal component analysis (PCA) of the 500 most variable genes in RNA-seq. Percentages indicate the proportion of variance explained by PC1 and PC4. COPD I and COPD II–IV samples are represented in light and dark green triangles, respectively, and no COPD samples as blue circles. COPD samples without a cancer background are displayed with a black contour. (C) Heatmap showing expression changes of selected genes across all samples. Genes associated with selected processes are shown. The expression deviation from the average expression across all samples (log(norm. expression/average expression)) is indicated by the color code. (D) Top upregulated (top, red label) and top downregulated (bottom, blue label) genes in AT2 cells from COPD patients. Normalized read counts from RNA-seq data for the specified genes in each donor (dots). Data points represent normalized counts from no COPD (blue, $n = 3$), COPD I (light green, $n = 3$), and COPD II–IV (dark green, $n = 5$). The group median is shown as a black bar. Adjusted $P$ values were calculated using DESeq2, which uses a negative binomial GLM (generalized linear model) and Wald statistics. Significance: **: adj. $P$ value < 0.01, ***: adj. $P$ value < 0.001. Specific adj. $P$ values of either COPD II–IV or COPD I compared to no COPD are as follows for each gene: ADAMTS10: <0.001, <0.001; SLC4A11: <0.001, <0.001; PDLIM3: <0.001, <0.001; KCNJ5: 0.001, 0.72. (E, F) Gene set enrichment analysis (GSEA) results of the gene expression of AT2 cells from COPD II–IV vs no COPD donors. ES enrichment score, NES normalized enrichment score, FDR false discovery rate. GSEA of genes associated with (E) chronic obstructive pulmonary diseases (Diseases Ontology ID: 3083) or (F) epithelial cell differentiation (Gene Ontology: GO:0030855). Genes were sorted based on the log2(fold change) in COPD II–IV (bottom panel). The $P$ value of the GSEA is indicated in the top right corner and calculated empirically from a null distribution of enrichment scores generated by permutations. Gene network plot of the respective terms shows genes associated with the term and driving the enrichment score (leading edge). The beige nodes symbolize enriched terms. The lines connecting the nodes denote the specific genes associated with each term. The color of the gene nodes signifies the expression change in severe COPD compared to no COPD. (G) Top, scatter plot showing genes differentially expressed in alveolar epithelial progenitor (AEP) cells from (Zacharias et al, 2018) compared to AT2 (Zacharias AEP) and their correlation to AT2 cells in COPD II–IV compared to AT2 from no COPD donors from our study (AT2 COPD). DEG in AEP were determined with the same pipeline and cutoffs (red dots; FDR of 10% and | log2(fold change) | >0.5) as used for RNA-seq of no COPD vs COPD II–IV (see "Methods"). DEGs in COPD II–IV are highlighted as red dots. The blue diagonal represents the linear regression between log2(fold-changes) in COPD II–IV against no COPD and AEP against AT2. Shaded areas are confidence intervals of the correlation coefficient at 95%. $P$ value < 2.2e$^{-16}$ was derived from linear regression analysis and the Pearson correlation coefficient ($R = -0.7$) is indicated. Right corner, Venn diagram indicating the overlap of DEG in AT2 COPD and AEP cells from (Zacharias et al, 2018) compared to AT2 cells. (H) Representative immunofluorescence staining images of HT2-280 and KRT5 expression in FFPE human lung tissue slices from no COPD, COPD I and COPD II–IV donors. The zoomed-in panel (right corner, bottom) demonstrates the presence of rare HT2-280/KRT5 double-positive cells in the alveoli of COPD patients. Slides were counterstained with DAPI, scale bars = 50 μm, 20 μm or 5 μm, as displayed in images.

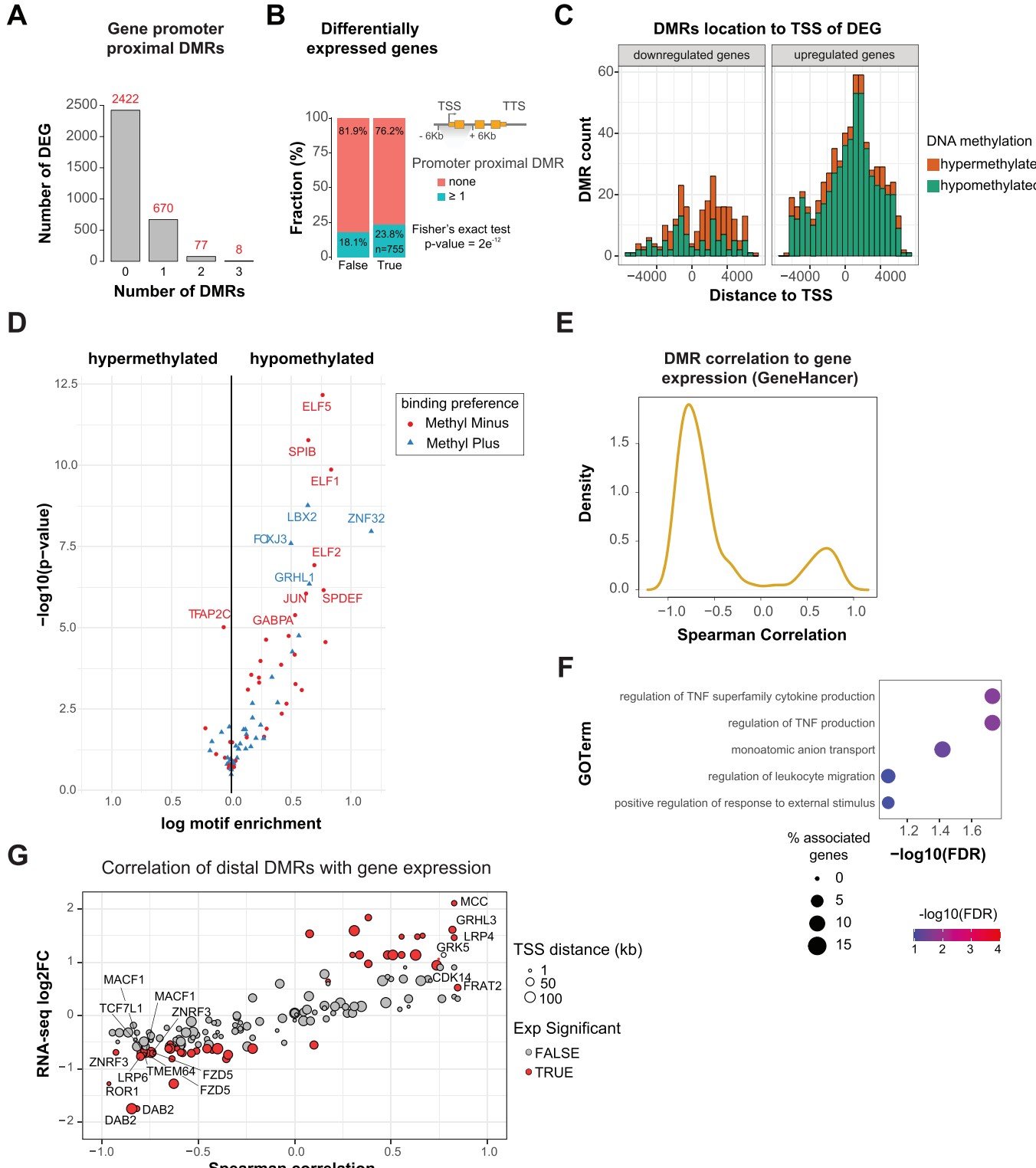

◀ **Figure EV3.  Integrated analysis reveals epigenetic regulation of key pathways in COPD.**

(Supporting information for Fig. 4). (**A**) DMRs within ± 6 kb from the TSS of DEG were assigned to their corresponding gene. (**B**) Fraction of genes associated with at least one DMR in the promoter proximity ($+/-$ 6 kb from TSS, blue) of non-DEG (left, 18.1%) or DEG (right, 23.8%). DMRs are significantly enriched at DEGs (Fisher's exact test $P$ value $= 2e^{-12}$). (**C**) Stacked histogram showing location of hyper- and hypomethylated DMRs relative to the TSS of DEGs in downregulated (left) and upregulated (right) genes. (**D**) Enrichment of methylation-sensitive binding motifs at hypo- (right) and hypermethylated (left) DMRs, using DMRs with a high correlation ( | Spearman correlation coefficient | >0.5) between methylation and gene expression. Methylation-sensitive motifs were derived from Yin et al (Yin et al, 2017b). Transcription factors, whose binding affinity is impaired upon methylation of their DNA binding motif, are shown in red (binding preference: Methyl Minus), and transcription factors, whose binding affinity upon CpG methylation is increased, are shown in blue (binding preference: Methyl Plus). (**E**) Spearman correlation between gene expression and DMR methylation of DMRs assigned to gene regulatory elements using the GeneHancer database. (**F**) GO-Term overrepresentation analysis of DEGs negatively correlated to DMRs in gene regulatory elements. The adjusted $P$ value is indicated by the color code and the percentage number of associated DEGs is indicated by the node size. Exact $P$ values are included in Dataset EV10. (**G**) Scatter plot showing distal DMR-DEG pairs associated with Wnt-signaling. Pairs were extracted from GREAT analysis (hypermethylated, DMR-DEG distance <100 kb; see Fig. EV1C). The $Y$ axis represents the log2 fold change expression in COPD II–IV, and the X-axis denotes the Spearman correlation of the DEG-DMR pair. Node size indicates the distance of the DMR to the TSS. DEGs are highlighted in red.

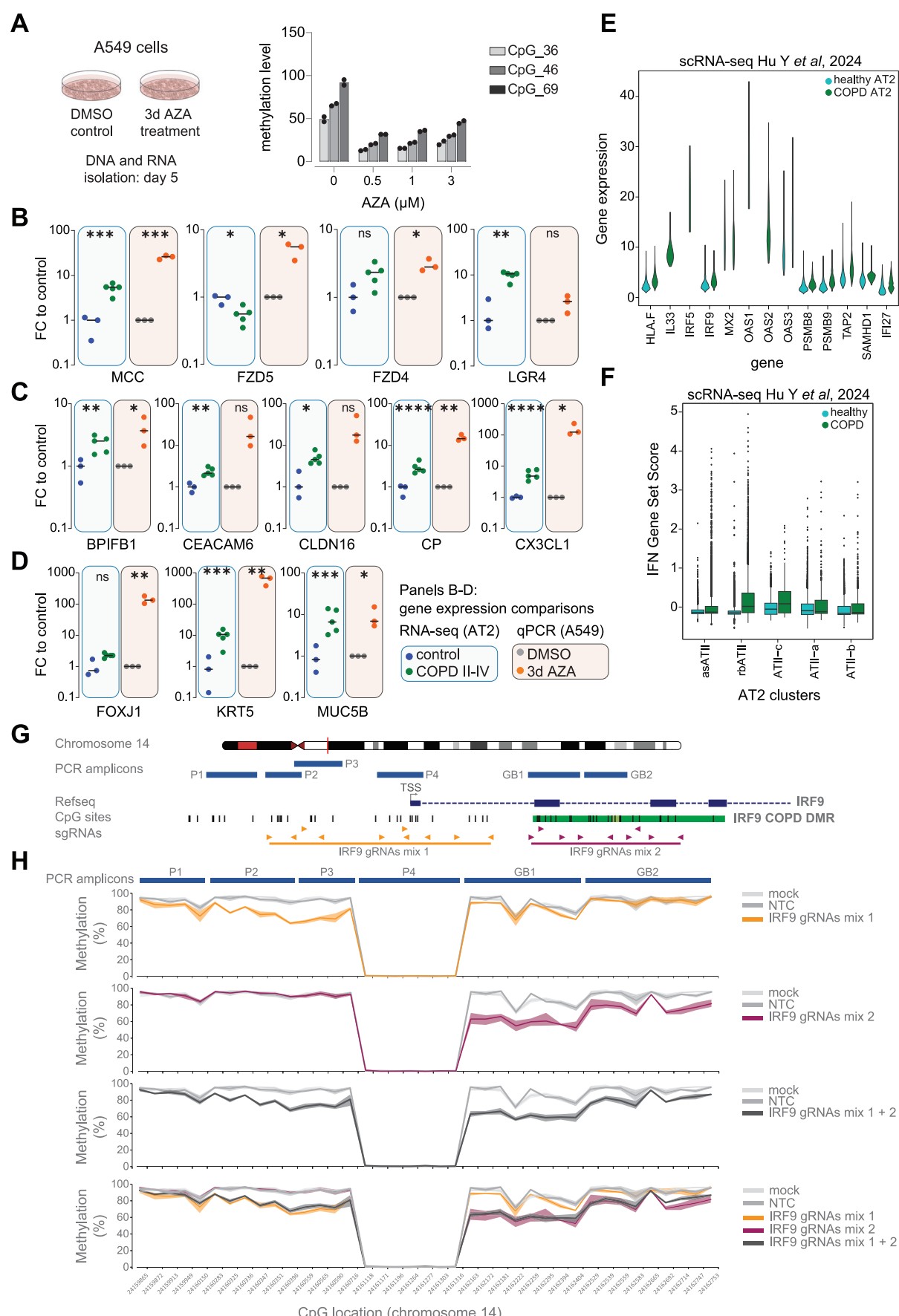

◀

**Figure EV4.    Epigenetic regulation of gene expression in AT2 and A549 cells (Supporting information for Fig. 5).**

(A) LINE methylation levels in A549 cells treated with the indicated amounts of 5-Aza-2′-deoxycytidine (AZA) and measured at three CpG sites by Mass Array ($n = 2$ biological replicates; barplot indicates the mean value; each data point is indicated). (B–D) Fold-change in gene expression of selected genes in AT2 cells in COPD (RNA-seq) and A549 cells treated with 0.5 µM AZA (RT-qPCR) compared to the median of control samples. Genes were selected based on the presence of a DMR and deregulation in AT2 cells in COPD. Left, RNA-seq data from AT2 cells (no COPD, blue, $n = 3$; COPD II–IV, green, $n = 5$), right, A549 treated with AZA (orange, $n = 3$) compared to control DMSO-treated cells (gray, $n = 3$). The group median is shown as a black bar. Selected genes from the Wnt/b-catenin pathway (B), antimicrobial peptides, alveolar progenitor (C) and airway epithelial markers (D) are shown. RNA-seq adjusted (adj.) P values were calculated using DESeq2 with negative binomial generalized linear model (GLM) and Wald statistics. *:adj. P value < 0.05; **: adj. P value < 0.01; ***: adj. P value < 0.001. Gene expression was measured by RT-qPCR using DMSO treatment as control (gray) and RPLP0 as housekeeping gene. For A549 samples, each point represents the mean of two technical replicates, and bars represent the median of 3 independent experiments ($n = 3$). Statistical analysis was performed by paired t test, FDR-corrected using the Benjamini, Krieger, and Yekutieli two-stage set-up method. Significance: *P value < 0.05; **P value < 0.01. (E) Expression values for the indicated genes of the IFN pathway in an external scRNA-seq dataset of AT2 cells from COPD patients and healthy controls (Hu et al, 2024). Y axis shows log-normalized gene expression levels. Boxplots middle line corresponds to the median; the lower and upper hinges correspond to first and third quartiles, respectively; the upper whisker extends from the hinge to the largest value no further than 1.5× the interquartile range (or the distance between the first and third quartiles) from the hinge and the lower whisker extends from the hinge to the smallest value at most 1.5x the interquartile range of the hinge. Data beyond the end of the whiskers are called 'outlying' points and are plotted individually. (F) Combined gene set score of the genes shown in (E) in different subsets of AT2 cells from (Hu et al, 2024). The IFN signature genes were identified in our integrative analysis of TWGBS and RNA-seq in sorted AT2 cells. Boxplots are defined as in (E). (G) Graphical representation of the IRF9 locus. In dark blue is the IRF9 coding region, with introns and exons represented by dashed and solid boxes, respectively. At the bottom, arrows represent individual gRNAs targeting the IRF9 promoter (orange) and gene body (magenta) regions, with overlapping lines representing the groups of gRNAs that comprise mix 1 and mix 2, respectively. Bisulfite-PCR amplicons targeting IRF9 (blue bars) are depicted as blue boxes, with P1-4 targeting the IRF9 promoter region, and GB1-2 targeting the gene body region. Individual CpG sites are depicted by black bars below the IRF9 coding region, with the light and dark green overlapping region displaying the core and extended differentially methylated regions, respectively, identified in the AT2 COPD T-WGBS data. Genomic positions were extracted from human genome assembly 38 (hg38) using the UCSC genome browser. (H) CpG-methylation percentage at individual IRF9 CpG sites in epi-edited A549 cells. The percentage of CpG-methylation for mix 1, 2, and 1 + 2 (orange, magenta, and dark gray) transfected samples are plotted separately against the pUC19 mock (mock) and non-targeted (NTC) transfection controls. For each sample, the opaque line plots the mean value, and the faint surrounding area plots the observed range across repeats. The bisulfite PCR targets are displayed across the top, showing which CpG-sites are sequenced by each target, and how these sites relate to the genomic locations of the sequencing targets seen in panel. Data information: In (B–D), the specific q-values for each gene from A549 treated samples compared to DMSO control are as follows: MCC, 0.001865; FZD4, 0.010078; FZD5, 0.010078; LGR4, 0.037818; BPIFB1, 0.027136; CX3CL1, 0.002964; CEACAM6, 0.013600; CLDN16, 0.011915; CP, 0.002817; FOXJ1, 0.037818; KRT5, 0.001715; MUC5B, 0.002530. For the ATII RNA-seq analysis, the specific adjusted P values for each gene can be found in Dataset EV6.

