## [Peer Review File · EMBO Molecular Medicine]

Epigenetic dysregulation of IRF9 drives excessive interferon signalling in COPD

Maria Llamazares Prada, Uwe Schwartz, Darius Pease, Stephanie Pohl, Deborah Ackesson, Renjiao Li, Annika Behrendt, Raluca Tamas, Vedrana Stammler, Mandy Richter, Thomas Muley, Michael Scherer, Joschka Hey, Elisa Espinet, Claus Heussel, Arne Warth, Marc Schneider, Hauke Winter, Felix Herth, Charles Imbusch, Benedikt Brors, Vladimir Benes, David Wyatt, Tomasz Jurkowski, Heiko Stahl, Christoph Plass, and Renata Jurkowska

Corresponding author: Renata Jurkowska (jurkowskar@cardiff.ac.uk)

Review Timeline:

Transferred from Review Commons:	12th Aug 25
Editorial Decision:	16th Oct 25
Revision Received:	17th Dec 25
Accepted:	29th Jan 26

Editor: Zeljko Durdevic

Transaction Report:

Review
COMMONS

This manuscript was transferred to EMBO Molecular Medicine following peer review at Review Commons.

Review #1**1. Evidence, reproducibility and clarity:****Evidence, reproducibility and clarity (Required)******Summary:****

This study by Prada et al. aimed to explore DNA methylation and gene expression in primary EpCAM^{high}/PDPN^{low} cells, consisting for (probably) the largest part of AT2 cells, to understand the molecular mechanisms behind the impaired regeneration of alveolar epithelial progenitor cells in COPD. They found that higher or lower promoter methylation in COPD-associated cells was inversely correlated with changes in gene expression, with interferon signaling emerging as one of the most upregulated pathways in COPD. IRF9 was identified as the master regulator of interferon signaling in COPD. Targeted DNA demethylation of IRF9 in an A549 cell line resulted in a robust activation of its downstream target genes, including OAS1, OAS3, PSMB8, PSMB9, MX2 and IRF7, demonstrating that demethylation of IRF9 is sufficient to activate the IFN signaling pathway, validating IRF9 as a master regulator of IFN signaling in (alveolar) epithelial cells.

****Major comments:****

- To remove airways (and blood vessels) completely from the lung tissue is difficult, if not impossible. This means that the assumption that the sorted EpCAM^{pos}/PDPN^{low} cells primarily consisted of AT2 cells remains valid only if a quantitative analysis is conducted on the proportion of HT2-280^{pos} cells in all samples in cytopins to exclude any significant contamination from bronchial epithelial cells. If authors cannot demonstrate >95% pure HT-280-positive cells, then the key conclusions suggesting that the epigenetic regulation of the IFN pathway might be crucial in AT2 progenitor cell regeneration could also potentially apply to bronchial progenitor cells. In addition, if >95% purity can not be demonstrated, the data should be adjusted to account for differences in cell type composition.

- The overrepresentation of several keratins (KRT5, KRT14, KRT16, KRT17), mucins (MUC12, MUC13, MUC16, MUC20) and the transcription factor FoxJ1 is now attributed by the authors to a possible dysregulation of AT2 identity and differentiation in COPD (lines 282 - 284) where they cite refs 28, 69, 70. Authors try to support this with IF double stains for KRT5 and HT-280 to identify co-expression of KRT5 and HT2-280 in lung tissue (Figure S2H). However, the evidence for the co-expression of both markers could be presented more convincingly.

- Double staining for KRT5 and HT2-280 did highlight the proximity of both cell types in lung tissue, underscoring the challenge of removing airways (including the smaller and terminal bronchi) from the tissue. In addition, HT-280/KRT5 co-expression is not consistent with recent studies from refs 28, 69, 70 where other markers for distal airway cell transition, such as SCGB3A2 and BPIFB1, have been demonstrated, which were not investigated in this study.

- The small (and not evenly divided) sample size of both COPD and non-COPD specimens may lead to a higher risk for false positive results as adjustments for multiple testing typically rely on the number of comparisons, and small sample sizes may not provide enough data points to adequately control for this.

****Minor comments:****

Introduction:

- In general, refer to the actual experimental studies rather than review papers where appropriate.

- Clearly specify whether a study was conducted in mice or humans, as this distinction is crucial for understanding the relevance of the findings to COPD.

Methods:

- Line 473, here is meant 3 ex-smoker controls instead of smoker controls?

Discussion:

- A list of limitations should be added to the discussion. One is the use of the alveolar cell line A549, which produces mucus, a characteristic more commonly associated with bronchial epithelial cells. (ref 43)

- Another limitation to consider is that cells were isolated primarily from individuals with lung cancer, except for patients with COPD stage IV. In particular as COPD stage II and IV samples were taken together.

- And discuss the small and unevenly divided sample size

References:

- Check references. For instance, there is no reference in the text to ref 43.

- Align format of references

2. Significance:

Significance (Required)

The strength of this study lies in its focus on the molecular mechanisms underlying the impaired regeneration of epithelial progenitor cells in COPD. The discovery of IRF9, which regulates IFN signaling and is prominently upregulated in COPD, together with the convincing validation of the epigenetic control of the IFN pathway by targeted DNA demethylation of the IRF9 gene, adds significant value to the COPD research field.

Main limitations of the study are the relatively small sample size of both COPD and non-COPD specimens and the claim that the sorted EpCAMpos/PDPNlow cells primarily consisted of AT2 cells.

- Describe the nature and significance of the advance (e.g. conceptual, technical, clinical) for the field.

The nature and significance of the advance in epigenetic editing of IRF9 in COPD can be described as both conceptual and potentially clinical:

Conceptual Advance: The epigenetic editing of IRF9 enhances our understanding of the molecular mechanisms underlying COPD pathogenesis. By targeting IRF9 through epigenetic modifications, researchers were able to modulate the activity of the IFN pathway, which plays a crucial role in the immune response and lung tissue regeneration. This approach offers insights into the epigenetic regulation of gene expression in epithelial progenitor cells in COPD and expands our understanding of how alterations in specific gene methylation could contribute to disease progression.

Clinical Significance: The potential clinical significance of epigenetic editing of IRF9 lies in its implications for COPD therapy. If successful, epigenetic editing techniques could offer a novel therapeutic strategy for COPD by downregulating IFN pathway activation and promoting regeneration of epithelial progenitor cells in the lungs. Obviously, further preclinical and clinical studies are needed to validate the efficacy and safety of epigenetic

editing approaches in COPD treatment.

- Place the work in the context of the existing literature (provide references, where appropriate).

Few experimental papers have been published on epigenetic editing in lung diseases, with limited research available beyond the study referenced in citation 43.

Song J, Cano-Rodriguez D, Winkle M, Gjaltema RA, Goubert D, Jurkowski TP, Heijink IH, Rots MG, Hylkema MN. Targeted epigenetic editing of SPDEF reduces mucus production in lung epithelial cells. *Am J Physiol Lung Cell Mol Physiol*. 2017 Mar 1;312(3):L334-L347. doi: 10.1152/ajplung.00059.2016. Epub 2016 Dec 23. PMID: 28011616.

- State what audience might be interested in and influenced by the reported findings.

This study is of broad interest to researchers investigating the pathogenesis and treatment of COPD.

- Define your field of expertise with a few keywords to help the authors contextualize your point of view.

Expertise in: Lung pathology, Immunology, COPD, Epigenetics

- Indicate if there are any parts of the paper that you do not have sufficient expertise to evaluate.

Less expertise in: Epigenetic Editing

3. How much time do you estimate the authors will need to complete the suggested revisions:

Estimated time to Complete Revisions (Required)

(Decision Recommendation)

Between 3 and 6 months

4. Review Commons values the work of reviewers and encourages them to get credit for their work. Select 'Yes' below to register your reviewing activity at Web of Science

Reviewer Recognition Service (formerly Publons); note that the content of your review will not be visible on Web of Science.

Yes

Review #2

1. Evidence, reproducibility and clarity:

Evidence, reproducibility and clarity (Required)

****Summary:****

This study aims to understand the molecular mechanisms underlying dysfunction in AT2 cells in COPD, by profiling bulk genome wide DNA methylation using Tagmentation-based whole-genome bisulfite sequencing (T-WGBS) and RNA sequencing in selectively sorted primary AT2 cells. The study stands out in its sequencing breadth and use of an incredibly difficult cell population, and has the potential to add substantially to our mechanistic understanding of epigenetic contributions to COPD. A further highlight is the concluding aspect of the study where the authors undertook targeted modification of specific CpG methylation, provided direct, site-specific evidence for transcriptional regulation by CpG methylation.

****Major comments:****

The authors clearly show that there is DNA methylation alteration in AT2 cells from COPD individuals that links functional to gene expression at some level. However, I think the statement "to identify genome-wide changes associated with COPD development and progression..." and similar other references to disease development understanding is not accurate given the DNA methylation primary comparison is between control and moderate to severe COPD, with no temporal detail or evidence that they drive progression rather than are a result of COPD development. The paragraph starting on line 186 where this is addressed to some extent is quite vague and doesn't really provide confidence that DNAm dysregulation occurs at an early stage in this context. This can be addressed by changing the focus/style of the text.

Results comments and suggestions:

For the integrated analysis, there is a focus on DMRs in promoters with very little analysis on other regions. The paragraph starting on line 317 describes some analysis on enhancers but is very brief, doesn't include information on how many/which DMRs were included, making it hard to interpret the impact of the 147 DMRs and 93 genes identified - is this nearly all DMRs and genes analysed or very few? A comparison to the promoter analysis would be of interest. Especially as the targeted region followed up with lovely functional assessment in the last sections is a gene body DMR, not a promoter DMR.

- Lines 299-301 - I'm not sure the graph in Fig S3A support the conclusion that there was a preferential negative relationship between DNAm and gene expression. Looks like there are a substantial number of cases where a positive relationship is observed and this needs to be acknowledged.

- Line 307 - what are the "analysed DEGs"? Are they the methylation associated genes?

- Line 307-309 - "Among the analyzed DEGs, 76.5% (492) displayed a negative correlation (16.8% of the total DEGs), indicating a possible direct regulation by DNA methylation, while 23.5% (151) showed a positive correlation between gene expression and DNA methylation"

- are the authors suggesting the positive correlation doesn't indicate direct regulation?

- Line 313 - why did the authors focus on only negatively correlated genes to identify their top dysregulated pathway of IFN signalling? Why not do pathway analysis on the DNAm associated genes separately to identify DNAm associated pathways?

- A comparison of the gene expression data with previous data in AT2 cell/single cell data would strengthen the gene expression section.

- The paragraph starting on line 173 feels a little redundant when we know there is RNA available to test if the differential DNAm links to altered gene expression - this selected of example regions/genes would be better placed after the gene expression has been reported, at which point you could say whether the linked genes displayed altered transcription.

- Similarly, the TF enrichment analysis is great but maybe would have added value to be done on DNA regions later shown to be linked to differential expression - was there different enrichment at DNA regions that are vs are not associated with altered expression? And could you test in vitro whether changing methylation of DNA (maybe a blunt tool like 5-aza would be ok) alters TF binding (cut+run/ChIP?). Furthermore it would be interesting to understand the TF sensitivity analysis within the context of positive versus negative DNA

methylation:gene expression correlations.

Methods:

- The authors should include more detail of the TWGBS rather than directing the reader to a previous publication. Also DNA concentration post bisphite conversion would be a useful metric to provide.

- Differential DNA methylation analysis: It is stated that DNA regions had to contain 3 CpG sites but was this within a defined DNA size range?

- Reference genome only provided for RNAseq not TWGBS?

- The tables do not appear in the PDF and I struggled to tally to the "Dataset" files provided if that is what they were referring to?

- For the gene expression analysis, can it be made clearer that a full analysis was done on COPD I samples. It is a little confusing to the reader as this was not done for DNAm so might be assumed the same targeted analysis on only genes found to be differentially expressed between control and COPD II-IV, but that cannot be the case as an overlap of COPD1 vs COPD II-IV genes if provided. For this overlap, do genes show the same effect direction?

- Replication is difficult on these studies as the samples are so difficult to come by. Also limited by sample size for the same reason. It doesn't mean the study is not worth doing and the data are still valuable. However, it may be pertinent to include technical validation of a few regions of interest, acknowledge the limitation (along side strengths) in the discussion, and perhaps provide actual p value rather than blanket $< p 0.1$, seems very lenient but may all be super significant (this may already be in the tables I wasn't able to find).

- It isn't clear to me if DNA and RNA are from the same cells? The results say "cells matching those used for T-WGBS" but the methods suggest separate extractions so not the same cells? If they are not the same cells a comment on the implications of this should be included in the discussion for example, potentially some differences in cell type composition, storage time etc.

- Line 193 the authors say "Since DMRs were overrepresented at cis-regulatory sites...." -

"cis" needs to be defined. If you link DNAm regions to gene via "closest gene" does this not automatically mean you're outputs will be cis? Just needs better definition/explanation.

****Minor comments:****

- Line 157: "we identified site-specific differences....". Change to region specific?

- Line 102-103: needs a reference for the statement "Alterations in DNA methylation patterns have been implicated....."

- Line 266 - what does "strong dysregulation" mean? Large fold change, very significant?

- Lines 423-425 - statement needs a reference

- Line 428 - word missing between "epigenetic , we"?

- Prior studies are well references, text and figures are clear and accurate.

2. Significance:

Significance (Required)

This study has several strengths:

1) Sample collection and characterisation. AT2 cells are incredibly hard to come by and the authors should be commended to generating the samples. However, proximity to cancer is always a potential issue, especially in epigenetic studies. Is it feasible to include any analysis to show the samples derived from those with cancer don't drive the changes observed? Even a high level PCA or an edit of fig 2A with non-cancer in a different colour in supplemental - looks like there is one outlier, is that a non-cancer? Or a correlation of change in beta between control and cancer/COPD and control and non-cancer:COPD (for want a better phrase!). just an indicator that the non-cancer COPD samples are not driving differences.

2) This is the first time DNAm has been profiled in AT2 cells. It is incredibly difficult, valuable and novel data that will increase the fields capability technically, their understanding of functional mechanisms and potential translation considerably. It's audience will be primarily translational respiratory however the fundamental science aspect of gene expression regulation by DNA methylation will have wider reach across

developmental and disease science.

3) the functional analysis using targeted CRISPR-Cas9 is very well done and adds impact.

Potential weaknesses/areas for development:

I feel the main weakness is the in the section integrating DNA methylation and gene expression. The rationale for a focus on various aspects, for example inversely related DNAm/gene expression pairs, the IFN pathway and IRF9, are not clear. Also further understanding of the differences between DNAm associated genes and non-DNAm associated genes could be expanded, at the pathway level, TF regulation level, effect size level (are DNAm associated changes to gene expression larger, enriched for earlier differential expression)

The authors could comment on potential masking of differences between 5hmC and mC and the implications it may have

Furthermore, while the rationale for looking at DMRs is clear, especially given the sample number, I am interested to understand what proportion of the assayed CpGs "fit" within the cut off stipulations of the DMR analysis - that is, is their potentially COPD effects at sparse CpG regions/individual CpG sites that are not being identified. A comment on this would be useful and seems the strength of profiling genome wide. I'm happy genomewide is beneficial it just feels a little circular that the authors have chosen whole genome to avoid the bias of the Illumina array and a focus on promoters, but have primarily reported promoter DNAm. This caught my attention again in the discussion where the authors state that cis-regulatory regions were also identified in their fibroblast data is this finding a factor of the analysis performed? (also a comparison of regions Id'ed in AT2 cells versus fibroblasts would be really interesting for a future paper)

My expertise:

Respiratory, cell biology, epigenetics.

3. How much time do you estimate the authors will need to complete the suggested revisions:

Estimated time to Complete Revisions (Required)

(Decision Recommendation)

Between 3 and 6 months

Yes

Full Revision

Manuscript number: RC-2024-02418

Corresponding author(s): Renata, Jurkowska

1. General Statements

We would like to thank the reviewers for taking the time to review our manuscript and for providing valuable comments on how to improve it. We are pleased to see that both reviewers recognize the novelty and importance of our study, its conceptual advance and potential clinical significance. They also noted the novelty and value of our functional mechanistic approach using epigenetic editing. Below, we provide a point-by-point response to their questions and points raised. The changes introduced in response to their feedback are highlighted in yellow in the revised manuscript file.

Reviewer #1 (Evidence, reproducibility and clarity (Required)):

Summary This study by Prada et al. aimed to explore DNA methylation and gene expression in primary EpCAM^{high}/PDPN^{low} cells, consisting of for (probably) the largest part of AT2 cells, to understand the molecular mechanisms behind the impaired regeneration of alveolar epithelial progenitor cells in COPD. They found that higher or lower promoter methylation in COPD-associated cells was inversely correlated with changes in gene expression, with interferon signaling emerging as one of the most upregulated pathways in COPD. IRF9 was identified as the master regulator of interferon signaling in COPD. Targeted DNA demethylation of IRF9 in an A549 cell line resulted in a robust activation of its downstream target genes, including OAS1, OAS3, PSMB8, PSMB9, MX2 and IRF7, demonstrating that demethylation of IRF9 is sufficient to activate the IFN signaling pathway, validating IRF9 as a master regulator of IFN signaling in (alveolar) epithelial cells.

Major comments:

- To remove airways (and blood vessels) completely from the lung tissue is difficult, if not impossible. This means that the assumption that the sorted EpCAM^{pos}/PDPN^{low} cells primarily consisted of AT2 cells remains valid only if a quantitative analysis is conducted on the proportion of HT2-280^{pos} cells in all samples in cytopspins to exclude any significant contamination from bronchial epithelial cells. If authors cannot demonstrate >95% pure HT-280-positive cells, then the key conclusions suggesting that the epigenetic regulation of the IFN pathway might be crucial in AT2 progenitor cell regeneration could also potentially apply to bronchial progenitor cells. In addition, if >95% purity cannot be demonstrated, the data should be adjusted to account for differences in cell type composition.

Response:

We thank the reviewer for raising this important point. Although, as pointed out by the reviewer, we cannot guarantee that our sorted cells do not contain a minor contamination from respiratory / terminal bronchial cells, we carefully selected donors, tissue regions, and sorting strategy to ensure the highest possible enrichment of AT2 cells, as we explain below. We have now expanded the methods and results section and covered this point in the manuscript discussion.

Full Revision

- The lung tissue pieces we received were distal, as evidenced by the presence of pleura. We collected representative tissue pieces for histology to validate sample quality. Our protocol includes a dissection of all visible airways and vessels using a dissecting microscope, which were cryopreserved separately from distal parenchyma. Hence, the starting material for tissue dissociation was depleted from airways and vessels. The importance of vessel/airway removal for enrichment of distal alveolar cells was established by Tata's group (PMID: 35712012).
- We selected the AT2 sorting protocol (EpCAM^{pos}/PDPN^{low}) based on previous publications that used tissue from both healthy and COPD lungs to separate AT2 cells from AT1 and airway basal cells, as AT1 and basal cells are both PDPN^{high} (PMID: 22033268, PMID: 23117565; PMID: 35078977). This protocol was favoured due to the lack of information about HT2-280 expression and distribution in COPD lungs.
- The sort quality for each sample was assessed by the FACS analysis (back sorting) of the sorted cells, where we observed 95-97% purity (EpCAM^{pos}/PDPN^{low}, **Fig. 1G** shown below). In addition, we validated the sorting protocol and high AT2 enrichment from both no COPD and COPD tissues by immunostaining the FACS-sorted cells with HT2-280, an AT2 marker widely used in the field (strategy suggested by the reviewer) and observed that close to 100% of cells were positive for this marker (**Fig. 1H** shown below). However, we could not do it retrospectively for those patients, where we didn't have enough material. Sorting primary AT2 from small tissue pieces is challenging, and we need at least 20,000 cells to obtain high-quality methylation & RNA-seq data.
- AT2 marker genes (ABCA3, LPCAT1, LAMP3 and the surfactant genes SFTPA2, SFTPB and SFTPC) were among the top highly expressed genes in our RNA-seq data and were not significantly changed in COPD (please see expression data in **Fig. S2A** in the manuscript, and below for convenience), as well as **Table 6**, providing further evidence that the sorted cells carry a strong AT2 transcriptional signature.

Figure 1

Figure S2

Fig. 1G FACS plot examples showing the analysis of sorted AT2 cells (back sorting) from control (blue) and COPD (green) donors displayed over total cell lung suspensions (grey) **H** Representative IF staining of HT2-280 expression in sorted AT2 cells from no COPD (top) and COPD (bottom) donors. Nuclei (blue) were stained with DAPI, scale bars=20µm **Fig. S2A** Normalized read counts from RNA-seq data for AT2-specific genes in sorted AT2 cells from each donor (dots). Data points represent normalised counts from no COPD (blue), COPD I (light green) and COPD II-IV (dark green). Group median is shown as a black bar.

- In agreement with a previous study which profiled bulk AT2 using expression arrays (PMID: 23117565), we also observed upregulation of IFN signaling pathway in COPD AT2s. The enrichment of IFN α/β signature was also observed in COPD in the inflammatory AT2 cluster (iAT2) in a recent scRNA-seq study (PMID: 36108172). As part of the revision, we compared the IFN gene signature identified in our bulk AT2 RNA-seq with a recent scRNA-seq study (published after the submission of our manuscript, PMID: 39147413) that profiled EpCAM^{pos} cells from COPD and non-smoker donor lungs. We observed an upregulation of our IFN signature genes in AT2 in COPD (mostly in AT2c and rbAT2 subsets), suggesting that similar signatures were observed in COPD AT2s in this dataset as well (please see **Fig. S4E-F** below).

Figure S4

Figure S4E Expression values for the indicated genes of the IFN pathway from an external scRNA-seq dataset of AT2 cells from COPD patients and healthy controls (Hu et al, 2024). Y-axis shows log-normalized gene expression levels. **F**. Combined gene set score of the genes shown in (E) in different subsets of AT2 cells from Hu et al, 2024. The IFN signature genes were identified in our integrative analysis of TWGBS and RNA-seq in sorted AT2 cells.

- We have also carefully examined DNA methylation profiles across all samples. The density plots of our T-WGBS DNA methylation data are very similar among the individual samples in all 3 groups, indicating that the sorted cells consist mostly of a single cell type, as there are no obvious intermediate (25-75%) methylation peaks, as observed in cell mixtures (**Fig. 2A** and the panel below). No reference DNA methylation profiles are available for respiratory or terminal bronchial cells; hence, we cannot compare how epigenetically different these cells would be from AT2 nor perform a deconvolution for potential minor contamination with distal airway cells.

DNA methylation density plots of sorted $EpCAM^{pos}/PDPN^{neg}$ cells from no COPD (blue, $n=3$), COPD I (light green, $n=3$) and COPD II-IV (dark green, $n=5$) showing a homogeneous methylation pattern and low abundance at intermediate (25%-75%) methylation values across all profiled samples, indicating that the sorted cells were mostly of a single cell type.

- We have now added a sentence to the limitations section of the discussion to cover that point specifically.

Full Revision

CHANGES IN THE MANUSCRIPT:

AT2 cells were isolated by fluorescence-activated cell sorting (FACS) from cryopreserved distal lung parenchyma, depleted of visible airways and vessels of three no COPD controls, three COPD I and five COPD II-IV patients as previously described (24, 52, 53)

The isolated cells were positive for HT2-280, a known AT2 marker (54), as confirmed by immunofluorescence (Fig. 1H), validating the identity and high enrichment of the isolated AT2 populations.

Known AT2-specific genes, including ABCA3, LAMP3 and surfactant genes (SFTPA2, SFTPB and SFTPC) were among the top highly expressed genes and were not significantly changed in COPD AT2s (Fig. S2A, Table 6), further confirming the AT2-characteristic transcriptional signature of our isolated cells.

However, 5-AZA is a global demethylating agent, and the observed effects may not be direct. To validate the epigenetic regulation of central AT2 pathways further, we took advantage of locus-specific epigenetic editing technology (73). We focused on the IFN pathway because it was the most significantly enriched Gene Ontology (GO) term in our integrative analysis of TWGBS and RNA-seq data. Several IFN pathway members had associated hypomethylated DMRs within promoter-proximal regions and concomitant increased gene expression (Fig. 4C and S2C). Additionally, we confirmed the elevated expression of IFN-related genes with associated DMRs identified in our study in AT2 cells and AT2 cell subclusters from a recently published scRNA-seq cohort (74) (Fig. S4E-F).

We observed upregulation of multiple IFN genes in AT2 in COPD, consistent with a previous expression array study (24). IFN α / β signaling was also enriched in COPD patients in the inflammatory AT2 cluster (iAT2) in a recent scRNA-seq study (84) and our INF signature genes were also upregulated in AT2c and AT2rb subsets in COPD, identified by another scRNA-seq study recently (74).

Finally, despite careful removal of airways from distal lung tissue using a dissecting microscope, we cannot exclude the presence of some terminal/respiratory bronchiole cells in our FACS-isolated EpCAM^{pos}/PDPN^{low} population. Recent scRNA-seq studies provided an unprecedented resolution and identified several epithelial subpopulations and transitional cells residing in the terminal/respiratory bronchioles and alveoli, including respiratory airway secretory cells (93), terminal airway-enriched secretory cells (28), terminal bronchiole-specific alveolar type-0 (AT0) (70), and emphysema-specific AT2 cells (74). These cells may contribute to alveolar repair in healthy and COPD lungs; however, with our bulk DNA methylation and RNA-seq study, we are unable to resolve all these subpopulations. Future development of single-cell methylation and non-reference-based algorithms for DNA methylation deconvolution will enable deeper epigenetic phenotyping of specific AT2 and bronchiolar cell subsets.

(Methods) Validation of IFN gene upregulation in a published scRNA-seq dataset

scRNA-seq data from (74), generously provided by M. Köningshoff, were processed using the default Seurat workflow (117). Expression of IFN-related genes was extracted and plotted as log-normalised gene expression levels in AT2 cells from control and COPD donors. Seurat's AddModuleScore() function was used to compute a gene set score for a custom IFN program

using the genes listed in Fig. S4E and to analyse the IFN gene set scores in AT2 cell subclusters identified in (74). Briefly, average gene expression scores were computed for the gene set of interest, and the expression of control features (randomly selected) was subtracted as described in (118).

Fig. S4E and F: E. Expression values for the indicated genes of the IFN pathway from an external scRNA-seq dataset of AT2 cells from COPD patients and healthy controls (74). Y-axis shows log-normalized gene expression levels. **F.** Combined gene set score of the genes shown in (E) in different subsets of AT2 cells from (74). The IFN signature genes were identified in our integrative analysis of TWGBS and RNA-seq in sorted AT2 cells.

- The overrepresentation of several keratins (KRT5, KRT14, KRT16, KRT17), mucins (MUC12, MUC13, MUC16, MUC20) and the transcription factor FoxJ1 is now attributed by the authors to a possible dysregulation of AT2 identity and differentiation in COPD (lines 282 - 284) where they cite refs 28, 69, 70. Authors try to support this with IF double stains for KRT5 and HT-280 to identify co-expression of KRT5 and HT2-280 in lung tissue (Figure S2H). However, the evidence for the co-expression of both markers could be presented more convincingly.

Response:

We found the potential co-expression of airway and alveolar markers in COPD lungs interesting and hence included it in the original manuscript. The initial discovery came from our bulk RNA-seq data, where we observed upregulation of several genes typically found in more proximal airways in COPD (mentioned above by the reviewer). Of note, some of them (e.g., FoxJ1) are expressed at very low levels. Following reviewer's comments, to validate possible colocalization of AT2 and airway markers on protein level, we performed further IF analysis. We took Z-stack images to demonstrate the co-localization of HT2-280 and Krt5 more convincingly and co-stained the same tissue regions with SCGB3A2 (a TASC/distal airway cell marker, PMID 36796082). Even though these are rare events, we were able to reproduce the existence of HT2-280/Krt5 positive, SCGB3A2 negative cells in the alveoli of COPD patients on the protein level (Fig. S2H and panels below). Although interesting, we decided to keep this finding in the supplement and did not include it in the discussion to focus the story on the epigenetic regulation of the IFN pathway, which is the main discovery of our study. We will investigate this observation in future studies.

Partially in Figure S2H: Examples of HT2-280/Krt5 double positive cells. Top, immunofluorescence staining of the alveolar region of a COPD II donor showing the existence of AT2 cells (HT2-280 positive (red), which are SCGB3A2 negative (green, left) but KRT5 positive (green, right). In conclusion, double-positive HT2-280/KRT5 cells are rare but present in the alveoli of COPD patients. Magnification: 20x. Scale bar: 50 μ m. Bottom, Z-stack images highlighting HT2-280 (red) and KRT5 (green) double-positive cells at 63x magnification. Scale bar: 5 μ m.

CHANGES IN THE MANUSCRIPT:

In addition, we observed an upregulation of several keratins (KRT5, KRT14, KRT16, KRT17) and mucins (MUC12, MUC13, MUC16, MUC20), suggesting a potential dysregulation of alveolar epithelial cell differentiation programs in COPD (Table 6, Fig. S2F). Immunofluorescence staining confirmed the presence of KRT5-positive cells in the distal lung in COPD and identified cells positive for both KRT5 and HT2-280 (Fig. S2H). Collectively, these results indicate a dysregulation of stemness and identity in the alveolar epithelial cells in COPD.

Fig. S2H legend: The zoomed-in panel (right corner, bottom) demonstrates the presence of rare HT2-280/KRT5 double-positive cells in the alveoli of COPD patients. Slides were counterstained with DAPI, scale bars = 50µm, 20µm or 5µm, as displayed in images.

- Double staining for KRT5 and HT2-280 did highlight the proximity of both cell types in lung tissue, underscoring the challenge of removing airways (including the smaller and terminal bronchi) from the tissue. In addition, HT-280/KRT5 co-expression is not consistent with recent studies from refs 28, 69, 70 where other markers for distal airway cell transition, such as SCGB3A2 and BPIFB1, have been demonstrated, which were not investigated in this study.

Response:

We provided a general overview of the different signatures observed in our data, but we could not validate every deregulated pathway or gene. We include the relevant tables detailing all differentially expressed genes and differentially methylated regions to enable and encourage the community to follow up on the data in subsequent studies.

As demonstrated above, we detect the co-occurrence of HT2-280/KRT5 staining on the protein level in the same cells in the alveoli of COPD patients. We would like to emphasize that alveolar epithelial cell identity in COPD lungs has not been investigated in detail on the protein or RNA level, and HT2-280/KRT5 co-expression/co-localization has not been directly tested in the studies mentioned by the reviewer since, among other reasons, the gene encoding HT2-280 has not been identified. Notably, a recent study (published after the submission of our manuscript) focusing on enriched epithelial cells from the distal lungs of COPD patients (PMID 35078977), identified an emphysema-specific AT2 subtype co-expressing the AT2 marker SFTPC and distal airway cell transition marker SCGB3A2, indicating that disease-specific AT2 populations with possible co-occurrence of AT2 and airway markers exist. In our dataset, SCGB3A2 was not deregulated (log₂ fold change=0.22, adj p-value= 0.47), as shown in **Table 6**, and the HT2-280/Krt5 positive cells were negative for SCGB3A2 in our IF staining (see above).

BPIFB1 is one of the antimicrobial peptides genes with an associated DMR and is significantly upregulated in COPD cells in our study (log₂ fold change=1.17, adj p-value=0.0016), as shown in the supplementary figure **Fig S4C** and here below for convenience.

Figure S4C Fold-change in gene expression of *BPIFB1* in AT2 cells in COPD (RNA-seq) and A549 cells treated with 0.5 μ M AZA (RT-qPCR) compared to control samples. Left, RNA-seq data from AT2 cells (no COPD, blue, $n=3$; COPD II-IV, green, $n=5$). Right, A549 treated with AZA (orange, $n=3$) compared to control DMSO-treated cells (grey, $n=3$). The group median is shown as a black bar.

- The small (and not evenly divided) sample size of both COPD and non-COPD specimens may lead to a higher risk for false positive results as adjustments for multiple testing typically rely on the number of comparisons, and small sample sizes may not provide enough data points to adequately control for this.

Response:

We acknowledge the problem of testing for multiple traits with relatively small numbers of samples. The availability of donor tissue, especially from non-COPD and COPD-I donors, was limited, and we applied very strict donor matching and quality control criteria for sample inclusion to avoid additional variability and confounding factors. The importance of strict quality control in selecting appropriate control samples was highlighted in our previous study (PMID: 33630765), where we demonstrated that approximately 50% of distal lung tissue from cancer patients with normal spirometry has pathological changes. Hence, we believe that the quality of the tissue was paramount to the reliability of the data. Strict quality control and sample matching for multiple parameters, including age, BMI, smoking status and smoking history (critical for DNA methylation studies), and cancer type (for background tissue), is a key strength of our approach, but it inevitably limited our sample size.

First, all samples were cryopreserved and then processed in parallel in groups of 1 non-COPD and 2-3 COPD samples. This process included tissue dissociation, FACS sorting, back sorting (always), and immunofluorescence staining (when enough material was available). Cell pellets were stored at -80°C until the entire cohort was ready for sequencing. This was done to limit the potential variation introduced by processing and sorting. RNA and DNA isolations were performed in parallel for all the sorted cell pellets, which were then sequenced as a single batch.

During data analysis, we applied stringent cutoffs for DMR detection to reduce the risk of false positives due to multiple comparisons and a small sample size. Specifically, we filtered for regions with at least 10% methylation difference and containing at least 3 CpGs. Additionally, we applied a non-parametric Wilcoxon test using average DMR methylation levels to remove potentially false-positive regions, as the t-statistic is not well suited for non-normally distributed values, as expected at very low/high (close to 0% / 100%) methylation levels. A significance level of 0.1 has been used. Therefore, we are confident that the rigorous analysis and strict criteria applied in this study allowed us to detect trustworthy DMRs that we could further functionally validate using epigenetic editing. All the details of the DMR analysis are provided in the methods section. To address this point and limitation, we have added the following paragraphs in the discussion section of the manuscript:

Full Revision

CHANGE IN THE MANUSCRIPT:

The strengths of our study include the use of purified human alveolar type 2 epithelial progenitor cells from a well-matched and carefully validated cohort of human samples, including mild and severe COPD patients, providing high relevance to human COPD.

However, we acknowledge several limitations of our study that warrant further investigation. First, the sample size was small. The use of strict quality criteria for donor selection limited the available samples, particularly for the ex-smoker control group. This resulted in an unequal distribution of COPD and control samples. This impacts the power of statistical analysis, particularly in the WGBS analysis, where millions of regions genome-wide are tested. Nevertheless, the clear negative correlation between promoter methylation and corresponding gene expression highlights the robustness of the DMR selection. Additionally, we were able to experimentally validate interferon-associated DMRs using epigenetic editing, highlighting the power of integrated epigenetic profiling in identifying disease-relevant regulators.

Minor suggestions for improvement

Introduction

- In general, refer to the actual experimental studies rather than review papers where appropriate.

Response:

We have now carefully checked all the references and amended them to refer to experimental studies when required.

- Clearly specify whether a study was conducted in mice or humans, as this distinction is crucial for understanding the relevance of the findings to COPD.

Response:

All our experiments were performed with human lung cells and tissues. No mouse samples were used. As suggested, we have now clearly stated that our study was performed using human tissue samples and cells in different parts of the manuscript, including the discussion, where we now explicitly highlight the strengths and limitations of our study.

CHANGES IN THE MANUSCRIPT:

*...we generated whole-genome DNA methylation and transcriptome maps of sorted **human** primary alveolar type 2 cells (AT2) at different disease stages.*

*However, the regulatory circuits that drive aberrant gene expression programs in **human** AT2 cells in COPD are poorly understood*

*Therefore, we set out to profile DNA methylation of **human** AT2 cells at single CpG-resolution across COPD stages.*

*...suggesting that aberrant epigenetic changes may drive COPD phenotypes in **human** AT2.*

*To identify genome-wide DNA methylation changes **associated with COPD** in purified **human** AT2 cells...*

Full Revision

The similarity of the methylation and gene expression profiles in the PCAs suggested that epigenetic and transcriptomic changes in human AT2 cells during COPD might be interrelated ...

In this work, we demonstrate that genome-wide DNA methylation changes occurring in human AT2 cells may drive COPD pathology by dysregulating key pathways that control inflammation, viral immunity and AT2 regeneration.

Using high-resolution epigenetic profiling, we uncovered widespread alterations of the DNA methylation landscape in human AT2 cells in COPD that were associated with global gene expression changes.

Currently, it is unclear how cigarette smoking leads to changes in DNA methylation patterns in human AT2

The strengths of our study include the use of purified human alveolar epithelial progenitor cells from a well-matched and carefully validated cohort of human samples, including mild and severe COPD patients, providing high relevance to human COPD.

Methods

- Line 473, here is meant 3 ex-smoker controls instead of smoker controls?

Response:

All donors (no COPD and COPD) used in our study are ex-smokers. Matching the samples with regard to smoking status and history is critical for epigenetic studies, as cigarette smoke profoundly affects DNA methylation genome-wide (PMID: 38199042, PMID: 27651444). This has now been clarified in the revised manuscript.

CHANGE IN THE MANUSCRIPT:

Of note, we included only ex-smokers in our profiling to avoid acute smoking-induced inflammation as a confounding factor (50).

Importantly, we matched the smoking status and smoking history of all donors, which is key in epigenetic studies, as cigarette smoking profoundly impacts the DNA methylation landscape of tissues (96).

In total, 3 ex-smoker controls (no COPD), 3 mild COPD donors ex-smokers (GOLD I, COPD I) and 5 moderate-to-severe COPD donors ex-smokers (GOLD II-IV, COPD II-IV) were profiled (Fig. 1A-C, Table 1)

Discussion

- A list of limitation should be added to the discussion. One is the use of the alveolar cell line A549, which produces mucus, a characteristic more commonly associated with bronchial epithelial cells. (ref 43)|530:

Response:

The profiling was performed using purified primary human alveolar epithelial progenitor cells. For technical reasons, A549 cells were only used for validation of the results using epigenetic editing. The A549 phenotype depends on the growth medium used, in our case, Ham's F-12 medium, which is recommended for long-term A549 culture and promotes multilamellar body

Full Revision

formation and differentiation toward an AT2-like phenotype (PMID: 27792742). We are developing epigenetic editing technology for use in primary lung cells; however, the approach currently relies on the high efficiency of transient transfections, which cannot yet be achieved with primary adult AT2 cells. We were positively surprised by how well the methylation data obtained from patient AT2s translated into mechanistic insights when using A549 cells, despite being a cancer cell line. This suggests that the fundamental mechanisms of epigenetic regulation of IRF9 and the IFN signaling pathway are conserved between A549 and primary AT2 cells.

- Another limitation to consider is that cells were isolated primarily from individuals with lung cancer, except for patients with COPD stage IV. In particular as COPD stage II and IV samples were taken together. And discuss the small and unevenly divided sample size

Response:

We thank the reviewer for bringing up this important point, which we carefully considered when designing our study. To match our samples across the cohort, all the no-COPD, COPD I, and two of the COPD II-IV samples were obtained from cancer resections. In addition to other characteristics, like age, BMI and smoking status, we also matched the donors by cancer type (all profiled donors had squamous cell carcinoma). We collected lung tissue as far away from the carcinoma as possible and sent representative pieces for histological analysis by an experienced lung pathologist to confirm the absence of visible tumours. In addition, to ensure that our data represents COPD-relevant signatures, we intentionally included samples from three COPD donors undergoing lung resections (without a cancer background) in the profiling.

Following the reviewer's suggestion, to investigate the potential impact of non-cancer samples on driving the observed differences, we carefully checked the PCAs for both DNA methylation and RNA-seq. We could not identify a clear separation of no-cancer COPD samples from the cancer COPD samples (or other cancer samples) in any examined PCs, indicating no confounding effect of cancer background in the samples. We observed that one sample contributing to PC2 is a non-cancer sample, but this was a rather sample-specific effect, as the other two non-cancer samples clustered together with the other severe COPD samples with a cancer background. Notably, in our DNA methylation data, we do not observe typical features of cancer methylomes, like global loss of DNA methylation or aberrant methylation of CpG islands (e.g., in tumour suppressor genes) (see **Fig 2A**), further suggesting that we do not "pick up" confounding cancer signatures in our data.

Following the comments from both reviewers, to clarify that point, we added the information about cancer and non-cancer samples to the PCA figures for DNA methylation (**new Fig. 2B**) and RNA-seq (**new Fig. 3A**) data in the revised manuscript, as shown below

CHANGE IN THE MANUSCRIPT:

COPD samples from donors with a cancer background clustered together with the COPD samples from lung resections, confirming that we detected COPD-relevant signatures (Fig. 2B).

Fig.2B Principal component analysis (PCA) of methylation levels at CpG sites with > 4-fold coverage in all samples. COPD I and COPD II-IV samples are represented in light and dark green triangles, respectively, and no COPD samples as blue circles. COPD samples without a cancer background are displayed with a black contour. The percentage indicates the proportion of variance explained by each component.

Unsupervised principal component analysis (PCA) on the top 500 variable genes revealed a clear influence of the COPD phenotype in separating no COPD and COPD II-IV samples, as previously observed with the DNA methylation analysis, irrespective of the cancer background of COPD samples (Fig.3A, Fig. S2B).

Figure 3A: RNA-seq/gene expression

Figure S2B: RNA-seq/gene expression

Principal component analysis (PCA) of 500 most variable genes in RNA-seq analysis. PCA 1 and 2 are shown in Fig.3A, PCA 1 and 4 in Fig.S2B. COPD I and COPD II-IV samples are represented in light and dark green triangles, respectively, and no COPD samples as blue circles. COPD samples without a cancer background are displayed with a black contour. The percentage indicates the proportion of variance explained by each component.

Response:

We thank the reviewer for suggestions on how to improve the discussion of our manuscript. We have now added a strength/limitation section to our discussion and included the points suggested by both reviewers.

CHANGE IN THE MANUSCRIPT:

The strengths of our study include the use of purified human alveolar epithelial progenitor cells from a well-matched and carefully validated cohort of human samples, including mild and severe COPD patients, providing high relevance to human COPD. Importantly, we matched the smoking status and smoking history of all donors, which is key in epigenetic studies, as cigarette

Full Revision

smoking profoundly impacts the DNA methylation landscape of tissues (96). With the first genome-wide high-resolution methylation profiles of isolated cells across COPD stages, we offer novel insights into the epigenetic regulation of gene expression in epithelial progenitor cells in COPD, expanding our understanding of how alterations in regulatory regions and specific genes could contribute to disease development. We identified IRF9 as a key IFN transcription factor regulated by DNA methylation. Notably, by targeting IRF9 through epigenetic modifications, we modulated the activity of the IFN pathway, which plays a crucial role in the immune response and lung tissue regeneration. Epigenetic editing techniques could offer a novel therapeutic strategy for COPD by downregulating IFN pathway activation and promoting the regeneration of epithelial progenitor cells in the lungs. Further preclinical and clinical studies are needed to validate the efficacy and safety of epigenetic editing approaches in COPD treatment (33).

However, we acknowledge several limitations to our study that warrant further investigation. First is the small sample size and replication difficulty due to the lack of available data, common challenges for studies working with sparse human material and hard-to-purify cell populations. The use of strict quality criteria in donor selection limited the available samples, especially for the ex-smoker control group, leading to an unequal distribution of COPD and control samples. Overall, this impacts the power of statistical analysis, especially in the WGBS analysis, where millions of regions genome-wide are tested. Nevertheless, the clear negative correlation of promoter methylation to the corresponding gene expression highlights the robustness of the DMR selection. Furthermore, we could experimentally validate interferon-associated DMRs using epigenetic editing, highlighting the power of integrated epigenetic profiling for the discovery of disease-relevant regulators.

Overall, we detected a higher number of correlated DMR-DEG associations using our simple promoter-proximal linkage compared to the GeneHancer approach. Assigning enhancers to their target genes with high confidence is a complex and challenging task. Enhancers are often located far from the genes they regulate and can interact with their target genes through three-dimensional chromatin loops. Furthermore, enhancers can operate in a highly context-dependent manner, with the same enhancer regulating different genes depending on the cell type, developmental stage, or environmental signals. Determining which enhancer is active under specific conditions remains a hurdle in the field, especially since the AT2-specific chromatin profiles of enhancer marks are not yet available.

In addition, while WGBS provides unprecedented resolution and high coverage of the DNA methylation sites across the genome, it does not allow distinguishing 5-methylcytosine from 5-hydroxymethylcytosine. Therefore, we cannot exclude that some methylated sites we detected are 5-hydroxymethylated. However, as 5-hydroxymethylcytosine is present at very low levels in the lung tissue (97), its effect is likely marginal.

Finally, despite careful removal of airways from distal lung tissue using a dissecting microscope, we cannot exclude the presence of some terminal/respiratory bronchiole cells in our FACS-isolated EpCAM^{pos}/PDPN^{low} population. Recent scRNA-seq studies provided an unprecedented resolution and identified several epithelial subpopulations and transitional cells residing in the terminal/respiratory bronchioles and alveoli, including respiratory airway secretory cells (93), terminal airway-enriched secretory cells (28), terminal bronchiole-specific alveolar type-0 (AT0)

Full Revision

(70), and emphysema-specific AT2 cells (74). These cells may contribute to alveolar repair in healthy and COPD lungs; however, with our bulk DNA methylation and RNA-seq study, we are unable to resolve all these subpopulations. Future development of single-cell methylation and non-reference-based algorithms for DNA methylation deconvolution will enable deeper epigenetic phenotyping of specific AT2 and bronchiolar cell subsets.

References

- Check references. For instance, there is no reference in the text to ref 43.
- Align format of references

Response:

We thank the reviewer for spotting this inconsistency. We have carefully checked and aligned the format of all references. The (old) reference 43 is now mentioned in the discussion part.

Reviewer #1 (Significance (Required)):

The strength of this study lies in its focus on the molecular mechanisms underlying the impaired regeneration of epithelial progenitor cells in COPD. The discovery of IRF9, which regulates IFN signaling and is prominently upregulated in COPD, together with the convincing validation of the epigenetic control of the IFN pathway by targeted DNA demethylation of the IRF9 gene, adds significant value to the COPD research field.

Main limitations of the study are the relatively small sample size of both COPD and non-COPD specimens and the claim that the sorted EpCAMpos/PDPNlow cells primarily consisted of AT2 cells.

- Describe the nature and significance of the advance (e.g. conceptual, technical, clinical) for the field.

The nature and significance of the advance in epigenetic editing of IRF9 in COPD can be described as both conceptual and potentially clinical:

Conceptual Advance: The epigenetic editing of IRF9 enhances our understanding of the molecular mechanisms underlying COPD pathogenesis. By targeting IRF9 through epigenetic modifications, researchers were able to modulate the activity of the IFN pathway, which plays a crucial role in the immune response and lung tissue regeneration. This approach offers insights into the epigenetic regulation of gene expression in epithelial progenitor cells in COPD and expands our understanding of how alterations in specific gene methylation could contribute to disease progression.

Clinical Significance: The potential clinical significance of epigenetic editing of IRF9 lies in its implications for COPD therapy. If successful, epigenetic editing techniques could offer a novel therapeutic strategy for COPD by downregulating IFN pathway activation and promoting regeneration of epithelial progenitor cells in the lungs. Obviously, further preclinical and clinical studies are needed to validate the efficacy and safety of epigenetic editing approaches in COPD treatment.

Response:

We thank the reviewer for recognising the importance of our study, its conceptual advance and

Full Revision

potential clinical significance. We are pleased to see that the reviewer highlights the promise of epigenetic editing in both furthering our basic understanding of molecular mechanisms of chronic diseases and its future potential as a therapeutic strategy.

- Place the work in the context of the existing literature (provide references, where appropriate).

Few experimental papers have been published on epigenetic editing in lung diseases, with limited research available beyond the study referenced in citation 43. Song J, Cano-Rodriguez D, Winkle M, Gjaltema RA, Goubert D, Jurkowski TP, Heijink IH, Rots MG, Hylkema MN. Targeted epigenetic editing of SPDEF reduces mucus production in lung epithelial cells. *Am J Physiol Lung Cell Mol Physiol*. 2017 Mar 1;312(3):L334-L347. doi: 10.1152/ajplung.00059.2016. Epub 2016 Dec 23. PMID: 28011616.

Response:

We thank the reviewer for recognising the uniqueness and novelty of our study and the lack of research on the functional understanding of DNA methylation in the context of lung and lung diseases.

- State what audience might be interested in and influenced by the reported findings.

This study is of broad interest to researchers investigating the pathogenesis and treatment of COPD.

- Define your field of expertise with a few keywords to help the authors contextualize your point of view.

Expertise in: Lung pathology, Immunology, COPD, Epigenetics

- Indicate if there are any parts of the paper that you do not have sufficient expertise to evaluate.

Less expertise in: Epigenetic Editing

Reviewer #2 (Evidence, reproducibility and clarity (Required)):

Summary:

This study aim to understand the molecular mechanisms underlying dysfunction in AT2 cells in COPD, by profiling bulk genome wide DNA methylation using Tagmentation-based whole-genome bisulfite sequencing (T-WGBS) and RNA sequencing in selectively sorted primary AT2 cells. The study stands out in it's sequencing breadth and use of an incredibly difficult cell population, and has the potential to add substantially to our mechanistic understanding of epigenetic contributions to COPD. A further highlight is the concluding aspect of the study where the authors undertook targeted modification of specific CpG methylation, provided direct, site-specific evidence for transcriptional regulation by CpG methylation.

Full Revision

Response:

We thank the reviewer for recognizing the conceptual and methodological advance of our study and for noting the value of our functional mechanistic approach.

Major comments:

The authors clearly show that there is DNA methylation alteration in AT2 cells from COPD individuals that links functional to gene expression at some level. However, I think the statement "to identify genome-wide changes associated with COPD development and progression..." and similar other references to disease development understanding is not accurate given the DNA methylation primary comparison is between control and moderate to severe COPD, with no temporal detail or evidence that they drive progression rather than are a result of COPD development. The paragraph starting on line 186 where this is addressed to some extent is quite vague and doesn't really provide confidence that DNAm dysregulation occurs at an early stage in this context. This can be addressed by changing the focus/style of the text.

Response:

Thank you for raising this point. We agree with the reviewer that our cross-sectional study describes the association of methylation changes with either COPD I or more established disease (COPD II-IV) and that the observed changes may be either the driver or a result of COPD development. This has been clarified in the revised manuscript, and we removed the statements about disease initiation and progression. This is an important point; hence, we added an extra line to the discussion to make that clear.

CHANGE IN THE MANUSCRIPT:

*Therefore, we set out to profile DNA methylation of human AT2 cells at single CpG-resolution across COPD stages **to identify epigenetic changes associated with disease** and combine this with RNA-seq expression profiles.*

*To identify epigenetic changes **associated with COPD**, we collected lung tissue from patients with different stages of COPD,*

*....**to identify methylation changes associated with mild disease**, we included TWGBS data from AT2 isolated from COPD I patients (n=3) in the analysis.*

***Currently, we do not know whether the identified DNA methylation changes are the cause or the consequence of the disease process** and not much is known about the correlation of DNA methylation with disease severity.*

*However, **our study is cross-sectional**, our cohort included only 3 COPD I donors, and we did not have any follow-up data on the patients, so future large-scale profiling of mild disease (or even pre-COPD cohorts) in an extended patient cohort will be crucial for a better understanding of early disease and its progression trajectories.*

Results comments and suggestions:

For the integrated analysis, there is a focus on DMRs in promoters with very little analysis on other regions. The paragraph starting on line 317 describes some analysis on enhancers but is very brief, doesn't include information on how many/which DMRs were included, making it hard

Full Revision

to interpret the impact of the 147 DMRs and 93 genes identified - is this nearly all DMRs and genes analysed or very few? A comparison to the promoter analysis would be of interest. Especially as the targeted region followed up with lovely functional assessment in the last sections is a gene body DMR, not a promoter DMR.

Response:

We thank the reviewer for pointing out the importance of changes in enhancers. We agree that extending the enhancer analysis is very interesting. However, assigning enhancers to their target genes with high confidence is a complex and challenging task. Enhancers are often located far from the gene they regulate, sometimes spanning hundreds of kilobases. They can interact with their target genes through three-dimensional chromatin loops, potentially bypassing nearby genes to activate more distant ones, making it difficult to confidently link specific enhancers to their target genes. Furthermore, enhancers can operate in a highly context-dependent manner. The same enhancer can regulate different genes depending on the cell type, developmental stage, or environmental signals. Another challenge is that enhancers often work in clusters or "enhancer landscapes," where multiple enhancers contribute to the regulation of a single gene. Disentangling the contribution of individual enhancers within such clusters and determining which enhancer is active under specific conditions remains an ongoing hurdle in the field, especially since the AT2-specific chromatin profiles of enhancer marks are not yet available.

One approach we tried to account for more distal regulatory regions was to assign DMRs to the nearest gene with a maximum distance of up to 100 kb using GREAT (Genomic Regions Enrichment of Annotations Tool) and simultaneously perform gene enrichment analysis of the associated genes. **The old Figure S1C (now S1D)** shows the top 10 enriched terms of either hyper- or hypomethylated DMRs, and **Table 4** shows the full list of enriched terms. However, in this analysis, we did not integrate the results of the RNA-seq analysis. To demonstrate that we can correlate methylation with gene expression associations in this analysis, we then took a closer look at the WNT/b-catenin pathway, which contains 147 DMRs associated with 93 genes from the respective pathway (**old Figure S3D, now S3G**). Here, we showed that distal DMRs up to 100 kb away from the TSS show a high correlation with gene expression. We are including the two figures below for convenience:

Figure S1 D

Figure S3 G

Left panels, functional annotation of genes located next to hypermethylated (top) and hypomethylated (bottom) DMRs using GREAT. Hits were sorted according to the binominal adjusted p -value and the top 10 hits are shown. The adjusted p -value is indicated by the color code and the number of DMR associated genes is indicated by the node size. Right panel, scatter plot showing distal DMR-DEG pairs associated with Wnt-signaling. Pairs were extracted from GREAT analysis (hypermethylated, DMR-DEG distance < 100 kb; see Fig. S1C). Y-axis represents the \log_2 fold-change expression in severe COPD, and the x-axis denotes the Spearman correlation of the DEG-DMR pair. Node size indicates the distance of the DMR to the TSS. DEGs are highlighted in red.

Following the reviewer's suggestion, we have now extended the enhancer analysis using the GeneHancer database, the most comprehensive, integrated resource of enhancer/promoter-gene associations. We used the GeneHancer version 5.14, which annotates 392,372 regulatory genomic elements (GeneHancer element) on the hg19 reference genome. Of the 25,028 DMRs, 18,289 DMRs (73% of all DMRs) coincided with at least one GeneHancer element, resulting in 19,661 DMR-GeneHancer associations. Next, we extracted the GeneHancer elements associated with protein-coding or long-non-coding RNAs genes, which left us with 2,144 DMR-GeneHancer associations. Next, we used only high-scoring gene GeneHancer associations ("Elite"), leaving 1,485 DMR-GeneHancer associations. Of those, we selected the GeneHancer elements, which are linked to genes differentially expressed in our RNA-seq analysis resulting in a final table of 376 DMR-GeneHancer associations (**Table 9 DMR_DEG_GeneHancer, Tab 2**). Similar to the promoter-proximal analysis, we analysed the correlation of expression and methylation changes of the DMR-GeneHancer associations, demonstrating a high number of negatively and positively correlated events (**Fig.S3D**). Finally, we performed the gene enrichment analysis for positively and negatively correlating genes. We detected significant GO term enrichments only for negatively correlating genes (**Fig.S3E and Table 10_Enrichment_results, Tab2**).

CHANGE IN THE MANUSCRIPT

Full Revision

To harness the full resolution of our whole-genome DNA methylation data, we extended the analysis beyond promoter-proximal regions and assessed how epigenetic changes in distal regulatory regions (enhancers) may relate to transcriptional differences in COPD. As the assignment of enhancer elements to the corresponding genes is challenging, we tried two different approaches. First, we used the GeneHancer database (72) to link DMRs to regulatory genomic elements (GeneHancer element). Of the 25,028 DMRs, 18,289 DMRs (73%) coincided with at least one GeneHancer element. Of those 2,144 DMR-GeneHancer associations were linked either to protein-coding or lncRNA genes. Next, we filtered for high-scoring gene GeneHancer associations (“Elite”), leaving 1,485 DMR-GeneHancer Elite associations. Of those, we selected the GeneHancer elements, which are linked to genes differentially expressed in our RNA-seq analysis, resulting in 376 DMR-GeneHancer associations (Table 9). Similar to the promoter-proximal analysis, we assessed the correlation of expression and methylation changes of the DMR-GeneHancer associations, demonstrating a high proportion of negatively and positively correlated events (Fig. S3E). Finally, we performed gene enrichment analysis for positively and negatively correlated genes. We detected significant GO term enrichments for negatively correlating genes only (Fig. S3F and Table 10), with the most pronounced term “regulation of tumor necrosis factor”. In an alternative approach, we linked proximal and distal (within 100 kb from TSS) DMRs to the next gene using GREAT (57) (Fig S1C, Table 4) and calculated Spearman correlation between DMRs and associated DEGs. 147 DMRs were associated with high correlation rates with 93 genes from the WNT/ β -catenin pathway (Fig. S3G), suggesting that DNA methylation may also drive the expression of genes of the WNT/ β -catenin family.

Figure S3E and F: E. Spearman correlation between gene expression and DMR methylation of DMRs assigned to gene regulatory elements using the GeneHancer database. F. GO-Term over-representation analysis of DEGs negatively correlated to DMRs in gene regulatory elements. The adjusted p-value is indicated by the color code and the percentage number of associated DEGs is indicated by the node size.

(Methods) For enhancer analysis, the GeneHancer database version 5.14, which annotates 392,372 regulatory genomic elements (GeneHancer element) on the hg19 reference genome, was used (72). Of the 25,028 DMRs 18,289 DMRs coincided with at least one GeneHancer

Full Revision

element, resulting in 19,661 DMR-GeneHancer associations. Next, the GeneHancer elements were filtered for association with protein-coding or long-non-coding RNAs genes and high-scoring gene GeneHancer associations ("Elite"), leaving 1,485 DMR-GeneHancer associations. Of those, the GeneHancer elements were selected, which are linked to differentially expressed genes in COPD resulting in a final table of 376 DMR-GeneHancer associations. Similar to the promoter-proximal analysis, the Spearman correlation of expression and methylation changes of the DMR-GeneHancer associations was assessed. GO gene enrichment analysis for positively and negatively correlating genes was done using Metascape (111).

A comparison to the promoter analysis would be of interest.

Response:

We detected more highly correlated ($|\text{correlation coefficient}| > 0.5$) DMR-DEG associations using our simple promoter proximal linkage ($n=643$) in comparison with the GeneHancer approach comprising annotated enhancer elements ($n=327/2,144$). Gene enrichment results pointed to the interferon pathway, which we could confirm using epigenetic editing. This pathway was not present in the GeneHancer analysis, indicating that regulation of the IFN pathway may be controlled by proximal elements.

CHANGE IN THE MANUSCRIPT:

Overall, we detected a higher number of correlated DMR-DEG associations using our simple promoter-proximal linkage compared to the GeneHancer approach. Assigning enhancers to their target genes with high confidence is a complex and challenging task. Enhancers are often located far from the genes they regulate and can interact with their target genes through three-dimensional chromatin loops. Furthermore, enhancers can operate in a highly context-dependent manner, with the same enhancer regulating different genes depending on the cell type, developmental stage, or environmental signals. Determining which enhancer is active under specific conditions remains a hurdle in the field, especially since the AT2-specific chromatin profiles of enhancer marks are not yet available.

Especially as the targeted region followed up with lovely functional assessment in the last sections is a gene body DMR, not a promoter DMR.

Response:

We thank the reviewer for bringing up that point. To clarify, we defined the promoter regions for the analysis as regions located ± 6 kb (upstream and downstream) from the transcriptional start site (TSS). Since the term "promoter" often refers to the region upstream of the transcriptional start site, its use may have been misleading. For clarity, we changed the text correspondingly to **promoter proximal methylation** and explained in the methods how the regions for analysis were defined.

CHANGE IN THE MANUSCRIPT:

"DMR association per gene promoter" was changed to "Gene promoter proximal DMRs"

Fig. S3B: "DMR in promoter" was changed to "promoter proximal DMR(s)"

Full Revision

“by DNA methylation changes in promoters” was changed to “by DNA methylation changes in promoter proximity”

“regulated by promoter methylation” was changed to “regulated by promoter-proximal methylation”

“analysis of the promoter DMRs” was changed to “analysis of the promoter-proximal DMRs”

“between promoter methylation” was changed to “between promoter proximal methylation”

Cytoscape was used to analyse negatively or positively correlated DMR DEG pairs. ClueGO (v2.5.6) analysis was conducted using all DEG associated with a promoter proximal DMR (+/- 6 kb from TSS) and the Spearman correlation coefficient < -0.5 or > 0.5 (112).

- Lines 299-301 - I'm not sure the graph in Fig S3A support the conclusion that there was a preferential negative relationship between DNAm and gene expression. Looks like there are a substantial number of cases where a positive relationship is observed and this needs to be acknowledged.

Response:

In this part, we refer to **Fig S3C**. In the left panel, downregulated genes clearly show higher counts for the hypermethylated DMRs, whereas the hypomethylated DMRs are enriched at upregulated genes (right panel), indicating a preference for negative correlation: lower methylation, higher gene expression. If there were no preference, we would expect a 50:50 ratio of hypo- and hypermethylated DMRs, and we observed a 77:23 ratio. Nevertheless, we agree that there is a substantial number of cases ($n=151$) with a high positive correlation, which we now highlight in the text. For clarity, we also modified the figure legend to indicate that a stacked histogram is represented in the panel.

CHANGE IN THE MANUSCRIPT:

L303: Interestingly, 23.5% of the identified DMR DEG pairs ($n=151$) showed a positive correlation between gene expression and DNA methylation.

Figure legend in **Fig. S3C** was changed to: **C** Stacked histogram showing location of hyper- and hypomethylated DMRs relative to the TSS of DEGs in downregulated (left) and upregulated (right) genes.

- Line 307 - what are the "analysed DEGs"? Are they the methylation associated genes?

Response:

Full Revision

Those are the DEGs we identified in RNA-seq analysis. To clarify, we changed the text to “identified DEGs”.

CHANGE IN THE MANUSCRIPT:

“analysed DEGs” was changed to “identified DEGs”

- Line 307-309 - "Among the analyzed DEGs, 76.5% (492) displayed a negative correlation (16.8% of the total DEGs), indicating a possible direct regulation by DNA methylation, while 23.5% (151) showed a positive correlation between gene expression and DNA methylation" - are the authors suggesting the positive correlation doesn't indicate direct regulation?

Response:

Thank you for highlighting this point. We did not intend to suggest that negative correlation indicates direct regulation, while positive correlation suggests a lack thereof. To clarify that point, we have reformulated this sentence.

CHANGE IN THE MANUSCRIPT:

Among the identified DEGs, 76.5% (n=492) displayed a negative correlation (16.8% of the total DEGs), consistent with a repressive role of promoter DNA methylation. Interestingly, 23.5% of the identified DEG (n=151) showed a positive correlation between gene expression and DNA methylation.

- Line 313 - why did the authors focus on only negatively correlated genes to identify their top dysregulated pathway of IFN signalling? Why not do pathway analysis on the DNAm associated genes separately to identify DNAm associated pathways?

Response:

We have also performed a pathway enrichment analysis using the positively correlated genes but did not identify any significantly enriched pathways/process/terms. When we examined the top hit of the gene set enrichment analysis, the interferon signaling pathway, we observed only negatively correlated DMR gene associations (**Fig. 5B**). Therefore, we decided to use only the negatively correlated DMRs, as using all correlated genes would give a higher background and dilute our results.

CHANGE IN THE MANUSCRIPT:

Cytoscape was used to analyse negatively or positively correlated DMR DEG pairs. ClueGO (v2.5.6) analysis was conducted using all DEG associated with a promoter proximal DMR (+/- 6 kb from TSS) and the Spearman correlation coefficient < -0.5 or > 0.5 (113).

- A comparison of the gene expression data with previous data in AT2 cell/single cell data would strengthen the gene expression section.

Response:

We compared our gene expression signatures with the study of Fujino *et al.*, who profiled sorted AT2 cells (EpCAM^{high}PDPN^{low}) from COPD/controls using expression arrays (PMID: 23117565). Consistent with our study, the authors also observed the upregulation of interferon signalling

Full Revision

(among other pathways) in COPD AT2s. However, no raw data was available in the published manuscript for a more in-depth analysis.

Several recent scRNA-seq studies identified transcriptional signatures of COPD and control cells (e.g., PMIDs: 36108172, 35078977, 36796082, 39147413). However, most studies did not match the smoking status of the control and COPD donors and looked at the whole lung tissue, with limited power to detect gene expression changes in distal alveolar cells. It is difficult to directly compare our data to the gene expression data from non-smokers vs COPD patients, as cigarette smoking profoundly remodels the epigenome and transcriptional signatures of cells. In addition, differences in technologies and depth of sequencing make such comparisons challenging. However, one study (PMID: 36108172) performed scRNA-seq analysis on 3 non-smokers, 4 ex-smokers and 7 COPD ex-smoker lungs. Despite relatively limited coverage of epithelial cells in the dataset (<10K, 5K for COPD), in agreement with our bulk study, the authors observed the enrichment of IFN α/β signatures in COPD in the inflammatory AT2 cluster (iAT2).

We also compared the main AT2 IFN signature identified in the integration of our DNA methylation in promoter-proximal regions and RNA-seq with a recent study (published after the submission of our manuscript, PMID: 39147413) that profiled EpCAM^{POS} cells from COPD and control lungs (non-smokers) using scRNA-seq. We observed an upregulation of our IFN signature genes in AT2 in COPD (specifically in AT2-c and rbAT2 subsets), suggesting that similar signatures were observed in this dataset as well. However, ex-smokers were not included in this study, making direct comparisons difficult. **We have now included the panels shown below as Figure S4E and S4F:**

Figure S4

Figure S4E and F: Expression values for the indicated genes of the IFN pathway from an external scRNA-seq dataset of AT2 cells from COPD patients and healthy controls (74). Y-axis shows log-normalized gene expression levels. **F.** Combined gene set score of the genes shown in (E) in different subsets of AT2 cells from (74). The IFN signature genes were identified in our integrative analysis of TWGBS and RNA-seq in sorted AT2 cells.

CHANGES IN THE MANUSCRIPT:

Full Revision

However, 5-AZA is a global demethylating agent, and the observed effects may not be direct. To validate the epigenetic regulation of central AT2 pathways further, we took advantage of locus-specific epigenetic editing technology (73). We focused on the IFN pathway because it was the most significantly enriched Gene Ontology (GO) term in our integrative analysis of TWGBS and RNA-seq data. Several IFN pathway members had associated hypomethylated DMRs within promoter-proximal regions and concomitant increased gene expression (Fig. 4C and Fig.S2C). Additionally, we confirmed the elevated expression of IFN-related genes with associated DMRs identified in our study in AT2 cells and AT2 cell subclusters from a recently published scRNA-seq cohort (74) (Fig. S4E-F).

(Methods) Validation of IFN gene upregulation in a published scRNA-seq dataset

scRNA-seq data from (74), generously provided by M. Köningshoff, were processed using the default Seurat workflow (117). Expression of IFN-related genes was extracted and plotted as log-normalised gene expression levels in AT2 cells from control and COPD donors. Seurat's AddModuleScore() function was used to compute a gene set score for a custom IFN program using the genes listed in Fig. S4E and to analyse the IFN gene set scores in AT2 cell subclusters identified in (74). Briefly, average gene expression scores were computed for the gene set of interest, and the expression of control features (randomly selected) was subtracted as described in (118).

Fig. S4 E and F. E. Expression values for the indicated genes of the IFN pathway from an external scRNA-seq dataset of AT2 cells from COPD patients and healthy controls (74). Y-axis shows log-normalized gene expression levels. **F.** Combined gene set score of the genes shown in (E) in different subsets of AT2 cells from (74). The IFN signature genes were identified in our integrative analysis of TWGBS and RNA-seq in sorted AT2 cells.

- The paragraph starting on line 173 feels a little redundant when we know there is RNA available to test if the differential DNAm links to altered gene expression - this selected of example regions/genes would be better placed after the gene expression has been reported, at which point you could say whether the linked genes displayed altered transcription.

Response:

The current structure (with DNA methylation, followed by RNA-seq and integration) is intentional and serves several important purposes. As this is the first genome-wide high-resolution COPD DNA methylation study of AT2, we aimed to describe the methylation landscape independently of gene expression (noting the limitation of current understanding of how DNA methylation regulates expression). This early focus on DMRs lays clear groundwork by highlighting potential regulatory elements and pathways that could be disrupted, independent of or even before corroborative transcriptional data. Additionally, positioning these examples early in the narrative helps to frame subsequent gene expression analyses. Once RNA data are introduced later, the reader can directly compare the methylation patterns with transcriptional outcomes, thereby enhancing the overall story. In other words, by first showcasing disease-relevant methylation changes, we underscore a hypothesis that these epigenetic modifications are functionally meaningful. The later integration of gene expression data then serves as a confirmatory or complementary layer, rather than the sole basis for inferring biological significance. This is important as we still do not fully understand the function of DNA methylation outside promoters,

Full Revision

and its role is also important for splicing, 3D genome organisation, non-coding RNA regulation, enhancer regulation, etc.

- Similarly, the TF enrichment analysis is great but maybe would have added value to be done on DNA regions later shown to be linked to differential expression - was there different enrichment at DNA regions that are vs are not associated with altered expression? And could you test in vitro whether changing methylation of DNA (maybe a blunt too like 5-aza would be ok) alters TF binding (cut+run/ChIP?). Furthermore, it would be interesting to understand the TF sensitivity analysis within the context of positive versus negative DNA methylation:gene expression correlations.

Response:

As suggested by the reviewer, we now performed the TF enrichment analysis using the DMRs with a high correlation ($|\text{correlation coefficient}| > 0.5$) between methylation and expression (**Figure S3D**) and expanded the method section to include TF analysis. We observed ETS domain motifs enriched at hypomethylated regions. They prefer unmethylated DNA (MethylMinus) and are therefore expected to bind with higher affinity to the respective DMRs in COPD. We agree with the reviewer that further verifying altered TF binding using cut&run or ChIP assays would be very interesting, but it is out of the scope of this manuscript. Such analysis is technically very challenging to perform with low numbers of primary AT2 cells and will be the focus of our follow-up mechanistic studies.

CHANGE IN THE MANUSCRIPT:

Additionally, motif analysis of DMRs that were highly correlated ($|\text{Spearman correlation coefficient}| > 0.5$) with DEGs revealed a prominent enrichment of the cognate motif for ETS family transcription factors, such as ELF5, SPIB, ELF1 and ELF2 at hypomethylated DMRs (**Fig. S3D**). Interestingly, SPIB was shown to facilitate the recruitment of IRF7, activating interferon signaling (71), and our WGBS data uncovers SPIB motifs at hypomethylated DMRs, which aligns with its binding preferences at unmethylated DNA (methyl minus, **Fig. S3D**).

D

Figure S3D: Enrichment of methylation-sensitive binding motifs at hypo- (right) and hypermethylated (left) DMRs, using DMRs with a high correlation ($|\text{Spearman correlation coefficient}| > 0.5$) between methylation and gene expression. Methylation-sensitive motifs were derived from Yin et al (64). Transcription factors, whose binding affinity is impaired upon methylation of their DNA binding motif, are shown in red (Methyl Minus), and transcription factors, whose binding affinity upon

Full Revision

CpG methylation is increased, are shown in blue (Methyl Plus).

(Methods) To obtain information about methylation-dependent binding for transcription factor motifs which are enriched at DMRs, the results of a recent SELEX study (64) were integrated into the analysis. They categorised transcription factors based on the binding affinity of their corresponding DNA motif to methylated or unmethylated motifs. Those whose affinity was impaired by methylation were categorised as MethylMinus, while those whose affinity increased were categorised as MethylPlus. A motif database of 1,787 binding motifs with associated methylation dependency was constructed. The log odds detection threshold was calculated for the HOMER motif search as follows. Bases with a probability > 0.7 got a score of $\log(\text{base probability}/0.25)$; otherwise, the score was set to 0. The final threshold was calculated as the sum of the scores of all bases in the motif. Motif enrichment analysis was carried out against a sampled background of 50,000 random regions with matching GC content using the findMotifsGenome.pl script of the HOMER software suite, omitting CG correction and setting the generated SELEX motifs as the motif database.

Methods:

- The authors should include more detail of the TWGBS rather than directing the reader to a previous publication. Also DNA concentration post bisulfite conversion would be a useful metric to provide.

Response:

Following the suggestion, we have now expanded the details of TWGBS in the methods part of the manuscript. Due to limited space, we did not include a detailed protocol but instead referred to a published step-by-step protocol (55). Of note, we do not measure DNA concentration post-bisulfite conversion but consistently use the starting input of 30 ng of genomic DNA across all samples.

CHANGE IN THE MANUSCRIPT:

(Methods): 15 pg of unmethylated DNA phage lambda was spiked in as a control for bisulfite conversion. Tagmentation was performed in TAPS buffer using an in-house purified Tn5 assembled with load adapter oligos (55) at 55 °C for 8 min. Tagmentation was followed by purification using AMPure beads, oligo replacement and gap repair as described (55). Bisulfite treatment was performed using EZ DNA Methylation kit (Zymo) following the manufacturer's protocol.

The T-WGBS library preparations were performed for all donors in parallel and sequenced in a single batch to minimize batch effects and technical variability.

- Differential DNA methylation analysis: It is stated that DNA regions had to contain 3 CpG sites but was this within a defined DNA size range?

Response:

Full Revision

The maximum distance between individual CpGs within DMR was set to 300 bp. To clarify, we added that information to the methods part.

CHANGE IN THE MANUSCRIPT:

“regions with at least 10% methylation difference and containing at least 3 CpGs with a maximum distance of 300 bp between them.”

- Refence genome only provided for RNAseq not TWGBS?

Response:

We used hg19 as the reference genome. The information on the reference genome for DNA methylation analysis was provided in the methods L574 (original manuscript_: “The reads were aligned to the transformed strands of the hg19 reference genome using BWA MEM”)

- The tables do not appear in the PDF and I struggled to tally to the "Dataset" files provided if that is what they were referring to?

Response:

Full tables (uploaded as Datasets in the manuscript central due to their size) were uploaded together with the manuscript files. They are quite large and will not convert to pdf, so they may not have been included in the merged pdf file. We assume that they should be available to the reviewers with the other files and will clarify that with the editorial staff in the resubmission cover letter.

- For the gene expression analysis, can it be made clearer that a full analysis was done on COPD I samples. It is a little confusing to the reader as this was not done for DNAm so might be assumed the same targeted analysis on only genes found to be differentially expressed between control and COPD II-IV, but that cannot be the case as an overlap of COPD1 vs COPD II-IV genes if provided. For this overlap, do genes show the same effect direction?

Response:

To clarify, for the RNA-seq analysis, we performed DEG analysis for no-COPD versus COPD II-IV, as well as no-COPD versus COPD I. We then took all differentially expressed genes (presented in the Venn diagram) and plotted them for all samples as a heatmap. To split the genes into groups displaying similar effect directions, we applied a clustering approach and identified 3 main signatures. Cluster 3 primarily comprises genes unique to COPD I samples, which are associated with the adaptive immune system and hemostasis (**Fig. 4E**). In the other two clusters, we mainly observe a transitioning pattern from control to severe COPD samples, correlating with the FEV₁ values of the patients. This has now been clarified in the manuscript.

- Replication is difficult on these studies as the samples are so difficult to come by. Also limited by sample size for the same reason. It doesn't mean the study is not worth doing and the data are still valuable. However, it may be pertinent to include technical validation of a few regions of interest, acknowledge the limitation (along side strengths) in the discussion, and perhaps provide actual p value rather than blanket < p 0.1, seems very lenient but may all be super significant (this may already be in the tables I wasn't able to find).

Response:

Full Revision

We thank the reviewer for acknowledging the replication challenges for studies working with sparse human material and hard-to-purify cell populations. Following the reviewer's suggestion, we have now included a strengths and limitations section in the discussion where we summarised the points highlighted by both reviewers.

Regarding technical validation, we would like to note that the whole genome bisulfite sequencing (WGBS) technology, as well as the tagmentation-based WGBS (T-WGBS), have been validated in the past few years in several publications (e.g., PMID: 24071908) and shown to yield reliable DNA methylation quantification in comparison to other technologies (PMID: 27347756). For us, technical validation using alternative methods (e.g. bisulfite sequencing or pyrosequencing) is difficult as it requires significantly more input DNA than the low-input T-WGBS we have performed and obtaining sufficient amounts of material from primary human AT2 cells (especially from severe COPD) is not possible with the size of tissue we can access. However, while establishing the T-WGBS for this project, we initially validated our approach using Mass Array, a sequencing-independent method. For this, we performed T-WGBS on the commercially available smoker and COPD lung fibroblasts and selected 9 regions with different methylation levels for validation using a Mass Array. We obtained an excellent correlation between both methods, providing technical validation of T-WGBS and our analysis workflow. This validation was published in our earlier manuscript (PMID: 37143403), but we provided the data below for convenience.

Scatter plots showing correlation of average methylation obtained with T-WGBS and Mass Array from COPD and smoker fibroblasts. Each dot represents one region with varying methylation levels. The blue diagonal represents the linear regression. Shaded areas are confidence intervals of the correlation coefficient at 95%. Correlation coefficients and P values were calculated by the Pearson correlation method.

To enable further validation and follow-up by the community, we included the full list of DMRs, associated p-values and additional information for DNA methylation analysis (DMR width, n.CpGs, MethylDiff, etc) in **Table 3** (Table_3_wgbs_dmr_info.xlsx) and the information about DEGs from RNA-seq in **Table 6** (Table_6_RNAseq_DEG_info.xlsx).

- It isn't clear to me if DNA and RNA are from the same cells? The results say "cells matching those used for T-WGBS" but the methods suggest separate extractions so not the same cells? If they are not the same cells a comment on the implications of this should be included in the discussion for example, potentially some differences in cell type composition, storage time etc.

Response:

Full Revision

Lung tissue samples were freshly cryopreserved, and H&E slides derived from exemplary pieces of the tissue analyzed. Once we had a group of at least 3 samples comprising one non-COPD and 2 COPD samples, we processed them in parallel to limit sorting variation between control and disease samples. The sorted cells were counted, aliquoted and pelleted at 4°C before flash freezing and storing at -80°C. The storage time of the cell pellets varied between the donors. RNA and DNA were isolated from cell pellets collected from the same FACS sorting experiment; therefore, we do not expect differences in cell type composition. In addition, RNA and DNA isolation were performed for all sorted pellets in parallel. All library preparations for TWGBS and RNA-seq were performed for all donors in parallel and sequenced in a single batch to minimise batch effects and technical variability. This has now been clarified in the methods part of the manuscript.

CHANGE IN THE MANUSCRIPT:

To minimize potential technical bias, samples from no COPD and COPD donors were processed in parallel in groups of 3 (one no COPD and 2 COPD samples).

RNA and genomic DNA for RNA-seq and TWGBS were isolated from identical aliquots of sorted cell pellets.

Genomic DNA was extracted from $1-2 \times 10^4$ sorted alveolar epithelial cells isolated from cryopreserved lung parenchyma from 11 different donors in parallel using QIAamp Micro Kit

The TWGBS library preparations were performed for all donors in parallel and sequenced in a single batch to minimize batch effects and technical variability.

RNA was isolated from flash-frozen pellets of 2×10^4 sorted AT2 cells from 11 different donors in parallel.

The RNA-seq library preparation for all donors was performed in parallel and all samples were sequenced in a single batch to minimize batch effects and technical variability.

- Line 193 the authors say "Since DMRs were overrepresented at cis-regulatory sites...." - "cis" needs to be defined. If you link DNAm regions to gene via "closest gene" does this not automatically mean you're outputs will be cis? Just needs better definition/explanation.

Response:

The term "cis-regulatory sites" in our manuscript is intended to denote regulatory elements—such as enhancers, promoters, and other nearby control regions—that reside on the same chromosome and close to the genes they regulate. While it's true that linking a DMR to its closest gene captures a cis association, our phrasing emphasises that the DMRs are enriched specifically at these functional regulatory elements (**Fig. 2E**) rather than being randomly distributed. This usage aligns with established conventions in the field. To avoid any misunderstandings, we have now changed the term to gene regulatory sites.

CHANGE IN THE MANUSCRIPT:

We changed the "cis-regulatory sites" to "gene regulatory sites"

Minor comments:

Full Revision

Line 157: "we identified site-specific differences....". Change to region specific?

Response:

This has now been corrected as suggested.

Line 102-103: needs a reference for the statement "Alterations in DNA methylation patterns have been implicated....."

Response:

Following the reviewer's suggestion, we added the relevant references (34-36) to this statement.

Line 266 - what does "strong dysregulation" mean? Large fold change, very significant?

Response:

We removed the word "strong" from this sentence.

Lines 423-425 - statement needs a reference

Response:

Following the reviewer's suggestion, we added the relevant reference to this statement.

Line 428 - word missing between "epigenetic , we"?

Response:

This has now been corrected. The text reads: "Through treatment with a demethylating drug and **targeted epigenetic editing**, we demonstrated the ability to modulate..."

Prior studies are well references, text and figures are clear and accurate.

Reviewer #2 (Significance (Required)):

This study has several strengths:

1) Sample collection and characterisation. AT2 cells are incredibly hard to come by and the authors should be commended to generating the samples. However, proximity to cancer is always a potential issue, especially in epigenetic studies. Is it feasible to include any analysis to show the samples derived from those with cancer don't drive the changes observed? Even a high level PCA or an edit of fig 2A with non-cancer in a different colour in supplemental - looks like there is one outlier, is that a non-cancer? Or a correlation of change in beta between control and cancer/COPD and control and non-cancer:COPD (for want a better phrase!). just an indicator that the non-cancer COPD samples are not driving differences.

Response:

We thank the reviewer for highlighting the value of generating data from hard-to-work-with AT2 populations and bringing up the important point of cancer proximity, which we considered very carefully when designing our study. To match our samples across the cohort, all the no-COPD, COPD I, and two of the COPD II-IV distal lung samples were obtained from cancer resections. In addition to other characteristics, like age, BMI and smoking status, we also matched the donors by cancer type (all profiled donors had squamous cell carcinoma). We collected lung tissue as far away from the carcinoma as possible and sent representative pieces for histological analysis

Full Revision

by an experienced lung pathologist to confirm the absence of visible tumours. In addition, to ensure that our data represents COPD-relevant signatures, we intentionally included samples from three COPD donors undergoing lung resections (without a cancer background) in the profiling.

Following the reviewer's suggestion, to investigate the potential impact of non-cancer samples on driving the observed differences, we carefully checked the PCAs for both DNA methylation and RNA-seq. We could not identify a clear separation of no-cancer COPD samples from the cancer COPD samples (or other cancer samples) in any examined PCs, indicating no confounding effect of cancer samples. We observed that one sample contributing to PC2 is a non-cancer sample, but this was a rather sample-specific effect, as the other two non-cancer samples clustered together with the other severe COPD samples with a cancer background. Notably, in our DNA methylation data, we do not observe typical features of cancer methylomes, like global loss of DNA methylation or aberrant methylation of CpG islands (e.g., in tumour suppressor genes) (see **Fig. 2A**), further suggesting that we do not "pick up" confounding cancer signatures in our data.

Following the comments from both reviewers, to clarify that point, we added the information about cancer and non-cancer samples to the PCA figures for DNA methylation (**new Fig. 2B**) and RNA-seq (**new Fig. 3A**) data in the revised manuscript, as shown below

CHANGE IN THE MANUSCRIPT:

COPD samples from donors with a cancer background clustered together with the COPD samples from lung resections, confirming that we detected COPD-relevant signatures (Fig. 2B).

Fig.2B

Fig. 2B. Principal component analysis (PCA) of methylation levels at CpG sites with > 4-fold coverage in all samples. COPD I and COPD II-IV samples are represented in light and dark green triangles, respectively, and no COPD samples as blue circles. COPD samples without a cancer background are displayed with a black contour. The percentage indicates the proportion of variance explained by each component.

Unsupervised principal component analysis (PCA) on the top 500 variable genes revealed a clear influence of the COPD phenotype in separating no COPD and COPD II-IV samples, as previously observed with the DNA methylation analysis, *irrespective of the cancer background of COPD samples (Fig.3A, Fig. S2B).*

Figure 3A: RNA-seq/gene expression

Figure S2B: RNA-seq/gene expression

Principal component analysis (PCA) of 500 most variable genes in RNA-seq analysis. PCA 1 and 2 are shown in **Fig.3A**, PCA 1 and 4 in **Fig.S2B**. COPD I and COPD II-IV samples are represented in light and dark green triangles, respectively, and no COPD samples as blue circles. **COPD samples without a cancer background are displayed with a black contour.** The percentage indicates the proportion of variance explained by each component.

2) This is the first time DNAm has been profiled in AT2 cells. It is incredibly difficult, valuable and novel data that will increase the fields capability technically, their understanding of functional mechanisms and potential translation considerably. It's audience will be primarily translational respiratory however the fundamental science aspect of gene expression regulation by DNA methylation with have wider reach across developmental and disease science.

Response:

We thank the reviewer for recognising the uniqueness and novelty of our study and highlighting the value and potential impact of our datasets for the lung field.

3) the functional analysis using targeted CRISPR-Cas9 is very well done and adds impact.

Response:

We thank the reviewer for recognising the strengths and added value of the functional analysis using epigenetic editing.

Potential weaknesses/areas for development

I feel the main weakness is the in the section integrating DNA methylation and gene expression. The rationale for a focus on various aspects, for example inversely related DNAm/gene expression pairs, the IFN pathway and IRF9, are not clear. Also further understanding of the differences between DNAm associated genes and non-DNAm associated genes could be expanded, at the pathway level, TF regulation level, effect size level (are DNAm associated changes to gene expression larger, enriched for earlier differential expression)

Response:

Our rationale for focusing on the inversely related DNAm/gene expression pairs in promoter proximal is purely data-driven, as they represent the biggest group in our data (**Fig. 4A-B**). Among those negatively correlated genes, we observed the strongest enrichment for the IFN

Full Revision

pathway (**Fig. C**), making it an obvious, data-driven target for further studies. The negative correlation of expression and methylation for IFN pathway genes could be validated in 5-AZA assays in A549 cells (**Fig. 5A**). Next, we made an interaction network analysis showing IRF9 and STAT2 as master regulators (**Fig. 5B**) of the negatively correlated IFN genes. As IRF9 itself displayed a negative correlation between DNA methylation and expression (**Fig. 5C**), we used the associated DMR for further epigenetic editing (**Fig. 5D-E**). We performed the additional requested analyses of the enhancer-associated changes and genes, as described above. We fully agree with the reviewer that our data sets are a great resource and can be further used to elaborate on other relationships of DNA methylation and RNA expression or other pathways, but this is out of the scope of this study. To enable further studies by the research community, we provide all necessary information about DMRs and DEGs in the associated supplementary tables and the raw data through the EGA, as well as the CRISPRa editing assay.

The authors could comment on potential masking of differences between 5hmC and mC and the implications it may have

Response:

We thank the reviewer for bringing up this important point. Indeed, bisulfite sequencing cannot differentiate between methylated and hydroxymethylated cytosines; hence, some of the methylated sites may be hydroxymethylated. However, the overall levels of hydroxymethylation in differentiated adult tissues are very low (except for the brain), orders of magnitude lower compared to DNA methylation. Following the reviewer's suggestion, we have added a sentence in the limitation section of the discussion to clarify that point.

CHANGE IN THE MANUSCRIPT:

In addition, while WGBS provides unprecedented resolution and high coverage of the DNA methylation sites across the genome, it does not allow distinguishing 5-methylcytosine from 5-hydroxymethylcytosine. Therefore, we cannot exclude that some methylated sites we detected are 5-hydroxymethylated. However, the 5-hydroxymethylcytosine is present at very low levels in the lung tissue (97).

Furthermore, while the rationale for looking at DMRs is clear, especially given the sample number, I am interested to understand what proportion of the assayed CpGs "fit" within the cut off stipulations of the DMR analysis - that is, is their potentially COPD effects at sparse CpG regions/individual CpG sites that are not being identified. A comment on this would be useful and seems the strength of profiling genome wide. I'm happy genome wide is beneficial it just feels a little circular that the authors have chosen whole genome to avoid the bias of the Illumina array and a focus on promoters, but have primarily reported promoter DNAm. This caught my attention again in the discussion where the authors state that cis-regulatory regions were also identified in their fibroblast datais this finding a factor of the analysis performed? (also a comparison of regions Identified in AT2 cells versus fibroblasts would be really interesting for a future paper)

Response:

We decided to focus our analysis on regions rather than individual CpG sites when looking at differential methylation, as DNA methylation is spatially correlated, and methylation changes in

Full Revision

larger regions are more likely to have a biological function. Extending the analysis to single CpG sites would require a higher number of samples for a reliable analysis compared to the DMR analysis (as mentioned by the reviewer).

Of note, we addressed the platform comparison between Illumina array technology and WGBS in our previous fibroblast study (PMID: 37143403), where we compared our WGBS data with the published 450k array data of COPD parenchymal fibroblasts (Clifford et al., 2018). We observed only a marginal overlap between the CpGs from our DMRs and the CpGs probes available on the array (which was due to the differences in technologies used and the limited coverage of the 450K array in comparison to our genome-wide approach, in which we covered 18 million CpGs). Out of the 6279 DMRs identified in our fibroblast study, only 1509 DMRs overlapped with at least one CpG probe on the 450K array, and after removing low-quality CpGs from the array data, only 1419 DMRs were left. This comparison highlighted the increased resolution of the WGBS compared to Illumina arrays.

The reason why we focused on promoter proximal DMRs are the following: 1) the assignment of the enhancer elements in AT2 to the corresponding gene is still too inaccurate in the absence of AT2 specific enhancer chromatin maps 2) regulation at enhancers by DNA methylation might be more complex and might change (increase or attenuate) binding affinities of certain transcription factors (**Fig.2H**), which might lead to gene expression changes or 3) methylation changes might be an indirect effect of differential TF binding (PMID: 22170606). However, we agree with the reviewer that despite these limitations, expanding the analysis beyond promoters adds value to the manuscript; hence, as described above, we expanded the analysis of non-promoter regions, including enhancers, in the revised manuscript.

We thank the reviewer for the suggestion to compare the regions identified in AT2 cells and fibroblasts in a future paper.

My expertise:Respiratory, cell biology, epigenetics.

16th Oct 2025

Dear Dr. Jurkowska,

Thank you for the submission of your revised manuscript to EMBO Molecular Medicine. We have now heard back from the one referee who agreed to re-evaluate your manuscript. This referee also assessed author responses to concerns raised by other referees. I am pleased to inform you that we will be able to accept your manuscript pending the following final amendments:

- 1) Authors: There is a discrepancy in how the name is displayed in our system (Marc Schneider) and in the manuscript (Mark Schneider), please correct.
- 2) Figures:
 - In Figure 1C 'no COPD' image is already published in your previous publication (<https://insight.jci.org/articles/view/140443/figure/2>). Please reference it in the figure legend or use an unpublished image.
 - Please rename EV Figures to Figure EV1 - EV4 and add the heading "Expanded View Figure Legends". Please update their callouts in the manuscript text.
- 3) Author checklist: Please submit a complete checklist. <https://www.embopress.org/pb-assets/embosite/EMBO%20Press%20Author%20Checklist-1642513524327.xlsx>
- 4) In the main manuscript file, please do the following:
 - Please address all comments suggested by our data editors listed below:
 - o Data availability statement:
 1. Please note that the specific URLs for EGAS00001007386, EGAS00001007387 datasets are not provided in the data availability statement.
 - o Figure legends:
 1. Please define the annotated p values ****/***/**/* as well as provide the exact p-values for the same in the legend of figure 2A as appropriate.
 2. Please note that the exact p values are not provided in the legends of figures 1B, 5A, E-G; S2 D, S2 G, S4 B-D.
 3. Please indicate the statistical test used for data analysis in the legends of figures 2A, H; 3B, C; 4C, S1 C, D; S2 E, F.
 4. Please note that the box plots need to be defined in terms of minima, maxima, centre, bounds of box and whiskers, and percentile in the legend of figure S4 F.
 5. Please note that information related to n is missing in the legends of figures 3B, C; S2 A, D; S4 A.
 6. Please note that the error bars are not defined in the legend of figure S4 A.
 - Limit keywords to max. 5.
 - Remove "Teaser" paragraph.
 - Please correct the order and headings of the manuscript sections to: Abstract / Keywords / The Paper Explained / Introduction / Results / Discussion / Methods / Data Availability / Acknowledgements / Disclosure and Competing Interests Statement / References / Main Figure Legends / Tables / Expanded View Figure Legends
 - In Methods, provide the statement that informed consent was obtained from all human subjects and that the experiments conformed to the principles set out in the WMA Declaration of Helsinki and the Department of Health and Human Services Belmont Report.
 - Please include structured Methods section that includes a Reagents and Tools Table (should be uploaded as a separate file) followed by a Methods and Protocols section. More information on how to adhere to this format as well as downloadable templates (.docx) for the Reagents and Tools Table can be found in our author guidelines: <https://www.embopress.org/page/journal/17574684/authorguide#structuredmethods>

An example of a paper with Structured Methods can be found here: <https://www.embopress.org/doi/full/10.1038/s44320-024-00037-6#sec-4>

 - Indicate in legends exact n and exact p values, not a range, along with the statistical test used. To keep the figures "clear" some authors found providing an Appendix table Sx with all exact p-values preferable. You are welcome to do this if you want to.
 - Merge Fundijng with Acknowledgements.
 - Please rename "Competing interests" to "Disclosure and competing interests statement". We updated our journal's competing interests policy in January 2022 and request authors to consider both actual and perceived competing interests. Please review the policy <https://www.embopress.org/competing-interests> and update your competing interests if necessary.
 - Author contributions: Please remove it from the manuscript and specify author contributions in our submission system. CRediT has replaced the traditional author contributions section because it offers a systematic machine-readable author contributions format that allows for more effective research assessment. You are encouraged to use the free text boxes beneath each contributing author's name to add specific details on the author's contribution. More information is available in our guide to authors: <https://www.embopress.org/page/journal/17574684/authorguide#authorshipguidelines>
 - In data availability please use the following format to report the accession number of your data:

[data type]: [full name of the resource] [accession number/identifier] [(doi or URL or identifiers.org/DATABASE:ACCESSION)]

Please check "Author Guidelines" for more information.

<https://www.embopress.org/page/journal/17574684/authorguide#availabilityofpublishedmaterial>

- Correct the reference citation in the text and reference list. In the text, a reference should be cited by author and year of publication. Include a space between a word and the opening parenthesis of the reference that follows. In the reference list, citations should be listed in alphabetical order. Where there are more than 10 authors on a paper, 10 will be listed, followed by "et al.". Please check "Author Guidelines" for more information.

<https://www.embopress.org/page/journal/17574684/authorguide#referencesformat>

5) Tables:

- Please rename Table 4-10 to "Dataset EV1 - Dataset EV6" and ensure that each dataset has a legend added. Update their callouts in the manuscript text.

- Please double-check the numbering of the tables: the numbers in the legends don't always correspond to the number in the file name (e.g. Table 2 = labeled Table 5 in legend). Please add legends for Table 3.

6) The Paper Explained: Please provide "The Paper Explained" and add it to the main manuscript text. Please check "Author Guidelines" for more information. <https://www.embopress.org/page/journal/17574684/authorguide#researcharticleguide>

7) Synopsis: Every published paper now includes a 'Synopsis' to further enhance discoverability. Synopses are displayed on the journal webpage and are freely accessible to all readers. They include separate synopsis image and synopsis text.

- Synopsis image: Please provide a visual abstract as a high-resolution jpeg file 550 px-wide x 300-600 pixels high to illustrate your article.

- Synopsis text: Please provide a short standfirst (maximum of 300 characters, including space) as well as 2-5 one sentence bullet points that summarise the paper as a .doc file. Please write the bullet points to summarise the key NEW findings. They should be designed to be complementary to the abstract - i.e. not repeat the same text. We encourage inclusion of key acronyms and quantitative information (maximum of 30 words / bullet point). Please use the passive voice.

8) As part of the EMBO Publications transparent editorial process initiative (see our Editorial at <http://embomolmed.embopress.org/content/2/9/329>), EMBO Molecular Medicine will publish online a Review Process File (RPF) to accompany accepted manuscripts. This file will be published in conjunction with your paper and will include the anonymous referee reports, your point-by-point response and all pertinent correspondence relating to the manuscript. Let us know whether you agree with the publication of the RPF and as here, if you want to remove or not any figures from it prior to publication. Please note that the Authors checklist will be published at the end of the RPF.

9) Please provide a point-by-point letter INCLUDING my comments as well as the reviewer's reports and your detailed responses (as Word file).

I look forward to reading a new revised version of your manuscript as soon as possible.

Yours sincerely,

Zeljko Durdevic

Zeljko Durdevic
Senior Editor
EMBO Molecular Medicine

*** Instructions to submit your revised manuscript ***

- 1) a .docx formatted version of the manuscript text (including Figure legends and tables)
 - 2) Separate figure files*
 - 3) supplemental information as Expanded View and/or Appendix. Please carefully check the authors guidelines for formatting Expanded view and Appendix figures and tables at <https://www.embopress.org/page/journal/17574684/authorguide#expandedview>
 - 4) a letter INCLUDING the reviewer's reports and your detailed responses to their comments (as Word file).
 - 5) The paper explained: EMBO Molecular Medicine articles are accompanied by a summary of the articles to emphasize the major findings in the paper and their medical implications for the non-specialist reader. Please provide a draft summary of your article highlighting
 - the medical issue you are addressing,
 - the results obtained and
 - their clinical impact.This may be edited to ensure that readers understand the significance and context of the research. Please refer to any of our published articles for an example.
 - 6) Author contributions: the contribution of every author must be detailed in a separate section.
 - 7) EMBO Molecular Medicine now requires a complete author checklist (<https://www.embopress.org/page/journal/17574684/authorguide>) to be submitted with all revised manuscripts. Please use the checklist as guideline for the sort of information we need WITHIN the manuscript. The checklist should only be filled with page numbers where the information can be found. This is particularly important for animal reporting, antibody dilutions (missing) and exact values and n that should be indicated instead of a range.
 - 8) Every published paper now includes a 'Synopsis' to further enhance discoverability. Synopses are displayed on the journal webpage and are freely accessible to all readers. They include a short stand first (maximum of 300 characters, including space) as well as 2-5 one sentence bullet points that summarise the paper. Please write the bullet points to summarise the key NEW findings. They should be designed to be complementary to the abstract - i.e. not repeat the same text. We encourage inclusion of key acronyms and quantitative information (maximum of 30 words / bullet point). Please use the passive voice. Please attach these in a separate file or send them by email, we will incorporate them accordingly.
- You are also welcome to suggest a striking image or visual abstract to illustrate your article. If you do please provide a jpeg file 550 px-wide x 300-600px high.
- 9) A Conflict of Interest statement should be provided in the main text
 - 10) Please note that we now mandate that all corresponding authors list an ORCID digital identifier. This takes <90 seconds to complete. We encourage all authors to supply an ORCID identifier, which will be linked to their name for unambiguous name identification.

Currently, our records indicate that the ORCID for your account is 0000-0002-4507-2222.

Link Not Available

- 11) Include a Reagents and Tools Table as part of the Methods section, which can be downloaded from our author guidelines (<https://www.embopress.org/page/journal/17574684/authorguide#structuredmethods>)

Photos 400-800 DPI

*Additional important information regarding figures and illustrations can be found at <https://bit.ly/EMBOPressFigurePreparationGuideline>. See also figure legend preparation guidelines: <https://www.embopress.org/page/journal/17574684/authorguide#figureformat>

***** Reviewer's comments *****

Referee #1 (Comments on Novelty/Model System for Author):

Technical quality - high level of attention to detail with sample choice, including honesty and statement of limitations where they remain. Statistical analyses are appropriate and stringent to help account for the acknowledged small sample number. Functional validation strengthens main conclusions.

Novelty - cell specific DNA methylation studies are incredibly rare and incredibly important. Profiling AT2 cell is holds further novelty as they are so difficult to obtain.

Medical Impact - this fundamental study is several steps from medical impact but paves the way.

Adequacy of model system - non-passaged primary cells are the best model available

Referee #1 (Remarks for Author):

I thank the authors for their comprehensive response and update of the manuscript.

I am happy that all concerns have been addressed and limitations have been honestly reported where they remain.

Rev_Com_number: RC-2024-02418

New_manu_number: EMM-2025-22435-T

Corr_author: Jurkowska

Title: Epigenetic dysregulation of IRF9 drives excessive interferon signalling in COPD

The authors addressed the remaining editorial issues.

29th Jan 2026

Dear Dr. Jurkowska,

We are pleased to inform you that your manuscript is accepted for publication and is now being sent to our publisher to be included in the next available issue of EMBO Molecular Medicine.

You may qualify for financial assistance for your publication charges - either via a Springer Nature fully open access agreement or an EMBO initiative. Check your eligibility: <https://link.springer.com/journal/44321/how-to-publish-with-us>

>>> Please note that it is EMBO Molecular Medicine policy for the transcript of the editorial process (containing referee reports and your response letter) to be published as an online supplement to each paper. If you do NOT want this, you will need to inform the Editorial Office via email immediately. More information is available here: <https://link.springer.com/partners/embo-press/editorial-policies#Peer%20review>